# ADMM for Nonconvex Optimization under Minimal Continuity Assumption

**Ganzhao Yuan**
Peng Cheng Laboratory, China
yuangzh@pcl.ac.cn

## Abstract

This paper introduces a novel approach to solving multi-block nonconvex composite optimization problems through a proximal linearized Alternating Direction Method of Multipliers (ADMM). This method incorporates an Increasing Penalization and Decreasing Smoothing (IPDS) strategy. Distinguishing itself from existing ADMM-style algorithms, our approach (denoted IPDS-ADMM) imposes a less stringent condition, specifically requiring continuity in just one block of the objective function. IPDS-ADMM requires that the penalty increases and the smoothing parameter decreases, both at a controlled pace. When the associated linear operator is bijective, IPDS-ADMM uses an over-relaxation stepsize for faster convergence; however, when the linear operator is surjective, IPDS-ADMM uses an under-relaxation stepsize for global convergence. We devise a novel potential function to facilitate our convergence analysis and prove an oracle complexity $\mathcal{O}(\epsilon^{-3})$ to achieve an $\epsilon$-approximate critical point. To the best of our knowledge, this is the first complexity result for using ADMM to solve this class of nonsmooth nonconvex problems. Finally, some experiments on the sparse PCA problem are conducted to demonstrate the effectiveness of our approach. [1]

## 1 Introduction

We consider the following multi-block nonconvex nonsmooth composite optimization problem:

$$\min_{\mathbf{x}_1, \mathbf{x}_2, \ldots, \mathbf{x}_n} \sum_{i=1}^{n}[f_i(\mathbf{x}_i) + h_i(\mathbf{x}_i)], \ s.t. \ \big[\sum_{i=1}^{n} \mathbf{A}_i \mathbf{x}_i\big] = \mathbf{b}, \tag{1}$$

where $\mathbf{b} \in \mathbb{R}^{m \times 1}$, $\mathbf{A}_i \in \mathbb{R}^{m \times \mathbf{d}_i}$, $\mathbf{x}_i \in \mathbb{R}^{\mathbf{d}_i \times 1}$, and $i \in [n] \triangleq \{1, 2, \ldots, n\}$. We assume $f_i(\cdot) : \mathbb{R}^{\mathbf{d}_i \times 1} \mapsto (-\infty, \infty)$ is differentiable and potentially nonconvex for all $i \in [n]$. The function $h_i(\cdot) : \mathbb{R}^{\mathbf{d}_i \times 1} \mapsto (-\infty, \infty]$ is assumed to be closed, proper, lower semi-continuous, and potentially nonsmooth. While $h_n(\cdot)$ is convex, we do not require convexity for $h_i(\cdot)$ where $i \in [n-1]$. Additionally, we assume the nonconvex proximal operator of $h_i(\cdot)$ is easy to compute for all $i \in [n]$.

Problem (1) has a wide range of applications in machine learning. The function $f_i(\cdot)$ plays a crucial role in handling empirical loss, including neural network activation functions (Liu et al., 2022; Zeng et al., 2021; Wang et al., 2019a; Huang et al., 2019). Incorporating multiple nonsmooth regularization terms $h_i(\cdot)$ enables diverse prior information integration, including structured sparsity, low-rank, binary, orthogonality, and non-negativity constraints, enhancing regularization model accuracy. These capabilities extend to various applications such as sparse PCA, overlapping group Lasso, graph-guided fused Lasso, and phase retrieval.

▶ **ADMM Literature**. The Alternating Direction Method of Multipliers (ADMM) is a versatile optimization tool suitable for solving composite constrained problems as in Problem (1), which pose challenges for other standard optimization methods, such as the accelerated proximal gradient method (Nesterov, 2003) and the augmented Lagrangian method (Zeng et al., 2022; Lu & Zhang, 2012; Zhu et al., 2023). The standard ADMM was initially introduced in (Gabay & Mercier, 1976), and its complexity analysis for the convex settings was first conducted in (He & Yuan, 2012; Monteiro & Svaiter, 2013). Since then, numerous papers have explored the iteration complexity of

---

[1] Future versions of this paper can be found at https://arxiv.org/abs/2405.03233.

ADMM in diverse settings. These settings include acceleration through multi-step updates (Pock & Sabach, 2016; Li et al., 2016; Ouyang et al., 2015; Shen et al., 2017; Franca et al., 2018; Hien et al., 2022; Tran Dinh, 2018), asynchronous updates (Zhang & Kwok, 2014), Jacobi updates (Deng et al., 2017), non-Euclidean proximal updates (Gonçalves et al., 2017b), and extensions to handle more specific or general functions such as strongly convex functions (Nishihara et al., 2015; Lin et al., 2015a; Ouyang et al., 2015), nonlinear constrained functions (Lin et al., 2022a), multi-block composite functions (Lin et al., 2015b; Xu et al., 2017), finite-sum functions (Bian et al., 2021; Liu et al., 2020), and expected-value functions (Huang et al., 2019). For a comprehensive overview, refer to the book (Lin et al., 2022b).

▶ **Nonconvex ADMM**. Compared to the classical subgradient methods (Li et al., 2021; Davis & Drusvyatskiy, 2019) and Smoothing Proximal Gradient Methods (SPGM) (Böhm & Wright, 2021; Yuan, 2024), designed for general nonconvex optimization, ADMM-type methods potentially offer faster convergence, enhanced parallelization, and greater numerical stability. However, the convergence analysis of the nonconvex ADMM is challenging due to the absence of Fejér monotonicity in iterations. In the past decade, significant research has focused on exploring various nonconvex ADMM variants (Li & Pong, 2015; Hong et al., 2016; Yang et al., 2017). (Li & Pong, 2015) establishes the convergence of a class of nonconvex problems when a specific potential function associated with the augmented Lagrangian satisfies the Kurdyka-Łojasiewicz inequality. (Yang et al., 2017) analyzes ADMM variants for solving low-rank and sparse optimization problems. (Hong et al., 2016) investigates ADMM variants for nonconvex consensus and sharing problems. Some researchers have examined ADMM variants under weaker conditions, such as restricted weak convexity (Wang et al., 2019b), restricted strong convexity (Barber & Sidky, 2024), and the Hoffman error bound (Zhang & Luo, 2020). However, existing methods typically assume that at least one block of the objective function is smooth. In contrast, our approach requires only continuity in one block, which is achieved through an Increasing Penalization and Decreasing Smoothing (IPDS) strategy.

▶ **Over-Relaxed and Under-Relaxed ADMM**. Prior studies have analyzed ADMM using either under-relaxation stepsizes $\sigma \in (0, 1)$, or over-relaxation stepsizes $\sigma \in [1, 2)$, for updating the dual variable. This contrasts with earlier approaches that employed fixed values, such as 1 or the golden ratio $(\sqrt{5} + 1)/2$. In nonconvex settings, most existing works require that the associated matrix of the problem be bijective (Gonçalves et al., 2017a; Yang et al., 2017; Yashtini, 2022; 2021; Boţ & Nguyen, 2020). However, the work of (Boţ et al., 2019) demonstrates that ADMM can still be applied when the associated matrix is surjective, provided that an under-relaxation stepsize is employed. Inspired by these findings, our work shows that when the associated linear operator is bijective, IPDS-ADMM uses an over-relaxation stepsize for faster convergence. In contrast, when the linear operator is surjective, we employ under-relaxation stepsizes to achieve global convergence.

▶ **Existing Challenges.** We consider the linearly constrained optimization problem in Problem (1), which involves $(n - 1)$ potentially nonsmooth, nonconvex, and non-Lipschitz composite functions $h_i(\cdot)$ for $i \in [n - 1]$, and one convex, nonsmooth composite function $h_n(\cdot)$. Existing ADMM-type methods cannot solve this problem, as they require at least one of the composite functions to be smooth (i.e., $h_n(\cdot) = 0$). In the special case where $n = 2$, $\mathbf{A}_2 = \mathbf{I}$, $h_2(\mathbf{x}_2) = \dot{\rho} \|\mathbf{x}_2\|_1$ with $\dot{\rho} > 0$, and $h_1(\cdot)$ is the indicator function of orthogonality constraints, the Riemannian ADMM (RADMM) algorithm (Li et al., 2022) is the only known method capable of solving Problem (1). However, RADMM cannot handle linearly constrained problems, particularly when $\mathbf{A}_n$ is bijective or surjective. Importantly, it results in suboptimal iteration complexity. A comparison of existing nonconvex ADMM approaches is provided in Table 1.

▶ **Our Contributions.** Our main contributions are summarized as follows. *(i)* We introduce IPDS-ADMM to solve the nonconvex nonsmooth optimization problem as in Problem (1). This approach imposes the least stringent condition, specifically requiring continuity in only one block of the objective function. It employs an Increasing Penalization and Decreasing Smoothing (IPDS) strategy to ensure convergence (See Section 3). *(ii)* IPDS-ADMM achieves global convergence when the associated matrix is either bijective or surjective. We establish that IPDS-ADMM converges to an $\epsilon$-critical point with a time complexity of $\mathcal{O}(\epsilon^{-3})$ (See Section 4). *(iii)* We have conducted experiments on the sparse PCA problem to demonstrate the effectiveness of our approach. (See Section 5).

▶ **Assumptions.** Through this paper, we impose the following assumptions on Problem (1).

Table 1: Comparison of existing ADMM approaches for solving the nonconvex problem in Problem (1). `CVX`: convex. `NC`: nonconvex. `LCONT`: Lipschitz continuous. `WC`: weakly convex. `RWC`: restricted weakly convex. $\mathbb{F}$: the constraint set is non-empty. $\mathbb{I}$: $\mathbf{A}_n$ is identity. $\mathbb{SU}$: $\mathbf{A}_n$ is surjective with $\lambda_{\min}(\mathbf{A}_n\mathbf{A}_n^\mathsf{T}) > 0$. $\mathbb{IN}$: $\mathbf{A}_n$ is injective with $\lambda_{\min}(\mathbf{A}_n^\mathsf{T}\mathbf{A}_n) > 0$. $\mathbb{BI}$: $\mathbf{A}_n$ is bijective (both surjective and injective). $\mathbb{IM}$: $\mathrm{Im}([\mathbf{A}_1, \mathbf{A}_2, \ldots, \mathbf{A}_{n-1}]) \subseteq \mathrm{Im}(\mathbf{A}_n)$ with Im being the image of the matrix.

| Reference | Optimization Problems and Main Assumptions | | | Complexity | Parameter $\sigma$ |
|---|---|---|---|---|---|
| | Blocks | Functions $f_i(\cdot)$ and $h_i(\cdot)^a$ | Matrices $\mathbf{A}_i$ | | |
| (He & Yuan, 2012) | $n = 2$ | `CVX`: $f_i, h_i, \forall i \in [2]$ | $\mathbb{F}$ | $\mathcal{O}(\epsilon^{-2})^{\,b}$ | $\sigma = 1$ |
| (Li & Pong, 2015) | $n = 2$ | `NC`: $h_1, f_2; f_1 = 0; \mathbf{\textit{h}_2 = 0}$ | $\mathbb{SU}$ | $\mathcal{O}(\epsilon^{-2})$ | $\sigma = 1$ |
| (Yang et al., 2017)$^c$ | $n = 3$ | `CVX`: $h_1, f_3$; `NC`: $h_2; f_1 = f_2 = 0; \mathbf{\textit{h}_3 = 0}$ | $\mathbb{I}$ | $\mathcal{O}(\epsilon^{-2})$ | $\sigma \in [1, 2)$ |
| (Yashtini, 2022) | $n = 2$ | `NC`: $f_{[1,2]}, h_{[1,2]}; \mathbf{\textit{h}_2 = 0}$ | $\mathbb{BI}$ | $\mathcal{O}(\epsilon^{-2})$ | $\sigma \in (0, 1)$ |
| (Yashtini, 2021) | $n \geq 2$ | `WC`: $f_{[1,n-1]}; h_{[1,n-1]} = 0; \mathbf{\textit{h}_n = 0}$ | $\mathbb{BI}, \mathbb{IM}$ | $\mathcal{O}(\epsilon^{-2})$ | $\sigma \in (0, 1)$ |
| (Wang et al., 2019b) | $n \geq 2$ | `RWC`: $h_{[1,n-1]}; \mathbf{\textit{h}_n = 0}$ | $\mathbb{IN}, \mathbb{IM}$ | $\mathcal{O}(\epsilon^{-2})$ | $\sigma = 1$ |
| (Boţ & Nguyen, 2020) | $n = 2$ | `NC`: $h_1, f_{[1,2]}; f_1 = 0; \mathbf{\textit{h}_2 = 0}$ | $\mathbb{I}$ | $\mathcal{O}(\epsilon^{-2})$ | $\sigma \in [1, 2)$ |
| (Boţ et al., 2019) | $n = 2$ | `NC`: $h_1, f_{[1,2]}; f_1 = 0; \mathbf{\textit{h}_2 = 0}$ | $\mathbb{SU}$ | $\mathcal{O}(\epsilon^{-2})$ | $\sigma \in (0, 1)$ |
| (Huang et al., 2019) | $n \geq 2$ | `NC`: $f_n; f_{[1,n-1]} = 0$; `CVX`: $h_{[1,n-1]}; \mathbf{\textit{h}_n = 0}$ | $\mathbb{BI}$ | $\mathcal{O}(\epsilon^{-2})$ | $\sigma = 1$ |
| (Li et al., 2022)$^d$ | $n = 2$ | `NC`: $f_1, h_1$; `CVX`: $h_2; f_2 = 0$; `LCONT`: $\mathbf{\textit{h}_2 \neq 0}$ | $\mathbb{I}$ | $\mathcal{O}(\epsilon^{-4})$ | $\sigma = 1$ |
| This work | $n \geq 2$ | `NC`: $h_{[1,n-1]}, f_{[1,n]}$; `CVX`: $h_n$; `LCONT`: $f_n, \mathbf{\textit{h}_n \neq 0}$ | $\mathbb{BI}$ | $\mathcal{O}(\epsilon^{-3})$ | $\sigma \in [1, 2)$ |
| This work | $n \geq 2$ | `NC`: $h_{[1,n-1]}, f_{[1,n]}$; `CVX`: $h_n$; `LCONT`: $f_n, \mathbf{\textit{h}_n \neq 0}$ | $\mathbb{SU}$ | $\mathcal{O}(\epsilon^{-3})$ | $\sigma \in (0, 1)$ |

Note $a$: $\mathbf{\textit{h}_n = 0}$ denotes that the $n$-th block has no non-smooth part, making the objective function smooth.

Note $b$: The iteration complexity relies on the variational inequality of the convex problem.

Note $c$: We adapt their application model into our optimization framework in Equation (1) with $(L, S, Z) = (\mathbf{x}_1, \mathbf{x}_2, \mathbf{x}_3)$, as their model additionally requires the linear operator for the first two blocks to be injective.

Note $d$: This paper studies manifold optimization with a fixed large penalty and small stepsize.

**Assumption 1.1.** *Each function $f_i(\cdot)$ is $L_i$-smooth for all $i \in [n]$ such that $\|\nabla f_i(\mathbf{x}_i) - \nabla f_i(\dot{\mathbf{x}}_i)\| \leq L_i\|\mathbf{x}_i - \dot{\mathbf{x}}_i\|$ holds for all $\mathbf{x}_i, \dot{\mathbf{x}}_i \in \mathbb{R}^{\mathbf{d}_i \times 1}$. This implies that $|f_i(\mathbf{x}_i) - f_i(\dot{\mathbf{x}}_i) - \langle \nabla f_i(\dot{\mathbf{x}}_i), \mathbf{x}_i - \dot{\mathbf{x}}_i \rangle| \leq \frac{L_i}{2}\|\mathbf{x}_i - \dot{\mathbf{x}}_i\|_2^2$ (cf. Lemma 1.2.3 in (Nesterov, 2003)).*

**Assumption 1.2.** *The functions $f_n(\cdot)$ and $h_n(\cdot)$ are Lipschitz continuous with some constants $C_f$ and $C_h$, satisfying $\|\nabla f_n(\mathbf{x}_n)\| \leq C_f$ and $\|\partial h_n(\mathbf{x}_n)\| \leq C_h$ for all $\mathbf{x}_n$.*

**Assumption 1.3.** *We define $\overline{\lambda} \triangleq \lambda_{max}(\mathbf{A}_n\mathbf{A}_n^\mathsf{T})$, $\underline{\lambda} \triangleq \lambda_{min}(\mathbf{A}_n\mathbf{A}_n^\mathsf{T})$, $\underline{\lambda}' = \lambda_{min}(\mathbf{A}_n^\mathsf{T}\mathbf{A}_n)$. Either of these two conditions holds for matrix $\mathbf{A}_n$:*

*a) Condition $\mathbb{BI}$: $\mathbf{A}_n$ is bijective (i.e., $\underline{\lambda} = \underline{\lambda}' > 0$), and it holds that $\kappa \triangleq \overline{\lambda}/\underline{\lambda} < 2$.*

*b) Condition $\mathbb{SU}$: $\mathbf{A}_n$ is surjective (i.e., $\underline{\lambda} > 0$, and $\underline{\lambda}'$ could be zero).*

**Assumption 1.4.** *Given any constant $\overline{\beta} \geq 0$, we let $\underline{\Theta}' \triangleq \inf_{\mathbf{x}_1, \mathbf{x}_2, \ldots, \mathbf{x}_n} \sum_{i=1}^{n}[f_i(\mathbf{x}_i) + h_i(\mathbf{x}_i)] + \frac{\overline{\beta}}{2}\|[\sum_{i=1}^{n} \mathbf{A}_i\mathbf{x}_i] - \mathbf{b}\|_2^2$. We assert that $\underline{\Theta}' > -\infty$.*

**Assumption 1.5.** *For all $i \in [n]$, the proximal operator $\mathrm{Prox}_i(\mathbf{x}_i; \mu) \triangleq \min_{\mathbf{x}_i'} \frac{\mu}{2}\|\mathbf{x}_i' - \mathbf{x}_i\|_2^2 + h_i(\mathbf{x}_i')$ can be computed efficiently and exactly for any given $\mathbf{x}_i \in \mathbb{R}^{\mathbf{d}_i \times 1}$ and $\mu > 0$.*

**Assumption 1.6.** *If $\sum_{i=1}^{n}[f_i(\mathbf{x}_i) + h_i(\mathbf{x}_i)] < +\infty$, it follows that $\|\mathbf{x}_i\| < +\infty$ for all $i \in [n]$.*

**Assumption 1.7.** *For any $i \in [n]$, if the vector $\mathbf{x}_i \in \mathbb{R}^{\mathbf{d}_i \times 1}$ is bounded, then the set $\mathrm{Prox}_i(\mathbf{x}_i; \mu)$ is also bounded for all $\mu \in (0, \infty)$.*

**Remarks**. *(i)* Assumption 1.1 is commonly used in the convergence analysis of nonconvex algorithms. *(ii)* Assumption 1.2 imposes a continuity assumption only for the last block, allowing other blocks of the function $\{h_i(\mathbf{x}_i)\}_{i=1}^{n-1}$ to be nonsmooth and non-Lipschitz, such as indicator functions of constraint sets. It ensures bounded (sub-)gradients for $f_n(\cdot)$ and $h_n(\cdot)$, a relatively mild requirement that has found use in nonsmooth optimization (Li et al., 2022; 2021; Huang et al., 2019; Böhm & Wright, 2021). *(iii)* Assumption 1.3 demands a condition on the linear matrix $\mathbf{A}_i$ for the last block ($i = n$), while leaving $\mathbf{A}_i$ unrestricted for $i \in [n-1]$. *(iv)* Assumption 1.4 ensures the well-defined nature of the penalty function associated with the problem, as has been used in (Gonçalves et al., 2017a). Furthermore, Assumption 1.4 can be satisfied if $\sum_{i=1}^{n}[f_i(\mathbf{x}_i) + h_i(\mathbf{x}_i)] > -\infty$. *(v)* Assumption 1.5 is frequently employed in nonconvex ADMM frameworks (Li & Pong, 2015; Boţ et al., 2019). Common examples of functions $h_i(\mathbf{x}_i)$ arising in practical applications include those discussed in (Gong et al., 2013), $\ell_0$ regularization, $\ell_{1/2}$ regularization (Zeng et al., 2014), and indicator functions of cardinality constraints, matrices with orthogonality constraints (Lai & Osher, 2014), and matrices with rank constraints, among others. *(vi)* Assumptions 1.6 and 1.7 are used to guarantee the boundedness of the solution.

▶ **Notations.** We define $[n] \triangleq \{1, 2, \ldots, n\}$ and $\mathbf{x} \triangleq \mathbf{x}_{[n]} \triangleq \{\mathbf{x}_1, \mathbf{x}_2, \ldots, \mathbf{x}_n\}$. For any $j \geq i$, we denote $\mathbf{x}_{[i,j]} \triangleq \{\mathbf{x}_i, \mathbf{x}_{i+1}, \ldots, \mathbf{x}_j\}$. We define $\lambda_{\min}(\mathbf{M})$ and $\lambda_{\max}(\mathbf{M})$ as the smallest and largest eigenvalue of the given matrix $\mathbf{M}$, respectively. We denote $\|\mathbf{A}_i\|$ as the spectral norm of the matrix $\mathbf{A}_i$. We denote $\mathbf{A}\mathbf{x} \triangleq \sum_{j=1}^{n} \mathbf{A}_j \mathbf{x}_j$, and $\|\mathbf{x}^+ - \mathbf{x}\|_2^2 = \sum_{i=1}^{n} \|\mathbf{x}_i^+ - \mathbf{x}_i\|_2^2$. Further notations and technical preliminaries are provided in Appendix A.

## 2 MOTIVATING APPLICATIONS

Many machine learning and data science models can be formulated as Problem (1). Below, we present two examples, with additional applications provided in Appendix B.

▶ **Sparse PCA**. Sparse PCA (Chen et al., 2016; Lu & Zhang, 2012) focuses on identifying a subset of informative variables with sparse loadings to enhance interpretability and reduce model complexity. It is formulated as: $\min_{\mathbf{V} \in \mathbb{R}^{d \times \dot{r}}} \frac{1}{2\dot{m}} \|\mathbf{D} - \mathbf{D}\mathbf{V}\mathbf{V}^{\mathsf{T}}\|_{\mathsf{F}}^2 + \dot{\rho}\|\mathbf{V}\|_1, \, s.t.\, \mathbf{V} \in \mathcal{M} \triangleq \{\mathbf{V} \mid \mathbf{V}^{\mathsf{T}}\mathbf{V} = \mathbf{I}\}$, where $\mathbf{D} \in \mathbb{R}^{\dot{m} \times \dot{d}}$ is the data matrix, and $\dot{\rho} \geq 0$. Introducing an additional variable $\mathbf{Y}$, this problem can be formulated as: $\min_{\mathbf{V},\mathbf{Y}} \frac{1}{2\dot{m}} \|\mathbf{D} - \mathbf{D}\mathbf{V}\mathbf{V}^{\mathsf{T}}\|_{\mathsf{F}}^2 + \dot{\rho}\|\mathbf{V}\|_1 + \iota_{\mathcal{M}}(\mathbf{Y}), \, \text{s. t.}\, -\mathbf{Y} + \mathbf{V} = \mathbf{0}$. It corresponds to Problem (1) with $\mathbf{x}_1 = \text{vec}(\mathbf{Y})$, $\mathbf{x}_2 = \text{vec}(\mathbf{V})$, $f_1(\mathbf{x}_1) = 0$, $h_1(\mathbf{x}_1) = \iota_{\mathcal{M}}(\mathbf{Y})$, $f_2(\mathbf{x}_2) = \frac{1}{2\dot{m}} \|\mathbf{D} - \mathbf{D}\mathbf{V}\mathbf{V}^{\mathsf{T}}\|_{\mathsf{F}}^2$, $h_2(\mathbf{x}_2) = \dot{\rho}\|\mathbf{V}\|_1$, $\mathbf{A}_1 = -\mathbf{I}$, $\mathbf{A}_2 = \mathbf{I}$, $\mathbf{b} = \mathbf{0}$, and Condition $\mathbb{BI}$.

▶ **Structured Sparse Phase Retrieval**. Sparse phase retrieval (Duchi & Ruan, 2018) aims to recover a sparse signal from the magnitudes of linear measurements. By incorporating additional linear constraints, recovery accuracy can be further improved. The problem is formulated as: $\min_{\mathbf{v}} \|(\mathbf{G}\mathbf{v}) \odot (\mathbf{G}\mathbf{v}) - \mathbf{z}\|_2^2 + \dot{\rho}\|\mathbf{v}\|_1, \, \text{s. t.}\, \mathbf{D}\mathbf{v} \geq \mathbf{0}$, where $\dot{\rho} \geq 0$, $\mathbf{G} \in \mathbb{R}^{\dot{m} \times \dot{d}}$, $\mathbf{z} \in \mathbb{R}^{\dot{m}}$, $\mathbf{D} \in \mathbb{R}^{\dot{r} \times \dot{d}}$, with $\mathbf{D}$ being surjective that $\mathbf{D}\mathbf{D}^{\mathsf{T}} \succ \mathbf{0}$. Introducing a new variable $\mathbf{y}$, this problem can be formulated as: $\min_{\mathbf{v},\mathbf{y}} \|(\mathbf{G}\mathbf{v}) \odot (\mathbf{G}\mathbf{v}) - \mathbf{z}\|_2^2 + \dot{\rho}\|\mathbf{v}\|_1 + \iota_{\geq \mathbf{0}}(\mathbf{y}), \, s.t.\, \mathbf{y} - \mathbf{D}\mathbf{v} = \mathbf{0}$. This corresponds to Problem (1) with $\mathbf{x}_1 = \mathbf{y}$, $\mathbf{x}_2 = \mathbf{v}$, $f_1(\mathbf{x}_1) = 0$, $h_1(\mathbf{x}_1) = \iota_{\geq \mathbf{0}}(\mathbf{y})$, $f_2(\mathbf{x}_2) = \frac{1}{2} \|(\mathbf{G}\mathbf{v}) \odot (\mathbf{G}\mathbf{v}) - \mathbf{b}\|_2^2$, $h_2(\mathbf{x}_2) = \dot{\rho}\|\mathbf{v}\|_1$, $\mathbf{A}_1 = \mathbf{I}$, $\mathbf{A}_2 = -\mathbf{D}$, $\mathbf{b} = \mathbf{0}$, and Condition $\mathbb{SU}$.

## 3 THE PROPOSED IPDS-ADMM ALGORITHM

This section describes the proposed IPDS-ADMM algorithm for solving Problem (1), featuring with using a new Increasing Penalization and Decreasing Smoothing (IPDS) strategy.

### 3.1 INCREASING PENALIZATION STRATEGY

We employ an increasing penalty update strategy that is crucial to our algorithm. A natural choice for this penalty update rule is to use functions from the $\ell_p$ family. Throughout this paper, we consider the following penalty update rule $\{\beta^t\}_{t=0}^{\infty}$ for any given parameters $\xi, \delta, p \in (0, 1)$:

$$\beta^t = \beta^0(1 + \xi t^p), \; \beta^0 \geq L_n/(\delta\overline{\lambda}). \tag{2}$$

Here, $L_n$ and $\overline{\lambda}$ are defined in Assumption 1.1 and Assumption 1.3, respectively.

We obtain the following useful lemma regarding the penalty update rule.

**Lemma 3.1.** *(Proof in Appendix C.1) Given $\xi, \delta, p \in (0, 1)$, assume Formulation (2) is used to choose $\{\beta^t\}_{t=0}^{\infty}$. We have: (**a**) $\beta^t \leq \beta^{t+1} \leq (1 + \xi)\beta^t$, (**b**) $L_n \leq \delta\beta^t\overline{\lambda}$.*

**Remarks** *(i)* The increasing penalty update strategy is closely coupled with the decreasing smoothing strategy and the diminishing stepsize approach in the literature. These strategies are frequently employed in subgradient methods (Li et al., 2021), smoothing gradient methods (Böhm & Wright, 2021; Sun & Sun, 2023; Lei Yang, 2021), penalty decomposition methods (Lu & Zhang, 2013), and stochastic optimization algorithms like ADAM (Kingma & Ba, 2015; Chen et al., 2022), but are less commonly utilized in ADMM frameworks. We examine this approach within ADMM but limit our discussion to specific form and condition as in Formulation (2). *(ii)* The condition $\beta^0 \geq L_n/(\delta\overline{\lambda})$ in Formulation (2) essentially mandates that the initial penalty value be sufficiently large. This condition can be automatically satisfied since an increasing penalty update is used. *(iii)* The result $\beta^{t+1} \leq (1 + \xi)\beta^t$ in Lemma 3.1 implies that the penalty parameter grows, but not excessively fast, with a constant $\xi$ to prevent rapid escalation.

### 3.2 DECREASING SMOOTHING STRATEGY

IPDS-ADMM is built upon the Moreau envelope smoothing technique (Li et al., 2022; Zeng et al., 2022; Sun & Sun, 2023; Böhm & Wright, 2021). Our algorithm gradually decreases the smoothing parameter at a controlled rate.

Initially, we provide the following useful definition.

**Definition 3.2.** *The Moreau envelope of a proper convex and Lipschitz continuous function $h(\mathbf{u})$ : $\mathbb{R}^{d \times 1} \mapsto \mathbb{R}$ with parameter $\mu \in (0, \infty)$ is defined as $h(\mathbf{u}; \mu) \triangleq \min_{\mathbf{v} \in \mathbb{R}^{d \times 1}} h(\mathbf{v}) + \frac{1}{2\mu} \|\mathbf{v} - \mathbf{u}\|_2^2$.*

We present several useful properties of Moreau envelope functions in the following lemmas.

**Lemma 3.3.** *(Beck, 2017, Chapter 6) Suppose the function $h(\mathbf{u})$ is $C_h$-Lipschitz continuous and convex w.r.t. $\mathbf{u}$. We have: (a) The function $h(\mathbf{u}; \mu)$ is $C_h$-Lipschitz continuous w.r.t. $\mathbf{u}$. (b) The function $h(\mathbf{u}; \mu)$ is $(1/\mu)$-smooth w.r.t. $\mathbf{u}$, and its gradient can be computed as: $\nabla h(\mathbf{u}; \mu) = \frac{1}{\mu}(\mathbf{u} - \text{Prox}_h(\mathbf{u}; \mu))$, where $\text{Prox}_h(\mathbf{u}; \mu) = \arg\min_{\mathbf{v}} h(\mathbf{v}) + \frac{1}{2\mu} \|\mathbf{v} - \mathbf{u}\|_2^2$. (c) $0 \le h(\mathbf{u}) - h(\mathbf{u}; \mu) \le \frac{1}{2}\mu C_h^2$.*

**Lemma 3.4.** *(Proof in Appendix C.2) Assuming $0 < \mu_2 < \mu_1$ and fixing $\mathbf{u} \in \mathbb{R}^{d \times 1}$, we have: $0 \le \frac{h(\mathbf{u}; \mu_2) - h(\mathbf{u}; \mu_1)}{\mu_1 - \mu_2} \le \frac{1}{2}C_h^2$.*

**Lemma 3.5.** *(Proof in Appendix C.3) Assuming $0 < \mu_2 < \mu_1$ and fixing $\mathbf{u} \in \mathbb{R}^{d \times 1}$, we have: $\|\nabla h(\mathbf{u}; \mu_1) - \nabla h(\mathbf{u}; \mu_2)\| \le (\frac{\mu_1}{\mu_2} - 1) \cdot C_h$.*

**Lemma 3.6.** *(Proof in Appendix C.4) Given constants $\{\mathbf{c}, \mu, \rho\}$, we consider the convex problem: $\bar{\mathbf{x}}_n = \arg\min_{\mathbf{x}_n} h_n(\mathbf{x}_n; \mu) + \frac{\rho}{2}\|\mathbf{x}_n - \mathbf{c}\|_2^2$. We have: (a) $\bar{\mathbf{x}}_n = \frac{\mu}{1+\mu\rho}(\frac{1}{\mu}\breve{\mathbf{x}}_n + \rho\mathbf{c})$, where $\breve{\mathbf{x}}_n = \arg\min_{\breve{\mathbf{x}}_n} h_n(\breve{\mathbf{x}}_n) + \frac{1}{2} \cdot \frac{\rho}{1+\mu\rho}\|\breve{\mathbf{x}}_n - \mathbf{c}\|_F^2 = \text{Prox}_n(\mathbf{c}; \mu + 1/\rho)$. (b) $\rho(\mathbf{c} - \bar{\mathbf{x}}_n) \in \partial h(\breve{\mathbf{x}}_n)$. (c) $\|\mathbf{x}_n - \breve{\mathbf{x}}_n\| \le \mu C_h$.*

**Remark 3.7.** *(i) Lemmas 3.4 and 3.5 are derived using standard convex analysis techniques and play a key role in analyzing the proposed IPDS-ADMM algorithm. (ii) Lemma 3.6 is essential for establishing the iteration complexity of Algorithm 1 in reaching a critical point. The results of Lemma 3.6 are analogous to those of Lemma 1 in (Li et al., 2022).*

### 3.3 THE MAIN ALGORITHM

This subsection provides the proposed IPDS-ADMM algorithm. Initially, we consider the following alternative optimization problem:

$$\min_{\mathbf{x}_1, \mathbf{x}_2, \dots, \mathbf{x}_n} h_n(\mathbf{x}_i; \mu) + [\textstyle\sum_{i=1}^{n-1} h_i(\mathbf{x}_i)] + [\textstyle\sum_{i=1}^{n} f_i(\mathbf{x}_i)], \, s.t. \, [\textstyle\sum_{i=1}^{n} \mathbf{A}_i \mathbf{x}_i] = \mathbf{b}, \tag{3}$$

where $\mu \to 0$, and $h_n(\mathbf{x}_n; \mu) \triangleq \min_{\mathbf{v} \in \mathbb{R}^{d_n \times 1}} h(\mathbf{v}) + \frac{1}{2\mu}\|\mathbf{v} - \mathbf{x}_n\|_2^2$ is the Moreau envelope of $h_n(\mathbf{x}_n)$ with parameter $\mu$. Lemma 3.3 confirms that $h_n(\mathbf{x}_n, \mu)$ is a $(1/\mu)$-smooth function assuming $h_n(\cdot)$ is convex. We present the augmented Lagrangian function for Problem (3), as follows:

$$\mathcal{L}(\mathbf{x}, \mathbf{z}; \beta, \mu) \triangleq h_n(\mathbf{x}_n; \mu) + \{\textstyle\sum_{i=1}^{n-1} h_i(\mathbf{x}_i)\} + G(\mathbf{x}, \mathbf{z}; \beta), \tag{4}$$

where $G(\mathbf{x}, \mathbf{z}; \beta)$ is differentiable and defined as:

$$G(\mathbf{x}, \mathbf{z}; \beta) \triangleq \textstyle\sum_{i=1}^{n} f_i(\mathbf{x}_i) + \langle [\textstyle\sum_{i=1}^{n} \mathbf{A}_i \mathbf{x}_i] - \mathbf{b}, \mathbf{z} \rangle + \frac{\beta}{2}\|[\textstyle\sum_{i=1}^{n} \mathbf{A}_i \mathbf{x}_i] - \mathbf{b}\|_2^2.$$

Here, $\mu \in (0, \infty)$, $\beta \in (0, \infty)$, and $\mathbf{z} \in \mathbb{R}^{m \times 1}$ are the smoothing parameter, the penalty parameter, and the dual variable, respectively. We employ an increasing penalty and decreasing smoothing update scheme throughout all iterations $t = \{0, 1, \dots, \infty\}$ with $\beta^t \to +\infty$ and $\mu^t \propto \frac{1}{\beta^t} \to 0$. Notably, the function $G(\mathbf{x}^t, \mathbf{z}^t; \beta^t)$ is $\mathsf{L}_i^t$-smooth w.r.t. $\mathbf{x}_i$ for all $i \in [m]$, where $\mathsf{L}_i^t = L_i + \beta^t\|\mathbf{A}_i\|_2^2$. For notation simplicity, for all $i \in [n]$, we denote $\ddot{\mathbf{g}}_i^t \triangleq \nabla_{\mathbf{x}_i} G(\mathbf{x}_{[1, i-1]}^{t+1}, \mathbf{x}_i^t, \mathbf{x}_{[i+1, n]}^t, \mathbf{z}^t; \beta^t)$ as the gradient of $G(\mathbf{x}, \mathbf{z}^t; \beta^t)$ w.r.t. $\mathbf{x}_i$ at the point $\mathbf{x}_i^t$.

In each iteration, we select suitable parameters $\{\beta^t, \mu^t\}$ and sequentially update the variables $(\mathbf{x}_1, \mathbf{x}_2, \dots, \mathbf{x}_n, \mathbf{z})$. We employ the proximal linearized method to cyclically update the variables $\{\mathbf{x}_1, \mathbf{x}_2, \dots, \mathbf{x}_n\}$. Specifically, we update each variable $\mathbf{x}_i$ by solving the following sub-problem for all $i \in [n]$: $\mathbf{x}_i^{t+1} \approx \arg\min_{\mathbf{x}_i \in \mathbb{R}^{d_i \times 1}} \mathcal{L}(\mathbf{x}_{[1, i-1]}^t, \mathbf{x}_i, \mathbf{x}_{[i+1, n]}^t, \mathbf{z}^t; \beta^t, \mu^t)$. To address

the $\mathbf{x}_i$-subproblem, we employ a proximal linearized minimization strategy for all $i \in [n-1]$: $\mathbf{x}_i^{t+1} \in \arg\min_{\mathbf{x}_i} h_i(\mathbf{x}_i) + \frac{\theta_1 \mathsf{L}_i^t}{2}\|\mathbf{x}_i - \mathbf{x}_i^t\|_2^2 + \langle \mathbf{x}_i - \mathbf{x}_i^t, \ddot{\mathbf{g}}_i^t; \mathbf{z}^t; \beta^t \rangle$. However, for the final block of the problem, we consider a subtly different proximal linearized minimization strategy: $\mathbf{x}_n^{t+1} = \arg\min_{\mathbf{x}_n} h_n(\mathbf{x}_n; \mu^t) + \frac{\theta_2 \mathsf{L}_n^t}{2}\|\mathbf{x}_n - \mathbf{x}_n^t\|_2^2 + \langle \mathbf{x}_n - \mathbf{x}_n^t, \ddot{\mathbf{g}}_n^t \rangle$. Importantly, we assign $\theta_1$ to blocks $[1, n-1]$ and $\theta_2$ to block $n$. Our algorithm updates the dual variable $\mathbf{z}^t$ using either an under-relaxation stepsize $\sigma \in (0,1)$ or an over-relaxation stepsize $\sigma \in [1,2)$.

---

**Algorithm 1:** IPDS-ADMM: The Proposed Proximal Linearized ADMM for Problem (1).

---

Choose suitable parameters $\{p, \xi, \delta\}$ and $\{\sigma, \theta_1, \theta_2\}$ using Formula (5) or Formula (6).

Initialize $\{\mathbf{x}^0, \mathbf{z}^0\}$. Choose $\beta^0 \geq L_n/(\delta\bar{\lambda})$.

**for** $t$ from 0 to $T$ **do**

> **S1)** IPDS Strategy: Set $\beta^t = \beta^0(1 + \xi t^p)$, $\mu^t = 1/(\bar{\lambda}\delta\beta^t)$.
> We define $\ddot{\mathbf{g}}_i^t \triangleq \nabla_{\mathbf{x}_i} G(\mathbf{x}_{[1,i-1]}^{t+1}, \mathbf{x}_i^t, \mathbf{x}_{[i+1,n]}^t, \mathbf{z}^t; \beta^t)$.
>
> **S2)** $\mathbf{x}_1^{t+1} \in \arg\min_{\mathbf{x}_1} h_1(\mathbf{x}_1) + \langle \mathbf{x}_1 - \mathbf{x}_1^t, \ddot{\mathbf{g}}_1^t \rangle + \frac{\theta_1 \mathsf{L}_1^t}{2}\|\mathbf{x}_1 - \mathbf{x}_1^t\|_2^2$
>
> **S3)** $\mathbf{x}_2^{t+1} \in \arg\min_{\mathbf{x}_2} h_2(\mathbf{x}_2) + \langle \mathbf{x}_2 - \mathbf{x}_2^t, \ddot{\mathbf{g}}_2^t \rangle + \frac{\theta_1 \mathsf{L}_2^t}{2}\|\mathbf{x}_2 - \mathbf{x}_2^t\|_2^2$
>
> $\cdots$
>
> **S4)** $\mathbf{x}_{n-1}^{t+1} \in \arg\min_{\mathbf{x}_{n-1}} h_{n-1}(\mathbf{x}_{n-1}) + \langle \mathbf{x}_{n-1} - \mathbf{x}_{n-1}^t, \ddot{\mathbf{g}}_{n-1}^t \rangle + \frac{\theta_1 \mathsf{L}_{n-1}^t}{2}\|\mathbf{x}_{n-1} - \mathbf{x}_{n-1}^t\|_2^2$
>
> **S5)** $\mathbf{x}_n^{t+1} \in \arg\min_{\mathbf{x}_n} h_n(\mathbf{x}_n; \mu) + \langle \mathbf{x}_n - \mathbf{x}_n^t, \ddot{\mathbf{g}}_n^t \rangle + \frac{\theta_2 \mathsf{L}_n^t}{2}\|\mathbf{x}_n - \mathbf{x}_n^t\|_2^2$. It can be solved
> using Lemma 3.6 as $\mathbf{x}_n^{t+1} = \frac{1}{1+\mu\rho}(\breve{\mathbf{x}}_n^{t+1} + \mu\rho\mathbf{c})$, where $\breve{\mathbf{x}}_n^{t+1} = \mathrm{Prox}_n(\mathbf{c}; \mu + 1/\rho)$,
> $\mu = \mu^t$, $\rho \triangleq \theta_2 \mathsf{L}_n^t$, and $\mathbf{c} \triangleq \mathbf{x}_n^t - \ddot{\mathbf{g}}_n^t/\rho$.
>
> **S6)** $\mathbf{z}^{t+1} = \mathbf{z}^t + \sigma\beta^t([\sum_{j=1}^n \mathbf{A}_j\mathbf{x}_j^{t+1}] - \mathbf{b})$

**end**

---

We present IPDS-ADMM in Algorithm 1, and have the following remarks.

**Remark 3.8.** *(i) Algorithm 1 can be viewed as a generalized cyclic coordinate descent method applied to the augmented Lagrangian function in Equation (4). (ii) The Moreau envelope smoothing technique has been used in the design of augmented Lagrangian methods (Zeng et al., 2022), ADMMs (Li et al., 2022; Yuan, 2025), and minimax optimization (Zhang et al., 2020). However, these algorithms typically utilize constant penalties, whereas we adopt an Increasing Penalization and Decreasing Smoothing (IPDS) strategy to improve the iteration complexity of RADMM (Li et al., 2022), reducing it from $\mathcal{O}(\epsilon^{-4})$ to $\mathcal{O}(\epsilon^{-3})$. (iii) Algorithm 1 is a fully splitting algorithm, where each step reduces to computing a proximal operator. For the first $(n-1)$ blocks, we have: $\mathbf{x}_i^{t+1} \in \mathrm{Prox}_i(\mathbf{x}_i^t - \ddot{\mathbf{g}}_i^t/\dot{\rho}; 1/\dot{\rho})$, where $\dot{\rho} = \theta_1 \mathsf{L}_i^t$. For the last block, Lemma 3.6 can be applied to compute the proximal operator of the smoothed function $h_n(\mathbf{x}_n; \mu)$ using the proximal operator of the original function $h_n(\mathbf{x}_n)$. (iv) The point $\breve{\mathbf{x}}_n^{t+1}$ in Step S5) of Algorihtm 1 plays a crucial role. As will be seen later in Theorem 4.15, the point $(\mathbf{x}_1^t, \mathbf{x}_2^t, \ldots, \mathbf{x}_{n-1}^t, \breve{\mathbf{x}}_n^t, \mathbf{z}^t)$, rather than the point $(\mathbf{x}_1^t, \mathbf{x}_2^t, \ldots, \mathbf{x}_{n-1}^t, \mathbf{x}_n^t, \mathbf{z}^t)$, will serve as an approximate critical point of Problem (1) in our complexity results. (v) RADMM (Li et al., 2022) uses a fixed large penalty parameter $\mathcal{O}(1/\epsilon)$ and a fixed small smoothing parameter $\mathcal{O}(\epsilon)$ to achieve an $\epsilon$-approximate critical point. However, this leads to overly conservative step sizes for the primal and dual updates, potentially hindering the algorithm's practical performance. (vi) We apply the smoothing strategy only to the last block to bound the dual variables via the primal ones. This leverages the Lipschitz continuity of the smoothed function to estimate $\frac{1}{\beta^t}\|\mathbf{z}^{t+1} - \mathbf{z}^t\|_2^2$ and construct a suitable potential function. (vii) Some may be concerned that using an increasing penalty could cause the parameter to grow excessively fast. However, by setting $\xi \ll 1$, we ensure that $\beta^{t+1} \leq (1+\xi)\beta^t$, meaning the penalty grows very slowly in practice.*

## 3.4 Choosing Suitable Parameters $\{p, \xi, \delta\}$ and $\{\sigma, \theta_1, \theta_2\}$

Selecting appropriate parameters $\{p, \xi, \delta\}$ and $\{\sigma, \theta_1, \theta_2\}$ is essential to ensuring the global convergence of Algorithm 1. In our theoretical analysis and empirical experiments, we suggest the

following choices for $\{p, \xi, \delta\}$ and $\{\sigma, \theta_1, \theta_2\}$:

$$\mathbb{BI} : p = \tfrac{1}{3}, \; \xi \in (0, \infty), \; \delta \in (0, \tfrac{1}{3}(\tfrac{2}{\kappa} - 1)), \sigma \in [1, 2), \theta_1 = 1.01, \theta_2 = \tfrac{1/\kappa - \delta}{1 + \delta} + \tfrac{1}{2\varrho(1 + \delta)^2}. \quad (5)$$

$$\mathbb{SU} : p = \tfrac{1}{3}, \; \xi = \delta = \sigma = \tfrac{0.01}{\kappa}, \; \theta_1 = 1.01, \theta_2 = 1.5. \quad (6)$$

Here, $\varrho \triangleq 6\omega\sigma_1\kappa$, $\sigma_1 \triangleq \frac{\sigma}{(1 - |1 - \sigma|)^2}$, and $\omega \triangleq 1 + \frac{\xi}{2\sigma} + \sigma\xi$. Notably, $\theta_2$ in (5) depends on $(\xi, \delta, \sigma)$.

**Remark 3.9.** *(i) We obverse from (6) that the parameters $\{\xi, \delta, \sigma\}$ is inversely proportional to the condition number $\kappa$. Such settings are partly consistent with those in (Boţ et al., 2019) (refer to Lemma 5 in (Boţ et al., 2019)). (ii) Introducing the relaxation parameter $\sigma \in (0, 2)$ enables handling cases where the matrix is surjective. Specifically, when the matrix is bijective, we can use an over-relaxation step size for faster convergence, whereas for surjective matrices, the algorithm requires conservative step sizes to ensure global convergence.*

## 4 GLOBAL CONVERGENCE

This section establishes the global convergence of Algorithm 1.

We begin with a high-level overview of the proof strategy. First, using the Lagrangian function, we derive sufficient decrease conditions for the four parameter sets: primal variables, dual variables, the penalty parameter, and the smoothing parameter. Next, using the first-order optimality conditions and dual update rules, we bound the difference in dual variables using primal by the difference in primal variables. Lastly, we show that the tail error term related to the smoothing parameter is constant, establishing the summability of the sequence linked to a potential function.

We provide the following three useful lemmas.

**Lemma 4.1.** *(Proof in Appendix D.1, A Sufficient Decrease Property) Fix $\varepsilon_3 \triangleq \xi$ and $\varepsilon_1 \triangleq \tfrac{1}{2}\theta_1 - \tfrac{1}{2}$. Let $\varepsilon_2 \in \mathbb{R}$. For all $t \geq 1$, we have:*

$$\mathcal{E}^{t+1} + \Theta_L^{t+1} - \Theta_L^t \leq (\tfrac{1}{2} - \theta_2 + \varepsilon_2) \cdot \mathsf{L}_n^t \|\mathbf{x}_n^{t+1} - \mathbf{x}_n^t\|_2^2 + \tfrac{\omega}{\sigma\beta^t} \|\mathbf{z}^{t+1} - \mathbf{z}^t\|_2^2, \quad (7)$$

*where*

$$\mathcal{E}^{t+1} \triangleq [\varepsilon_1 \textstyle\sum_{i=1}^{n-1} \mathsf{L}_i^t \|\mathbf{x}_i^{t+1} - \mathbf{x}_i^t\|_2^2] + \varepsilon_2 \mathsf{L}_n^t \|\mathbf{x}_n^{t+1} - \mathbf{x}_n^t\|_2^2 + \tfrac{\varepsilon_3}{\beta^t} \|\mathbf{z}^{t+1} - \mathbf{z}^t\|_2^2.$$

$$\Theta_L^t \triangleq \mathcal{L}(\mathbf{x}^t, \mathbf{z}^t; \beta^t, \mu^t) + \tfrac{1}{2} C_h \mu^t, \; \mathsf{L}_i^t = L_i + \beta^t \|\mathbf{A}_i\|_2^2, \; \omega \triangleq 1 + \tfrac{\xi}{2\sigma} + \sigma\xi.$$

**Lemma 4.2.** *(Proof in Appendix D.2, First-Order Optimality Condition) Assume $\sigma \in (0, 2)$. For all $t \geq 1$ and $i \in [n - 1]$, we have the following results.*

*(a) Let $\mathrm{w}_i^{t+1} \in \partial h_i(\mathbf{x}_i^{t+1}) + \nabla f_i(\mathbf{x}_i^t)$, and $\mathrm{u}_i^{t+1} \triangleq \theta_1 \mathsf{L}_i^t(\mathbf{x}_i^{t+1} - \mathbf{x}_i^t) - \beta^t \mathbf{A}_i^\mathsf{T}[\sum_{j=i}^n \mathbf{A}_j(\mathbf{x}_j^{t+1} - \mathbf{x}_j^t)]$. It holds that: $\mathbf{0} = \sigma\mathbf{A}_i^\mathsf{T}\mathbf{z}^t + \mathbf{A}_i^\mathsf{T}(\mathbf{z}^{t+1} - \mathbf{z}^t) + \sigma\mathrm{w}_i^{t+1} + \sigma\mathrm{u}_i^{t+1}$.*

*(b) Let $\mathrm{w}_n^{t+1} \triangleq \nabla h_n(\mathbf{x}_n^t, \mu^t) + \nabla f_n(\mathbf{x}_n^t)$, and $\mathrm{u}_n^{t+1} \triangleq \mathbf{Q}^t(\mathbf{x}_n^{t+1} - \mathbf{x}_n^t)$, where $\mathbf{Q}^t \triangleq \theta_2 \mathsf{L}_n^t \mathbf{I} - \beta^t \mathbf{A}_n^\mathsf{T}\mathbf{A}_n$. It holds that: $\mathbf{0} = \sigma\mathbf{A}_n^\mathsf{T}\mathbf{z}^t + \mathbf{A}_n^\mathsf{T}(\mathbf{z}^{t+1} - \mathbf{z}^t) + \sigma\mathrm{w}_n^{t+1} + \sigma\mathrm{u}_n^{t+1}$.*

*(c) We have the following two different identities:*

$$\mathbb{BI} : \underbrace{\mathbf{A}_n^\mathsf{T}(\mathbf{z}^{t+1} - \mathbf{z}^t)}_{\triangleq \mathrm{a}^{t+1}} = (1 - \sigma)\underbrace{(\mathbf{A}_n^\mathsf{T}(\mathbf{z}^t - \mathbf{z}^{t-1}))}_{\triangleq \mathrm{a}^t} + \sigma\underbrace{(\mathrm{u}_n^t - \mathrm{u}_n^{t+1} + \mathrm{w}_n^t - \mathrm{w}_n^{t+1})}_{\mathrm{c}^t}. \quad (8)$$

$$\mathbb{SU} : \underbrace{\mathbf{A}_n^\mathsf{T}(\mathbf{z}^{t+1} - \mathbf{z}^t) + \sigma\mathrm{u}_n^{t+1}}_{\triangleq \mathrm{a}^{t+1}} = (1 - \sigma)\underbrace{(\mathbf{A}_n^\mathsf{T}(\mathbf{z}^t - \mathbf{z}^{t-1}) + \sigma\mathrm{u}_n^t)}_{\triangleq \mathrm{a}^t} + \sigma\underbrace{(\sigma\mathrm{u}_n^t + \mathrm{w}_n^t - \mathrm{w}_n^{t+1})}_{\triangleq \mathrm{c}^t}. \quad (9)$$

**Lemma 4.3.** *(Proof in Appendix D.3) For all $t \geq 0$, we have: (a) $\mathsf{L}_n^t \leq \beta^t\overline{\lambda}(1 + \delta)$; (b) $\|\mathbf{Q}^t\| \leq \beta^t\overline{\lambda}q$, where $q \triangleq \theta_2(1 + \delta) - \underline{\lambda}'/\overline{\lambda}$; (c) $\|\mathrm{u}_n^{t+1}\| \leq q\overline{\lambda}\beta^t\|\mathbf{x}_n^{t+1} - \mathbf{x}_n^t\|$.*

We provide convergence analysis of Algorithm 1 under two conditions: Condition $\mathbb{BI}$ using Formulation (8), and Condition $\mathbb{SU}$ using Formulation (9). We define $\Theta_L^t \triangleq \mathcal{L}(\mathbf{x}^t, \mathbf{z}^t; \beta^t, \mu^t) + \tfrac{1}{2}C_h\mu^t$, and $\omega \triangleq 1 + \frac{\xi}{2\sigma} + \sigma\xi$. We define $\sigma_1 \triangleq \frac{\sigma}{(1 - |1 - \sigma|)^2}$, and $\sigma_2 \triangleq \frac{|1 - \sigma|}{\sigma(1 - |1 - \sigma|)}$, where $\sigma \in (0, 2)$. We construct a sequence associated with the potential (or Lyapunov) function for different Conditions

$\mathbb{BI}$ and $\mathbb{SU}$ as follows:

$$\mathbb{BI} : \Theta^t = \Theta_L^t + \underbrace{\frac{\omega\sigma_2}{\lambda}}_{\triangleq a} \cdot \underbrace{\frac{1}{\beta^t}\|\mathbb{a}^t\|_2^2}_{\triangleq \mathbb{A}^t} + \underbrace{\frac{3\omega\sigma_1}{\lambda}}_{\triangleq b} \cdot \underbrace{\frac{1}{\beta^t}(L_n\|\mathbf{x}_n^t - \mathbf{x}_n^{t-1}\| + \|\mathbb{u}_n^t\|)^2}_{\triangleq \mathbb{B}^t}. \tag{10}$$

$$\mathbb{SU} : \Theta^t = \Theta_L^t + \underbrace{\frac{2\omega\sigma_2}{\lambda}}_{\triangleq a} \cdot \underbrace{\frac{1}{\beta^t}\|\mathbb{a}^t\|_2^2}_{\triangleq \mathbb{A}^t} + \underbrace{\frac{6\omega\sigma_1}{\lambda}}_{\triangleq b} \cdot \underbrace{\frac{1}{\beta^t}(L_n\|\mathbf{x}_n^t - \mathbf{x}_n^{t-1}\| + \sigma\|\mathbb{u}_n^t\|)^2}_{\triangleq \mathbb{B}^t}. \tag{11}$$

## 4.1 Analysis for Condition $\mathbb{BI}$

We provide a convergence analysis of Algorithm 1 under Condition $\mathbb{BI}$, where $\mathbf{A}_n$ is a bijective matrix. We assume an over-relaxation stepsize is used with $\sigma \in [1, 2)$.

The subsequent lemma uses Equation (8) to establish an upper bound for the term $\frac{\omega}{\sigma\beta^t}\|\mathbf{z}^{t+1} - \mathbf{z}^t\|_2^2$.

**Lemma 4.4.** *(Proof in Appendix D.4, Bounding Dual Using Primal) We define $\omega$ as in Lemma 4.1. For all $t \geq 1$, we have:*

$$\frac{\omega}{\sigma\beta^t}\|\mathbf{z}^{t+1} - \mathbf{z}^t\|_2^2 \leq \Theta_+^t - \Theta_+^{t+1} + \chi\mathsf{L}_n^t\|\mathbf{x}_n^{t+1} - \mathbf{x}_n^t\|_2^2 + \mathbb{U}^t, \tag{12}$$

*where $\chi \triangleq \varrho(\delta + \theta_2 + \theta_2\delta - 1/\kappa)^2$, $\varrho \triangleq 6\omega\sigma_1\kappa$, $\Theta_+^t \triangleq a\mathbb{A}^t + b\mathbb{B}^t$, and $\mathbb{U}^t \triangleq C_h^2\frac{b}{\beta^t} \cdot (\frac{\mu^{t-1}}{\mu^t} - 1)^2$. Here, $\{a, \mathbb{A}^t, b, \mathbb{B}^t\}$ are defined in Equation (10).*

Assume Equation (5) is used to choose $\{p, \xi, \delta, \sigma, \theta_1, \theta_2\}$. We have the following lemma.

**Lemma 4.5.** *(Proof in Appendix D.5) We define $\{\chi, \varrho\}$ in Lemma 4.4. We have the following results:*

**(a)** *It holds that $\varepsilon_1 \triangleq \frac{1}{2}\theta_1 - \frac{1}{2} > 0$, and $\varepsilon_2 \triangleq \theta_2 - \frac{1}{2} - \chi > 0$.*

**(b)** *For all $t \geq 1$, we have $\mathcal{E}^{t+1} \leq \Theta^t - \Theta^{t+1} + \mathbb{U}^t$.*

## 4.2 Analysis for Condition $\mathbb{SU}$

We provide a convergence analysis of Algorithm 1 under Condition $\mathbb{SU}$, where $\mathbf{A}_n$ is a surjective matrix. We assume an under-relaxation stepsize is used with $\sigma \in (0, 1)$.

The following lemma utilizes Equation (9) to establish an upper bound for the term $\frac{\omega}{\sigma\beta^t}\|\mathbf{z}^{t+1} - \mathbf{z}^t\|_2^2$.

**Lemma 4.6.** *(Proof in Appendix D.6, Bounding Dual Using Primal) We define $\omega$ as in Lemma 4.1. For all $t \geq 1$, we have:*

$$\frac{\omega}{\sigma\beta^t}\|\mathbf{z}^{t+1} - \mathbf{z}^t\|_2^2 \leq \Theta_+^t - \Theta_+^{t+1} + \chi \cdot \mathsf{L}_n^t\|\mathbf{x}_n^{t+1} - \mathbf{x}_n^t\|_2^2 + \mathbb{U}^t, \tag{13}$$

*where $\chi \triangleq \frac{2\omega\kappa}{\sigma} \cdot \{\sigma^2 q^2 + 3\delta^2 + 3(\delta + \sigma q)^2\}$, $q \triangleq \theta_2 + \theta_2\delta$, $\Theta_+^t \triangleq a\mathbb{A}^t + b\mathbb{B}^t$, and $\mathbb{U}^t \triangleq C_h^2\frac{b}{\beta^t} \cdot (\frac{\mu^{t-1}}{\mu^t} - 1)^2$. Here, $\{a, \mathbb{A}^t, b, \mathbb{B}^t\}$ are defined in Equation (11).*

Assume Equation (6) is used to choose $\{p, \xi, \delta, \sigma, \theta_1, \theta_2\}$. We have the following lemma.

**Lemma 4.7.** *(Proof in Appendix D.7) We define $\chi$ in Lemma 4.6. We have the following results:*

**(a)** *It holds that $\varepsilon_1 \triangleq \frac{1}{2}\theta_1 - \frac{1}{2} > 0$, and $\varepsilon_2 \triangleq \theta_2 - \frac{1}{2} - \chi > 0$.*

**(b)** *For all $t \geq 1$, we have: $\mathcal{E}^{t+1} \leq \Theta^t - \Theta^{t+1} + \mathbb{U}^t$.*

## 4.3 Continuing Analysis for Conditions $\mathbb{BI}$ and $\mathbb{SU}$

Using Assumption 1.4, we show that $\Theta^t$ is consistently lower-bounded by the following lemma.

**Lemma 4.8.** *(Proof in Appendix D.8) For all $t \geq 1$, there exists a constant $\underline{\Theta}$ such that $\Theta^t \geq \underline{\Theta}$.*

The following lemma shows that both $\left(\sum_{t=1}^{\infty} \mathbb{U}^t\right)$ and $\left(\sum_{t=1}^{\infty} \mathcal{E}^{t+1}\right)$ are always upper-bounded.

**Lemma 4.9.** *(Proof in Appendix D.9) We define $\mathbb{U}^t$ as in Lemma 4.4 or Lemma 4.6. We define $\overline{\mathcal{E}}$ as in Lemma 4.1. We have:*

**(a)** *There exists a universal positive constant $\overline{\mathbb{U}}$ such that $\sum_{t=1}^{\infty} \mathbb{U}^t \leq \overline{\mathbb{U}}$.*

**(b)** Letting $\overline{\mathcal{E}} \triangleq \Theta^1 - \underline{\Theta} + \overline{U}$, we have: $\sum_{t=1}^{\infty} \mathcal{E}^{t+1} \leq \overline{\mathcal{E}}$.

The following lemmas are useful to provide upper bounds for the dual and primal variables.

**Lemma 4.10.** *(Proof in Appendix D.10) For all $t \geq 1$, there exist constants $\{Z, \ddot{Z}\}$ such that $\frac{1}{\beta^t}\|\mathbf{z}^t\|_2^2 \leq Z$, and $\sum_{t=1}^{\infty} \frac{1}{\beta^t}\|\mathbf{z}^{t+1} - \mathbf{z}^t\|_2^2 \leq \ddot{Z}$.*

**Lemma 4.11.** *(Proof in Appendix D.11) For all $i \in [n]$, we have $\|\mathbf{x}_i^{t+1}\| < +\infty$.*

Finally, we have the following theorem regrading to the global convergence of IPDS-ADMM.

**Theorem 4.12.** *(Proof in Appendix D.12) We have the following results.*

**(a)** $\sum_{t=1}^{T} \|\mathbf{z}^{t+1} - \mathbf{z}^t\|_2^2 + \|\beta^t(\mathbf{x}^{t+1} - \mathbf{x}^t)\|_2^2 \leq K\beta^T$, *where $K > 0$ is some constant.*

**(b)** *There exists an index $\bar{t}$ with $\bar{t} \leq T$ such that $\|\mathbf{z}^{\bar{t}+1} - \mathbf{z}^{\bar{t}}\|_2^2 + \|\beta^{\bar{t}}(\mathbf{x}^{\bar{t}+1} - \mathbf{x}^{\bar{t}})\|_2^2 \leq \frac{K\beta^T}{T}$.*

**Remark 4.13.** *(i) With the choice $\beta^T = \mathcal{O}(T^p)$ with $p \in (0, 1)$, we observe $\ddot{e}^{\bar{t}} \triangleq \|\mathbf{z}^{\bar{t}+1} - \mathbf{z}^{\bar{t}}\|_2^2 + \|\beta^{\bar{t}}(\mathbf{x}^{\bar{t}+1} - \mathbf{x}^{\bar{t}})\|_2^2 = \mathcal{O}(T^{p-1})$, indicating convergence of $\ddot{e}^{\bar{t}}$ towards 0. (ii) In light of Theorem 4.12, a reasonable stopping criterion for Algorithm 1 is $\|\mathbf{z}^{\bar{t}+1} - \mathbf{z}^{\bar{t}}\| + \|\beta^{\bar{t}}(\mathbf{x}^{\bar{t}+1} - \mathbf{x}^{\bar{t}})\| \leq \epsilon$, where $\epsilon \in (0, 1)$ is a user-defined parameter.*

### 4.4 ITERATION COMPLEXITY

We now establish the iteration complexity of Algorithm 1. We first restate the following standard definition of approximated critical points.

**Definition 4.14.** *($\epsilon$-Critical Point) We define $\mathrm{Crit}(\check{\mathbf{x}}, \check{\mathbf{z}}) \triangleq \|\mathbf{A}\check{\mathbf{x}} - \mathbf{b}\| + \sum_{i=1}^{n} \mathrm{dist}(\mathbf{0}, \nabla f_i(\check{\mathbf{x}}_i) + \partial h_i(\check{\mathbf{x}}_i) + \mathbf{A}_i^\top \check{\mathbf{z}})$. A solution $(\check{\mathbf{x}}, \check{\mathbf{z}})$ is an $\epsilon$-critical point if it holds that:*

$$\mathrm{Crit}(\check{\mathbf{x}}, \check{\mathbf{z}}) \leq \epsilon.$$

We obtain the following iteration complexity results.

**Theorem 4.15.** *(Proof in Appendix D.13) We define $\mathbf{q}^t \triangleq \{\mathbf{x}_1^t, \mathbf{x}_2^t, \ldots, \mathbf{x}_{n-1}^t, \check{\mathbf{x}}_n^t\}$. Let the sequence $\{\mathbf{q}^t, \mathbf{z}^t\}_{t=0}^{T}$ be generated by Algorithm 1. For all $p \in (0, 1)$, we have:*

$$\frac{1}{T} \sum_{t=1}^{T} \mathrm{Crit}(\mathbf{q}^{t+1}, \mathbf{z}^{t+1}) \leq \mathcal{O}(T^{(p-1)/2}) + \mathcal{O}(T^{-p}). \tag{14}$$

*In particular, with the choice $p = 1/3$, we have $\frac{1}{T} \sum_{t=1}^{T} \mathrm{Crit}(\mathbf{q}^{t+1}, \mathbf{z}^{t+1}) \leq \mathcal{O}(T^{-1/3})$. In other words, there exists $\bar{t} \leq T$ such that: $\mathrm{Crit}(\mathbf{q}^{\bar{t}+1}, \mathbf{z}^{\bar{t}+1}) \leq \epsilon$, provided that $T \geq \mathcal{O}(\epsilon^{-3})$.*

**Remark 4.16.** *(i) Minimizing the worst-case complexity of the right-hand side of Inequality (14) w.r.t. $p$ yields: $\arg\min_{p \in (0,1)} \max(-p, (p-1)/2) = 1/3$. Thus, choosing $p = 1/3$ achieves the optimal trade-off between the two terms, resulting in the best complexity bounds. (ii) To the best of our knowledge, this represents the first complexity result for using ADMM to solve this class of nonsmooth and nonconvex problems. Remarkably, we observe that it aligns with the iteration bound found in smoothing proximal gradient methods (Böhm & Wright, 2021).*

### 4.5 ON THE BOUNDEDNESS AND CONVERGENCE OF THE MULTIPLIERS

Questions may arise regarding whether the multipliers $\mathbf{z}^t$ in Algorithm 1 are bounded, given that $\|\mathbf{z}^t\|_2^2 \leq Z\beta^t$, as stated in Lemma 4.10. We argue that the boundedness of the multipliers is not an issue. We propose the following variable substitution: $\frac{\mathbf{z}^t}{\sqrt{\beta^t}} \triangleq \hat{\mathbf{z}}^t$ for all $t$. Consequently, we can implement the following update rule to replace the dual variable update rule of Algorithm 1: $\hat{\mathbf{z}}^{t+1} = \hat{\mathbf{z}}^t \frac{\sqrt{\beta^t}}{\sqrt{\beta^{t+1}}} + \frac{\beta^t}{\sqrt{\beta^{t+1}}} \cdot \sigma(\mathbf{A}\mathbf{x}^{t+1} - \mathbf{b})$. Additionally, $\mathbf{z}^t$ should be replaced with $\sqrt{\beta^t} \cdot \hat{\mathbf{z}}^t$ in the remaining steps of Algorithm 1. Importantly, such a substitution does not essentially alter the algorithm or our analysis throughout this paper.

We have the following results for the new multipliers $\hat{\mathbf{z}}^t$:

**Lemma 4.17.** *(Proof in Appendix D.14) We have: (a) $\forall t \geq 0$, $\|\hat{\mathbf{z}}^t\|_2^2 \leq Z$; (b) $\sum_{t=1}^{\infty} \|\hat{\mathbf{z}}^{t+1} - \hat{\mathbf{z}}^t\|_2^2 \leq 2(\ddot{Z} + Z)$. Here, $\{\ddot{Z}, Z\}$ are bounded constants defined in Lemma 4.10.*

**Remark 4.18.** *Thanks to the variable substitution, the new multiplier $\|\hat{\mathbf{z}}^t\|$ is bounded and convergent with $\left(\min_{t=1}^{T} \|\hat{\mathbf{z}}^{t+1} - \hat{\mathbf{z}}^t\|_2^2\right) \leq \frac{1}{T} \sum_{t=1}^{T} \|\hat{\mathbf{z}}^{t+1} - \hat{\mathbf{z}}^t\|_2^2 \leq \mathcal{O}(1/T)$.*

## 5 EXPERIMENTS

This section assesses the performance of IPDS-ADMM in solving the sparse PCA problem, as shown in Section 2.

▶ **Compared Methods**. We compare IPDS-ADMM against three state-of-the-art general-purpose algorithms that solve Problem (1): (***i***) the Subgradient method (SubGrad) (Li et al., 2021; Davis & Drusvyatskiy, 2019), (***ii***) the Smoothing Proximal Gradient Method (SPGM) (Böhm & Wright, 2021), (***iii***) the Riemannian ADMM with fixed and large penalty (RADMM) (Li et al., 2022).

▶ **Experimental Settings**. All methods are implemented in MATLAB on an Intel 2.6 GHz CPU with 64 GB RAM. We incorporate a set of 8 datasets into our experiments, comprising both randomly generated and publicly available real-world data. Appendix Section E describes how to generate the data used in the experiments. For IPDS-ADMM, the relaxation parameter $\sigma$ is set to be around the golden ratio 1.618, as suggested by (Li et al., 2016). Additionally, we set $(\xi, p, \delta, \theta_1, \theta_2) = (1/2, 1/3, 1/4, 1.01, 0.60)$. We denote $\dot\rho$ as the regularization parameter for sparse PCA model. The penalty parameter for RADMM is set to a reasonably large constant $\beta = 100\dot\rho$. We fix $\dot r = 20$ and compare objective values for all methods after running $T'$ seconds, where $T'$ is reasonably large to ensure the proposed method converges. The corresponding MATLAB code is available on the author's research webpage.

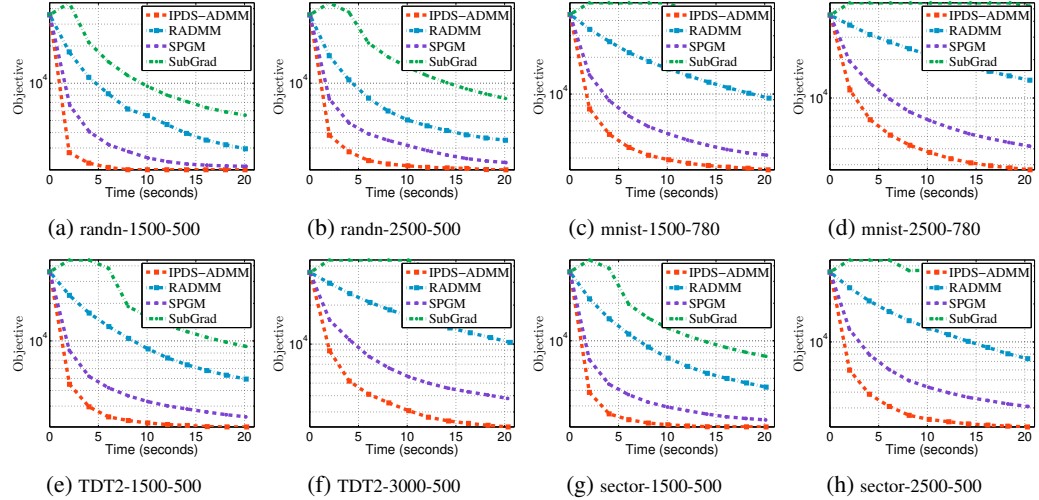

(a) randn-1500-500    (b) randn-2500-500    (c) mnist-1500-780    (d) mnist-2500-780

(e) TDT2-1500-500    (f) TDT2-3000-500    (g) sector-1500-500    (h) sector-2500-500

Figure 1: Convergence curves of methods for sparse PCA with $\dot\rho = 100$ and $\beta^0 = 50\dot\rho$.

▶ **Experiment Results**. We set $\dot\rho = 100$ and present the results for $\beta^0 = 50\dot\rho$ in IPDS-ADMM, as shown in Figure 1. These experimental results provide the following insights: (***i***) Sub-Grad tends to be less efficient in comparison to other methods. (***ii***) SPGM, utilizing a variable smoothing strategy, generally demonstrates slower performance than the multiplier-based variable splitting method. This observation corroborates the widely accepted notion that primal-dual methods are typically more robust and quicker than primal-only methods. (***iii***) The proposed IPDS-ADMM consistently achieves the lowest objective function values among all methods examined.

Due to space limitations, additional experimental results are provided in Appendix Section E.

## 6 CONCLUSIONS

In this paper, we introduce IPDS-ADMM, a proximal linearized ADMM that uses an Increasing Penalization and Decreasing Smoothing (IPDS) strategy for solving general multi-block noncon-vex composite optimization problems. IPDS-ADMM operates under a relatively relaxed condition, requiring continuity in just one block of the objective function. It incorporates relaxed strategies for dual variable updates when the associated linear operator is either bijective or surjective. We increase the penalty parameter and decrease the smoothing parameter at a controlled pace, and in-troduce a Lyapunov function for convergence analysis. We also derive the iteration complexity of IPDS-ADMM. Finally, we conduct experiments to demonstrate the effectiveness of our approaches.

## ACKNOWLEDGMENTS

This work was supported by NSFC (12271278, 61772570), and Guangdong Natural Science Funds for Distinguished Young Scholar (2018B030306025).

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

# Appendix

The organization of the appendix is as follows:

Appendix A covers notations, technical preliminaries, and relevant lemmas.

Appendix B provides additional motivating applications.

Appendix C contains proofs related to Section 3.

Appendix D offers proofs related to Section 4.

Appendix E includes additional experiments details and results.

## A  NOTATIONS, TECHNICAL PRELIMINARIES, AND RELEVANT LEMMAS

### A.1  NOTATIONS

We use the following notations in this paper.

- $[n]$: $\{1, 2, ..., n\}$.
- $\mathbf{x}$: $\mathbf{x} \triangleq \{\mathbf{x}_1, \mathbf{x}_2, \ldots, \mathbf{x}_n\} = \mathbf{x}_{[n]}$.
- $\mathbf{x}_{[i,j]}$: $\mathbf{x}_{[i,j]} \triangleq \{\mathbf{x}_i, \mathbf{x}_{i+1}, \mathbf{x}_{i+2}, \ldots, \mathbf{x}_j\}$, where $j \geq i$.
- $\mathsf{L}_i^t$: $\mathsf{L}_i^t = L_i + \beta^t \|\mathbf{A}_i\|_2^2$. Note that the function $G(\mathbf{x}, \mathbf{z}^t; \beta^t)$ is $\mathsf{L}_i^t$-smooth *w.r.t.* $\mathbf{x}_i$.
- $\sigma_1$: $\sigma_1 \triangleq \frac{\sigma}{(1-|1-\sigma|)^2} \in \mathbb{R}$, where $\sigma \in (0, 2)$. Refer to Lemma A.2.
- $\sigma_2$: $\sigma_2 \triangleq \frac{|1-\sigma|}{\sigma(1-|1-\sigma|)} \in \mathbb{R}$, where $\sigma \in (0, 2)$. Refer to Lemma A.2.
- $\|\mathbf{x}\|$: Euclidean norm: $\|\mathbf{x}\| = \|\mathbf{x}\|_2 = \sqrt{\langle \mathbf{x}, \mathbf{x} \rangle}$.
- $\langle \mathbf{a}, \mathbf{b} \rangle$ : Euclidean inner product, i.e., $\langle \mathbf{a}, \mathbf{b} \rangle = \sum_i \mathbf{a}_i \mathbf{b}_i$.
- $\mathbf{A}^\mathsf{T}$ : the transpose of the matrix $\mathbf{A}$.
- $\mathbf{x}_i$: the $i$-th block of the vector $\mathbf{x} \in \mathbb{R}^{(\mathbf{d}_1 + \mathbf{d}_2 + \ldots + \mathbf{d}_n) \times 1}$ with $\mathbf{x}_i \in \mathbb{R}^{\mathbf{d}_i \times 1}$.
- $\overline{\lambda}$: the largest eigenvalue of the matrix $\mathbf{A}_n \mathbf{A}_n^\mathsf{T}$.
- $\underline{\lambda}$: the smallest eigenvalue of the matrix $\mathbf{A}_n \mathbf{A}_n^\mathsf{T}$.
- $\underline{\lambda}'$: the smallest eigenvalue of the matrix $\mathbf{A}_n^\mathsf{T} \mathbf{A}_n$.
- $\|\mathbf{A}\|$: the spectral norm of the matrix $\mathbf{A}$.
- $\mathbf{I}_r$ : $\mathbf{I}_r \in \mathbb{R}^{r \times r}$, the identity matrix; the subscript is omitted at times.
- $\iota_\Omega(\mathbf{x})$ : Indicator function of a set $\Omega$ with $\iota_\Omega(\mathbf{x}) = 0$ if $\mathbf{x} \in \Omega$ and otherwise $+\infty$.
- $\mathrm{vec}(\mathbf{V})$ : Vector formed by stacking the column vectors of $\mathbf{V}$ with $\mathrm{vec}(\mathbf{V}) \in \mathbb{R}^{d' \times r'}$.
- $\mathrm{mat}(\mathbf{x})$ : Convert $\mathbf{x} \in \mathbb{R}^{(d' \cdot r') \times 1}$ into a matrix with $\mathrm{mat}(\mathrm{vec}(\mathbf{V})) = \mathbf{V}$ with $\mathrm{mat}(\mathbf{x}) \in \mathbb{R}^{d' \times r'}$.
- $\mathrm{dist}(\Omega, \Omega')$ : distance between two sets with $\mathrm{dist}(\Omega, \Omega') \triangleq \inf_{\mathbf{w} \in \Omega, \mathbf{w}' \in \Omega'} \|\mathbf{w} - \mathbf{w}'\|$.

### A.2  TECHNICAL PRELIMINARIES

We present some tools in non-smooth analysis including Fréchet subdifferential, and limiting (Fréchet) subdifferential (Mordukhovich, 2006; Rockafellar & Wets., 2009; Bertsekas, 2015). For any extended real-valued (not necessarily convex) function $F : \mathbb{R}^n \rightarrow (-\infty, +\infty]$, its domain is defined by

$$\mathrm{dom}(F) \triangleq \{\mathbf{x} \in \mathbb{R}^n : -\infty < F(\mathbf{x}) < +\infty\}.$$

The Fréchet subdifferential of $F$ at $\mathbf{x} \in \mathrm{dom}(F)$, denoted as $\hat{\partial} F(\mathbf{x})$, is defined as

$$\hat{\partial} F(\mathbf{x}) \triangleq \{\mathbf{v} \in \mathbb{R}^n : \lim_{\substack{\mathbf{z} \to \mathbf{x} \\ \mathbf{z} \neq \mathbf{x}}} \inf \frac{F(\mathbf{z}) - F(\mathbf{x}) - \langle \mathbf{v}, \mathbf{z} - \mathbf{x} \rangle}{\|\mathbf{z} - \mathbf{x}\|} \geq 0\}.$$

The limiting subdifferential of $F(\mathbf{x})$ at $\mathbf{x} \in \mathrm{dom}(F)$ is defined as:

$$\partial F(\mathbf{x}) \triangleq \{\mathbf{v} \in \mathbb{R}^n : \exists \mathbf{x}^k \to \mathbf{x}, F(\mathbf{x}^k) \to F(\mathbf{x}), \mathbf{v}^k \in \hat{\partial} F(\mathbf{x}^k) \to \mathbf{v}, \forall k\}.$$

These subdifferentials satisfy the following key properties:

**(a)** It holds that $\hat{\partial} F(\mathbf{x}) \subseteq \partial F(\mathbf{x})$.

**(b)** If $F(\cdot)$ is differentiable at $\mathbf{x}$, then $\hat{\partial} F(\mathbf{x}) = \partial F(\mathbf{x}) = \{\nabla F(\mathbf{x})\}$ with $\nabla F(\mathbf{x})$ being the gradient of $F(\cdot)$ at $\mathbf{x}$.

**(c)** When $F(\cdot)$ is convex, both $\hat{\partial} F(\mathbf{x})$ and $\partial F(\mathbf{x})$ reduce to the classical subdifferential for convex functions, i.e., $\hat{\partial} F(\mathbf{x}) = \partial F(\mathbf{x}) = \{\mathbf{v} : F(\mathbf{z}) - F(\mathbf{x}) - \langle \mathbf{v}, \mathbf{z} - \mathbf{x} \rangle \geq 0, \forall \mathbf{z}\}$.

## A.3 RELEVANT LEMMAS

We present several useful lemmas, each independent of context and specific methodology.

**Lemma A.1.** (Pythagoras Relation) For any vectors $\mathbf{a} \in \mathbb{R}^n$, $\mathbf{b} \in \mathbb{R}^n$, $\mathbf{c} \in \mathbb{R}^n$, we have:

$$\tfrac{1}{2}\|\mathbf{a} - \mathbf{b}\|_2^2 - \tfrac{1}{2}\|\mathbf{c} - \mathbf{b}\|_2^2 = \tfrac{1}{2}\|\mathbf{a} - \mathbf{c}\|_2^2 + \langle \mathbf{b} - \mathbf{c}, \mathbf{c} - \mathbf{a} \rangle.$$
$$\tfrac{1}{2}\|\mathbf{b}\|_2^2 - \tfrac{1}{2}\|\mathbf{c} - \mathbf{b}\|_2^2 = \tfrac{1}{2}\|\mathbf{c}\|_2^2 + \langle \mathbf{b} - \mathbf{c}, \mathbf{c} \rangle.$$

**Lemma A.2.** *Assume $\sigma \in (0, 2)$. Let $\mathbf{b}^+ = \sigma \mathbf{a} + (1 - \sigma)\mathbf{b}$, where $\mathbf{b}^+ \in \mathbb{R}^n$, $\mathbf{b} \in \mathbb{R}^n$, and $\mathbf{a} \in \mathbb{R}^n$. We have:*

$$\tfrac{1}{\sigma}\|\mathbf{b}^+\|_2^2 \leq \sigma_1 \|\mathbf{a}\|_2^2 + \sigma_2(\|\mathbf{b}\|_2^2 - \|\mathbf{b}^+\|_2^2),$$

*where $\sigma_1 \triangleq \frac{\sigma}{(1 - |1 - \sigma|)^2}$, and $\sigma_2 \triangleq \frac{|1 - \sigma|}{\sigma(1 - |1 - \sigma|)}$.*

*Proof.* **Part (a).** When $\sigma = 1$, we have $\sigma_1 = 1$, $\sigma_2 = 0$, and $\mathbf{b}^+ = \mathbf{a}$. The conclusion of this lemma clearly holds.

**Part (b).** We now focus on the case when $\sigma \neq 1$. Noticing $|1 - \sigma| \neq 0$ and $1 - |1 - \sigma| \neq 0$, we rewrite $\mathbf{b}^+ = (1 - \sigma)\mathbf{b} + \sigma \mathbf{a}$ into the following equivalent equality:

$$\mathbf{b}^+ = (1 - |1 - \sigma|) \cdot \tfrac{\sigma \mathbf{a}}{1 - |1 - \sigma|} + |1 - \sigma| \cdot \tfrac{(1 - \sigma)\mathbf{b}}{|1 - \sigma|}.$$

Using the fact that the function $\|\cdot\|_2^2$ is convex and $|1 - \sigma| \in (0, 1)$, we derive the following results:

$$\begin{aligned}
\|\mathbf{b}^+\|_2^2 &\leq (1 - |1 - \sigma|) \cdot \|\tfrac{\sigma \mathbf{a}}{1 - |1 - \sigma|}\|_2^2 + |1 - \sigma| \cdot \|\tfrac{(1 - \sigma)\mathbf{b}}{|1 - \sigma|}\|_2^2 \\
&= \tfrac{\sigma^2}{1 - |1 - \sigma|} \cdot \|\mathbf{a}\|_2^2 + |1 - \sigma| \cdot \|\mathbf{b}\|_2^2.
\end{aligned}$$

Subtracting $(|1 - \sigma| \cdot \|\mathbf{b}^+\|_2^2)$ from both sides of the above inequality, we have:

$$(1 - |1 - \sigma|)\|\mathbf{b}^+\|_2^2 \leq \tfrac{\sigma^2}{1 - |1 - \sigma|} \cdot \|\mathbf{a}\|_2^2 + |1 - \sigma|(\|\mathbf{b}\|_2^2 - \|\mathbf{b}^+\|_2^2).$$

Dividing both sides by $\sigma(1 - |1 - \sigma|)$, we have:

$$\tfrac{1}{\sigma}\|\mathbf{b}^+\|_2^2 \leq \tfrac{\sigma}{(1 - |1 - \sigma|)^2}\|\mathbf{a}\|_2^2 + \tfrac{|1 - \sigma|}{\sigma(1 - |1 - \sigma|)}(\|\mathbf{b}\|_2^2 - \|\mathbf{b}^+\|_2^2).$$

Using the definition of $\sigma_1$ and $\sigma_2$, we finish the proof of this lemma.

$\square$

**Lemma A.3.** *We let $t \geq 1$, and $q \in (0, 1)$. We have: $\frac{1}{q}(t + 1)^q - \frac{1}{q} \geq \frac{1}{2}t^q$.*

*Proof.* We let $h(t) \triangleq (t + 1)^q - 1 - \frac{q}{2}t^q$.

Initially, we prove that $f(q) \triangleq 2^q - \frac{q}{2} - 1 \geq 0$ for all $q \geq 0$. Given $\nabla f(q) = 2^q \log(2) - \frac{1}{2} \geq 2^0 \log(2) - \frac{1}{2} = 0.1931 > 0$, the function $f(q)$ is increasing for all $q \geq 0$. Combining with the fact that $f(0) = 0$, we have: $f(q) \geq 0$ for all $q \geq 0$.

We derive the following inequalities:

$$\nabla h(t) = qt^{q-1} \cdot \{(\tfrac{t+1}{t})^{q-1} - \tfrac{q}{2}\} \overset{\text{①}}{\geq} qt^{p-1} \cdot \{2^{q-1} - \tfrac{q}{2}\} \overset{\text{②}}{\geq} qt^{q-1} \cdot \{\tfrac{q/2+1}{2} - \tfrac{q}{2}\} \overset{\text{③}}{\geq} 0,$$

where step ① uses $\frac{t+1}{t} \leq 2$ and $q - 1 \leq 0$; step ② uses $2^q \geq \frac{q}{2} + 1$ for all $q \geq 0$; step ③ uses $1 - q \geq 0$. Therefore, $h(t)$ is an increasing function.

Finally, noticing that $h(1) = 2^q - 1 - \frac{q}{2} \geq 0$, we conclude that $h(t) \geq 0$ for all $t \geq 1$.

$\square$

**Lemma A.4.** *We let $p \in (0, 1)$ and $t \geq 1$. We have:* $(t + 1)^p - t^p \leq pt^{p-1}$.

*Proof.* We notice that $h(t) \triangleq t^p$ is concave for all $t \geq 1$ and $p \in (0, 1)$ since $\nabla h(t) = pt^{p-1}$ and $\nabla^2 h(t) = p(p - 1)t^{p-2} < 0$. It follows that: $\forall x, y \geq 1, h(y) - h(x) \leq \langle y - x, \nabla h(x) \rangle$. Letting $x = t$ and $y = t + 1$, for all $t \geq 1$ and $p \in (0, 1)$, we have: $(t + 1)^p - t^p \leq pt^{p-1}$.

$\square$

**Lemma A.5.** *We let $p \in (0, 1)$. We have:* $\sum_{t=1}^{\infty} (\frac{(t+1)^p - t^p}{t^p})^2 \leq 2$.

*Proof.* We have:

$$\sum_{t=1}^{\infty} \left(\frac{(t+1)^p - t^p}{t^p}\right)^2 \stackrel{①}{\leq} \sum_{t=1}^{\infty} \frac{1}{t^{2p}} t^{2p-2} = \sum_{t=1}^{\infty} t^{-2} \stackrel{②}{\leq} 2,$$

where step ① uses Lemma A.4 and $p \leq 1$; step ② uses $\sum_{t=1}^{\infty} \frac{1}{t^2} \leq \sum_{t=1}^{\infty} \frac{1}{t^2} = \frac{\pi^2}{6} < 2$. $\square$

**Lemma A.6.** *We let $p \in (0, 1)$. We have:* $\frac{1}{2} T^{1-p} \leq \sum_{t=1}^{T} t^{-p} \leq \frac{T^{(1-p)}}{1-p}$.

*Proof.* We define $h(x) = x^{-p}$ and $g(x) = \frac{1}{1-p} x^{1-p}$. Clearly, we have: $\nabla g(x) = h(x)$.

By the integral test for convergence [2], we obtain: $\int_1^{T+1} h(x)dx \leq \sum_{t=1}^{T} h(t) \leq h(1) + \int_1^{T} h(x)dx$.

**Part (a).** We have: $\sum_{t=1}^{T} t^{-p} \geq \int_1^{T+1} x^{-p}dx \stackrel{①}{=} g(T+1) - g(1) = \frac{1}{1-p}(T+1)^{1-p} - \frac{1}{1-p} \stackrel{②}{\geq} \frac{1}{2} T^{1-p}$, where step ① uses $\nabla g(x) = h(x) = x^{-p}$; step ② uses Lemma A.3 with $q = 1 - p$ and $t = T$.

**Part (b).** We have: $\sum_{t=1}^{T} t^{-p} \leq h(1) + \int_1^{T} x^{-p}dx \stackrel{①}{=} 1 + g(T) - g(1) = 1 + \frac{1}{1-p}(T)^{1-p} - \frac{1}{1-p} = \frac{T^{(1-p)} - p}{1-p} < \frac{T^{(1-p)}}{1-p}$, where step ① uses $h(1) = 1$, and $\nabla g(x) = h(x) = x^{-p}$.

$\square$

**Lemma A.7.** *Let $\sigma \in (0, 2)$, and $e^{t+1} - |1 - \sigma|e^t \leq \sigma p^t$ for all $t \geq 1$. We have:* $e^t \leq e^1 + \sigma_3 \max_{i=1}^{t-1} p^i$, *where* $\sigma_3 = \frac{\sigma}{1 - |1-\sigma|} \in [1, \infty)$.

*Proof.* Given $\sigma \in (0, 2)$, we define $\sigma_\star \triangleq |1 - \sigma| \in [0, 1)$.

We derive the following results:

$$
\begin{aligned}
t = 1, \quad e^2 &\leq \sigma_\star e^1 + \sigma p^1 \\
t = 2, \quad e^3 &\leq \sigma_\star e^2 + \sigma p^2 \leq \sigma_\star^2 e^1 + \sigma_\star \sigma p^1 + \sigma p^2 \\
t = 3, \quad e^4 &\leq \sigma_\star e^3 + \sigma p^3 \leq \sigma_\star^3 e^1 + \sigma_\star^2 \sigma p^1 + \sigma_\star \sigma p^2 + \sigma p^3 \\
&\cdots \\
t = T, \quad e^{T+1} &\leq \sigma_\star e^T + \sigma p^T \leq \sigma_\star^T e^1 + \sigma \sum_{i=1}^{T} \sigma_\star^{T-i} p^i.
\end{aligned}
$$

Therefore, we have:

$$
\begin{aligned}
e^{T+1} &\leq \sigma_\star^T e^1 + \sigma \sum_{i=1}^{T} \sigma_\star^{T-i} p^i \\
&\stackrel{①}{\leq} e^1 + \sigma \{\max_{i=1}^{T} p^i\}\{\sum_{i=1}^{T} \sigma_\star^{T-i}\} \\
&\stackrel{②}{\leq} e^1 + \sigma \{\max_{i=1}^{T} p^i\} \frac{1}{1-\sigma_\star},
\end{aligned}
$$

---

[2] https://en.wikipedia.org/wiki/Integral_test_for_convergence

where step ① uses $\sigma_\star^T \leq 1$; step ② uses the fact that:

$$\sum_{i=1}^{T} \sigma_\star^{T-i} = \sigma_\star^{T-1} + \ldots + \sigma_\star^1 + \sigma_\star^0 = \frac{1-\sigma_\star^T}{1-\sigma_\star} \leq \frac{1}{1-\sigma_\star}.$$

$\square$

**Lemma A.8.** *Assume $\kappa \in [1,2)$, $\delta \in (0, \frac{1}{3}(\frac{2}{\kappa}-1))$. For any $\varrho > 0$, we define*

$$f(\theta) = \theta - \tfrac{1}{2} - \varrho(\delta + \theta + \theta\delta - 1/\kappa)^2.$$

*We have $f(\bar{\theta}) \geq \frac{1}{8\varrho}$, where $\bar{\theta} = \frac{1}{2\varrho(1+\delta)^2} + \frac{1/\kappa-\delta}{\delta+1}$.*

*Proof.* Initially, given $\kappa \in [1,2)$, $\delta \in (0, \frac{1}{3}(\frac{2}{\kappa}-1))$, we have:

$$\tfrac{1}{2} < \tfrac{1/\kappa-\delta}{1+\delta} < 1. \tag{15}$$

Setting the gradient of $f(\theta)$ w.r.t. $\theta$ yields: $1 - 2\varrho(\delta + \theta + \delta\theta - 1/\kappa)(1+\delta) = 0$. It follows that the solution $\bar{\theta} = \frac{1}{2\varrho(1+\delta)^2} + \frac{1/\kappa-\delta}{\delta+1}$ is the maximizer of the concave function $f(\theta)$. We have:

$$
\begin{aligned}
f(\bar{\theta}) \;\overset{①}{=}\;& \bar{\theta} - \tfrac{1}{2} - \varrho(\delta + \theta_2 + \delta\theta_2 - 1/\kappa)^2 \\
=\;& \tfrac{1}{4(1+\delta)^2\varrho} + \tfrac{1/\kappa-\delta}{\delta+1} - \tfrac{1}{2} \\
\overset{②}{\geq}\;& \tfrac{1}{4(1+\delta)^2\varrho} + 0 \\
\overset{③}{\geq}\;& \tfrac{1}{4(1+1/3)^2\varrho} \\
\overset{④}{\geq}\;& \tfrac{1}{8\varrho},
\end{aligned}
$$

where step ① uses the definitions of $f(\theta)$ and $\bar{\theta}$; step ② uses the first Inequality in (15); step ③ uses the fact that $\delta \leq \frac{1}{3}$; step ④ uses $4 \times (1 + 1/3)^2 < 8$.

$\square$

# B ADDITIONAL MOTIVATING APPLICATIONS

▶ **Robust Sparse Regression**. Robust sparse regression (Liu et al., 2019) utilizes the $\ell_1$-norm of the residuals to ensure robustness against outliers while enforcing sparsity via $\ell_0$-norm constraints to identify key variables. The problem is formulated as: $\min_{\mathbf{v}} \|\mathbf{G}\mathbf{v} - \mathbf{z}\|_1$, s.t. $\mathbf{v} \in \Omega \triangleq \{\mathbf{v} \mid \|\mathbf{v}\|_0 \leq \dot{s}\}$, where $\dot{s} \geq 0$ is an integer, $\mathbf{G} \in \mathbb{R}^{\dot{m} \times \dot{d}}$, and $\mathbf{z} \in \mathbb{R}^{\dot{m}}$. By introducing a new variable $\mathbf{y}$, this problem can be formulated as: $\min_{\mathbf{v},\mathbf{y}} \iota_\Omega(\mathbf{v}) + \|\mathbf{y}\|_1$, s.t. $-\mathbf{G}\mathbf{v} + \mathbf{y} = -\mathbf{z}$. It corresponds to Problem (1) with $\mathbf{x}_1 = \mathbf{v}$, $\mathbf{x}_2 = \mathbf{y}$, $f_1(\mathbf{x}_1) = f_2(\mathbf{x}_2) = 0$, $h_1(\mathbf{x}_1) = \iota_\Omega(\mathbf{v})$, $h_2(\mathbf{x}_2) = \|\mathbf{y}\|_1$, and $\mathbf{A}_1 = -\mathbf{G}$, $\mathbf{A}_2 = \mathbf{I}$, $\mathbf{b} = -\mathbf{z}$, and Condition $\mathbb{BI}$.

▶ **Dual Principal Component Pursuit**. Dual principal component pursuit (Tsakiris & Vidal, 2018) is used primarily in subspace clustering and outlier detection, aiming to robustly represent data structures across different subspaces in the presence of noise and outliers. The problem is formulated as: $\min_{\mathbf{V}} \|\mathbf{G}\mathbf{V}\|_{2,1}$, s.t. $\mathbf{V} \in \Omega \triangleq \{\mathbf{V} \mid \mathbf{V}^\mathsf{T}\mathbf{V} = \mathbf{I}\}$, where $\mathbf{G} \in \mathbb{R}^{\dot{m} \times \dot{d}}$, and $\|\mathbf{Y}\|_{2,1} \triangleq \sum_i \|\mathbf{Y}(i,:)\|$. By introducing a new variable $\mathbf{Y}$, this problem can be formulated as: $\min_{\mathbf{V},\mathbf{Y}} \iota_\Omega(\mathbf{V}) + \|\mathbf{Y}\|_{2,1}$, s.t. $-\mathbf{G}\mathbf{V} + \mathbf{Y} = \mathbf{0}$. It corresponds to Problem (1) with $\mathbf{x}_1 = \mathrm{vec}(\mathbf{V})$, $\mathbf{x}_2 = \mathrm{vec}(\mathbf{Y})$, $f_1(\mathbf{x}_1) = f_2(\mathbf{x}_1) = 0$, $h_1(\mathbf{x}_1) = \iota_\Omega(\mathbf{V})$, $h_2(\mathbf{x}_2) = \|\mathbf{Y}\|_{2,1}$, and $\mathbf{A}_1 = -\mathbf{G}$, $\mathbf{A}_2 = \mathbf{I}$, $\mathbf{b} = \mathbf{0}$, and Condition $\mathbb{BI}$.

▶ **Robust Low-Rank Approximation**. Robust low-rank approximation (Candès et al., 2011) uses the $\ell_1$-norm of the residuals to ensure robustness against outliers while imposing a low-rank constraint on the solution matrix The problem is formulated as: $\min_{\mathbf{V}} \|\mathbf{G}(\mathbf{V}) - \mathbf{z}\|_1$, s.t. $\mathbf{V} \triangleq \{\mathbf{V} \mid \mathrm{rank}(\mathbf{V}) \leq \dot{s}\}$, where $\dot{s} \geq 0$ is an integer, $\mathbf{G}(\cdot) : \mathbb{R}^{\dot{d} \times \dot{r}} \mapsto \mathbb{R}^{\dot{m}}$, and $\mathbf{z} \in \mathbb{R}^{\dot{m}}$. By introducing a new variable $\mathbf{y}$, this problem can be formulated as: $\min_{\mathbf{V},\mathbf{y}} \iota_\Omega(\mathbf{V}) + \|\mathbf{y}\|_1$, s.t. $-\mathbf{G}(\mathbf{V}) + \mathbf{y} = -\mathbf{z}$. It corresponds to Problem (1) with $\mathbf{x}_1 = \mathrm{vec}(\mathbf{V})$, $\mathbf{x}_2 = \mathbf{y}$, $f_1(\mathbf{x}_1) = f_2(\mathbf{x}_1) = 0$, $h_1(\mathbf{x}_1) = \iota_\Omega(\mathbf{V})$, $h_2(\mathbf{x}_2) = \|\mathbf{y}\|_1$, $\mathbf{A}_1\mathbf{x}_1 = -\mathbf{G}(\mathbf{V})$, $\mathbf{A}_2 = \mathbf{I}$, $\mathbf{b} = -\mathbf{z}$, and Condition $\mathbb{BI}$.

## C    PROOFS FOR SECTION 3

### C.1    PROOF OF LEMMA 3.1

*Proof.* Consider the update rule $\beta^t = \beta^0 + \beta^0 \xi t^p$, where $p \in (0, 1)$.

**Part (a).** We have:

$$\beta^{t+1} - \beta^t - \xi\beta^t \overset{①}{=} \beta^0\xi((t+1)^p - t^p) - \xi\beta^0 \overset{②}{\leq} \beta^0\xi - \beta^0\xi = 0,$$

where step ① uses the update rule $\beta^t = \beta^0 + \beta^0\xi t^p$; step ② uses the fact that the function $h(t) \triangleq (t+1)^p - t^p$ is monotonically decreasing *w.r.t.* $t$ that: $h(t) \leq h(0) = 1$.

**Part (b).** We derive: $L_n \leq \beta^0\delta\overline{\lambda} \overset{①}{\leq} \beta^t\delta\overline{\lambda}$, where step ① uses $\beta^t \geq \beta^0$.

$\square$

### C.2    PROOF OF LEMMA 3.4

*Proof.* We let $\mathbf{u} \in \mathbb{R}^d$ be a fixed constant vector. We assume $0 < \mu_2 < \mu_1$.

We define: $h(\mathbf{u}; \mu_1) \triangleq \min_{\mathbf{v}} h(\mathbf{v}) + \frac{1}{2\mu_1}\|\mathbf{v} - \mathbf{u}\|_2^2$, and $h(\mathbf{u}; \mu_2) \triangleq \min_{\mathbf{v}} h(\mathbf{v}) + \frac{1}{2\mu_2}\|\mathbf{v} - \mathbf{u}\|_2^2$.

We define $\mathbf{p}_1 \triangleq \mathrm{Prox}_h(\mathbf{u}; \mu_1) \triangleq \arg\min_{\mathbf{v}} h(\mathbf{v}) + \frac{1}{2\mu_1}\|\mathbf{v} - \mathbf{u}\|_2^2$.

We define $\mathbf{p}_2 \triangleq \mathrm{Prox}_h(\mathbf{u}; \mu_2) \triangleq \arg\min_{\mathbf{v}} h(\mathbf{v}) + \frac{1}{2\mu_2}\|\mathbf{v} - \mathbf{u}\|_2^2$.

We let $\mathbf{g}_1 \in \partial h(\mathrm{Prox}_h(\mathbf{u}; \mu_1))$, and $\mathbf{g}_2 \in \partial h(\mathrm{Prox}_h(\mathbf{u}; \mu_2))$.

Initially, by the optimality of $\mathbf{p}_1 \triangleq \mathrm{Prox}_h(\mathbf{u}; \mu_1)$ and $\mathbf{p}_2 \triangleq \mathrm{Prox}_h(\mathbf{u}; \mu_2)$, we obtain:

$$\mathbf{u} - \mathbf{p}_1 \in \mu_1\partial h(\mathrm{Prox}_h(\mathbf{u}; \mu_1)) = \mu_1\mathbf{g}_1, \tag{16}$$
$$\mathbf{u} - \mathbf{p}_2 \in \mu_2\partial h(\mathrm{Prox}_h(\mathbf{u}; \mu_2)) = \mu_2\mathbf{g}_2. \tag{17}$$

**Part (a).** We now prove that $0 \leq \frac{h(\mathbf{u};\mu_2) - h(\mathbf{u};\mu_1)}{\mu_1 - \mu_2}$. We have:

$$
\begin{aligned}
h(\mathbf{u}; \mu_1) - h(\mathbf{u}; \mu_2) \quad &\overset{①}{=} \quad \tfrac{1}{2\mu_1}\|\mathbf{u} - \mathbf{p}_1\|_2^2 - \tfrac{1}{2\mu_2}\|\mathbf{u} - \mathbf{p}_2\|_2^2 + h(\mathbf{p}_1) - h(\mathbf{p}_2) \\
&\overset{②}{\leq} \quad \tfrac{1}{2\mu_1}\|\mathbf{u} - \mathbf{p}_1\|_2^2 - \tfrac{1}{2\mu_2}\|\mathbf{u} - \mathbf{p}_2\|_2^2 + \langle \mathbf{p}_1 - \mathbf{p}_2, \mathbf{g}_1\rangle \\
&\overset{③}{=} \quad \tfrac{\mu_1}{2}\|\mathbf{g}_1\|_2^2 - \tfrac{\mu_2}{2}\|\mathbf{g}_2\|_2^2 + \langle \mu_2\mathbf{g}_2 - \mu_1\mathbf{g}_1, \mathbf{g}_1\rangle \\
&= \quad -\tfrac{\mu_1}{2}\|\mathbf{g}_1\|_2^2 - \tfrac{\mu_2}{2}\|\mathbf{g}_2\|_2^2 + \mu_2\langle \mathbf{g}_2, \mathbf{g}_1\rangle \\
&\overset{④}{\leq} \quad -\tfrac{\mu_2}{2}\|\mathbf{g}_1\|_2^2 - \tfrac{\mu_2}{2}\|\mathbf{g}_2\|_2^2 + \mu_2\langle \mathbf{g}_2, \mathbf{g}_1\rangle \\
&= \quad -\tfrac{\mu_2}{2}\|\mathbf{g}_2 - \mathbf{g}_1\|_2^2 \leq 0,
\end{aligned}
$$

where step ① uses the definition of $h(\mathbf{u}; \mu)$; step ② uses the convexity of $h(\cdot)$; step ③ uses the optimality of $\mathbf{p}_1 \triangleq \mathrm{Prox}_h(\mathbf{u}; \mu_1)$ and $\mathbf{p}_2 \triangleq \mathrm{Prox}_h(\mathbf{u}; \mu_2)$ as in (16) and (17); step ④ uses $\mu_2 < \mu_1$.

**Part (b).** We now prove that $\frac{h(\mathbf{u};\mu_2) - h(\mathbf{u};\mu_1)}{\mu_1 - \mu_2} \leq \frac{1}{2}C_h^2$. We have:

$$
\begin{aligned}
h(\mathbf{u}; \mu_2) - h(\mathbf{u}; \mu_1) \quad &\overset{①}{=} \quad \tfrac{1}{2\mu_2}\|\mathbf{u} - \mathbf{p}_2\|_2^2 - \tfrac{1}{2\mu_1}\|\mathbf{u} - \mathbf{p}_1\|_2^2 + h(\mathbf{p}_2) - h(\mathbf{p}_1) \\
&\overset{②}{\leq} \quad \tfrac{1}{2\mu_2}\|\mathbf{u} - \mathbf{p}_2\|_2^2 - \tfrac{1}{2\mu_1}\|\mathbf{u} - \mathbf{p}_1\|_2^2 + \langle \mathbf{p}_2 - \mathbf{p}_1, \mathbf{g}_2\rangle \\
&\overset{③}{=} \quad \tfrac{\mu_2}{2}\|\mathbf{g}_2\|_2^2 - \tfrac{\mu_1}{2}\|\mathbf{g}_1\|_2^2 + \langle \mu_1\mathbf{g}_1 - \mu_2\mathbf{g}_2, \mathbf{g}_2\rangle \\
&= \quad -\tfrac{\mu_2}{2}\|\mathbf{g}_2\|_2^2 - \tfrac{\mu_1}{2}\|\mathbf{g}_1\|_2^2 + \mu_1\langle \mathbf{g}_2, \mathbf{g}_1\rangle \\
&\overset{④}{\leq} \quad \tfrac{\mu_1}{2}\|\mathbf{g}_2\|_2^2 - \tfrac{\mu_2}{2}\|\mathbf{g}_2\|_2^2 \\
&\overset{⑤}{\leq} \quad \tfrac{\mu_1 - \mu_2}{2} \cdot C_h^2,
\end{aligned}
$$

where step ① uses the definition of $h(\mathbf{u}; \mu)$; step ② uses the convexity of $h(\cdot)$; step ③ uses the optimality of $\mathbf{p}_1 \triangleq \mathrm{Prox}_h(\mathbf{u}; \mu_1)$ and $\mathbf{p}_2 \triangleq \mathrm{Prox}_h(\mathbf{u}; \mu_2)$ as in (16) and (17); step ④ uses the inequality that: $-\frac{1}{2}\|\mathbf{g}_1\|_2^2 + \langle \mathbf{g}_1, \mathbf{g}_2 \rangle \leq \frac{1}{2}\|\mathbf{g}_2\|_2^2$ for all $\mathbf{g}_1, \mathbf{g}_2 \in \mathbb{R}^{d \times 1}$; step ⑤ uses $\|\mathbf{g}_2\| \leq C_h$.  □

## C.3   PROOF OF LEMMA 3.5

*Proof.* We let $\mathbf{u}$ be a fixed constant vector. We assume $0 < \mu_2 < \mu_1$.

We define: $h(\mathbf{u}; \mu) \triangleq \min_{\mathbf{v} \in \mathbb{R}^{d \times 1}} h(\mathbf{v}) + \frac{1}{2\mu}\|\mathbf{v} - \mathbf{u}\|_2^2$.

We define: $\mathrm{Prox}_h(\mathbf{u}; \mu) \triangleq \arg\min_{\mathbf{v} \in \mathbb{R}^{d \times 1}} h(\mathbf{v}) + \frac{1}{2\mu}\|\mathbf{v} - \mathbf{u}\|_2^2$.

Using Claim (*b*) of Lemma 3.3, we establish that $h(\mathbf{u}; \mu)$ is smooth *w.r.t.* $\mathbf{u}$, and its gradient can be computed as:

$$\nabla h(\mathbf{u}; \mu) = \mu^{-1}(\mathbf{u} - \mathrm{Prox}_h(\mathbf{u}; \mu)).$$

We examine the following mapping $\mathcal{H}(\upsilon) \triangleq \upsilon(\mathbf{u} - \mathrm{Prox}_h(\mathbf{u}; \frac{1}{\upsilon}))$ with $\mathcal{H}(\upsilon) : \mathbb{R} \mapsto \mathbb{R}^n$. We derive:

$$\lim_{\delta \to 0} \tfrac{\mathcal{H}(\upsilon+\delta)-\mathcal{H}(\upsilon)}{\delta} = \lim_{\delta \to 0} \tfrac{(\upsilon+\delta)(\mathbf{u}-\mathrm{Prox}_h(\mathbf{u};\frac{1}{\upsilon+\delta}))-\upsilon(\mathbf{u}-\mathrm{Prox}_h(\mathbf{u};\frac{1}{\upsilon}))}{\delta}$$

$$= \lim_{\delta \to 0} \tfrac{\delta\mathbf{u}-(\upsilon+\delta)\mathrm{Prox}_h(\mathbf{u};\frac{1}{\upsilon})+\upsilon\mathrm{Prox}_h(\mathbf{u};\frac{1}{\upsilon})}{\delta} = \mathbf{u} - \mathrm{Prox}_h(\mathbf{u}; \tfrac{1}{\upsilon}).$$

Therefore, the first-order derivative of the mapping $\mathcal{H}(\upsilon)$ *w.r.t.* $\upsilon$ always exists and can be computed as $\nabla_\upsilon \mathcal{H}(\upsilon) = \mathbf{u} - \mathrm{Prox}_h(\mathbf{u}; \frac{1}{\upsilon})$, leading to:

$$\forall \upsilon, \upsilon' > 0, \tfrac{\|\mathcal{H}(\upsilon)-\mathcal{H}(\upsilon')\|}{|\upsilon-\upsilon'|} \leq \|\mathbf{u} - \mathrm{Prox}_h(\mathbf{u}; \tfrac{1}{\upsilon})\|.$$

Letting $\upsilon = 1/\mu_1$ and $\upsilon' = 1/\mu_2$, we derive:

$$\tfrac{\|\nabla h(\mathbf{u};\mu_1)-\nabla h(\mathbf{u};\mu_2)\|}{|1/\mu_1-1/\mu_2|} \leq \|\mathbf{u} - \mathrm{Prox}_h(\mathbf{u}; \mu_1)\| \overset{①}{=} \mu_1 \|\partial h(\mathrm{Prox}_h(\mathbf{u}; \mu_1))\| \overset{②}{\leq} \mu_1 C_h,$$

where step ① uses the optimality of $\mathrm{Prox}_h(\mathbf{u}; \mu)$ that $\mathbf{0} \in \partial h(\mathrm{Prox}_h(\mathbf{u}; \mu)) + \frac{1}{\mu}(\mathrm{Prox}_h(\mathbf{u}; \mu) - \mathbf{u})$ for all $\mu$; step ② uses the Lipschitz continuity of $h(\cdot)$. We further obtain:

$$\|\nabla h(\mathbf{u}; \mu_1) - \nabla h(\mathbf{u}; \mu_2)\| \leq |1/\mu_1 - 1/\mu_2| \cdot \mu_1 C_h = (\mu_1/\mu_2 - 1) \cdot C_h.$$

□

## C.4   PROOF OF LEMMA 3.6

*Proof.* The proof of this lemma is similar to that of Lemma 1 in (Li et al., 2022). For completeness, we include the proof here.

We consider the following strongly convex problems:

$$\bar{\mathbf{x}}_n = \arg\min_{\mathbf{x}_n} h_n(\mathbf{x}_n; \mu) + \tfrac{\rho}{2}\|\mathbf{x}_n - \mathbf{c}\|_2^2$$

$$\Leftrightarrow (\bar{\mathbf{x}}_n, \check{\mathbf{x}}_n) = \arg\min_{\mathbf{x}_n, \check{\mathbf{x}}_n} h_n(\check{\mathbf{x}}_n) + \tfrac{1}{2\mu}\|\mathbf{x}_n - \check{\mathbf{x}}_n\|_2^2 + \tfrac{\rho}{2}\|\mathbf{x}_n - \mathbf{c}\|_2^2.$$

We have the following first-order optimality conditions:

$$\mathbf{0} = \tfrac{1}{\mu}(\bar{\mathbf{x}}_n - \check{\mathbf{x}}_n) + \rho(\bar{\mathbf{x}}_n - \mathbf{c}) \tag{18}$$

$$\mathbf{0} \in \partial h_n(\check{\mathbf{x}}_n) + \tfrac{1}{\mu}(\check{\mathbf{x}}_n - \bar{\mathbf{x}}_n). \tag{19}$$

**Part (a).** Using (18), we obtain: $\bar{\mathbf{x}}_n = \frac{1}{1/\mu+\rho}(\frac{1}{\mu}\check{\mathbf{x}}_n + \rho\mathbf{c})$. Plugging this equation into (19) yields:

$$\mathbf{0} \in \partial h_n(\check{\mathbf{x}}_n) + \tfrac{1}{\mu}(\check{\mathbf{x}}_n - \tfrac{1}{1/\mu+\rho}(\tfrac{1}{\mu}\check{\mathbf{x}}_n + \rho\mathbf{c}))$$

$$= \partial h_n(\check{\mathbf{x}}_n) + \tfrac{\rho}{1+\mu\rho}(\check{\mathbf{x}}_n - \mathbf{c}).$$

The inclusion above implies that:

$$\check{\mathbf{x}}_n = \arg\min_{\check{\mathbf{x}}_n} h_n(\check{\mathbf{x}}_n) + \tfrac{1}{2} \cdot \tfrac{\rho}{1+\mu\rho} \|\check{\mathbf{x}}_n - \mathbf{c}\|_2^2.$$

**Part (b).** We derive:

$$-\rho(\bar{\mathbf{x}}_n - \mathbf{c}) \stackrel{①}{=} \tfrac{1}{\mu}(\bar{\mathbf{x}}_n - \check{\mathbf{x}}_n) \stackrel{②}{\in} \partial h_n(\check{\mathbf{x}}_n),$$

where step ① uses (18); step ② uses (19).

**Part (c).** Using (19), we have: $\check{\mathbf{x}}_n - \bar{\mathbf{x}}_n = -\mu \partial h_n(\check{\mathbf{x}}_n)$. This leads to $\|\check{\mathbf{x}}_n - \bar{\mathbf{x}}_n\| \le \mu C_h$.

$\square$

# D PROOFS FOR SECTION 4

## D.1 PROOF OF LEMMA 4.1

*Proof.* **Part (a).** We now focus on sufficient decrease for variables $\{\mathbf{x}_1, \mathbf{x}_2, \dots, \mathbf{x}_{n-1}\}$. We define $\Phi_i^t = G(\mathbf{x}_{[1,i-1]}^{t+1}, \mathbf{x}_i^{t+1}, \mathbf{x}_{[i+1,n]}^t, \mathbf{z}^t; \beta^t) - G(\mathbf{x}_{[1,i-1]}^{t+1}, \mathbf{x}_i^t, \mathbf{x}_{[i+1,n]}^t, \mathbf{z}^t; \beta^t) + h_i(\mathbf{x}_i^{t+1}) - h_i(\mathbf{x}_i^t)$, where $i \in [n-1]$.

Noticing the function $G(\mathbf{x}_{[1,i-1]}^{t+1}, \mathbf{x}_i, \mathbf{x}_{[i+1,n]}^t, \mathbf{z}^t; \beta^t)$ is $\mathsf{L}_i^t$-smooth *w.r.t.* $\mathbf{x}_i$ for the $t$-th iteration, we have:

$$G(\mathbf{x}_{[1,i-1]}^{t+1}, \mathbf{x}_i^{t+1}, \mathbf{x}_{[i+1,n]}^t, \mathbf{z}^t; \beta^t) - G(\mathbf{x}_{[1,i-1]}^{t+1}, \mathbf{x}_i^t, \mathbf{x}_{[i+1,n]}^t, \mathbf{z}^t; \beta^t)$$
$$\le \quad \langle \mathbf{x}_i^{t+1} - \mathbf{x}_i^t, \nabla_{\mathbf{x}_i} G(\mathbf{x}_{[1,n-1]}^{t+1}, \mathbf{x}_i^t, \mathbf{x}_{[i+1,n]}^t, \mathbf{z}^t; \beta^t) \rangle + \tfrac{\mathsf{L}_i^t}{2} \|\mathbf{x}_i^{t+1} - \mathbf{x}_i^t\|_2^2. \quad (20)$$

Given $\mathbf{x}_i^{t+1}$ is the minimizer of the following optimization problem:

$$\mathbf{x}_i^{t+1} \in \arg\min_{\mathbf{x}_i} h_i(\mathbf{x}_i) + \langle \mathbf{x}_i - \mathbf{x}_i^t, \nabla_{\mathbf{x}_i} G(\mathbf{x}_{[1,n-1]}^{t+1}, \mathbf{x}_i^t, \mathbf{x}_{[i+1,n]}^t, \mathbf{z}^t; \beta^t) \rangle + \tfrac{\theta_1 \mathsf{L}_i^t}{2} \|\mathbf{x}_i - \mathbf{x}_i^t\|_2^2.$$

The optimality of $\mathbf{x}_i^{t+1}$ leads to:

$$h_i(\mathbf{x}_i^{t+1}) - h_i(\mathbf{x}_i^t) + \langle \mathbf{x}_i^{t+1} - \mathbf{x}_i^t, \nabla_{\mathbf{x}_i} G(\mathbf{x}_{[1,n-1]}^{t+1}, \mathbf{x}_i^t, \mathbf{x}_{[i+1,n]}^t, \mathbf{z}^t; \beta^t) \rangle \le -\tfrac{\theta_1 \mathsf{L}_i^t}{2} \|\mathbf{x}_i^{t+1} - \mathbf{x}_i^t\|_2^2. \quad (21)$$

Combining equations (20) and (21), we derive the following expressions:

$$\Phi_i^t \le (\tfrac{1}{2} - \tfrac{\theta_1}{2}) \cdot \mathsf{L}_i^t \|\mathbf{x}_i^{t+1} - \mathbf{x}_i^t\|_2^2.$$

Telescoping the above inequality over $i$ from 1 to $(n-1)$ leads to:

$$\sum_{i=1}^{n-1} \Phi_i^t \le \sum_{i=1}^{n-1} \{ (\tfrac{1}{2} - \tfrac{\theta_1}{2}) \cdot \mathsf{L}_i^t \|\mathbf{x}_i^{t+1} - \mathbf{x}_i^t\|_2^2 \}.$$

Therefore, we obtain:

$$\mathcal{L}(\mathbf{x}_{[1,n-1]}^{t+1}, \mathbf{x}_n^t, \mathbf{z}^t; \beta^t, \mu^t) - \mathcal{L}(\mathbf{x}^t, \mathbf{z}^t; \beta^t, \mu^t) \le \sum_{i=1}^{n-1} \{ (\tfrac{1}{2} - \tfrac{\theta_1}{2}) \cdot \mathsf{L}_i^t \|\mathbf{x}_i^{t+1} - \mathbf{x}_i^t\|_2^2 \}. \quad (22)$$

**Part (b).** We now focus on sufficient decrease for variable $\{\mathbf{x}_n\}$. Noticing the function $G(\mathbf{x}_{[1,n-1]}^{t+1}, \mathbf{x}_n, \mathbf{z}^t; \beta^t)$ is $\mathsf{L}_n^t$-smooth *w.r.t.* $\mathbf{x}_n$ for the $t$-th iteration, we have:

$$G(\mathbf{x}_{[1,n-1]}^{t+1}, \mathbf{x}_n^{t+1}, \mathbf{z}^t; \beta^t) - G(\mathbf{x}_{[1,n-1]}^{t+1}, \mathbf{x}_n^t, \mathbf{z}^t; \beta^t)$$
$$\le \quad \langle \mathbf{x}_n^{t+1} - \mathbf{x}_n^t, \nabla_{\mathbf{x}_n} G(\mathbf{x}_{[1,n-1]}^{t+1}, \mathbf{x}_n^t, \mathbf{z}^t; \beta^t) \rangle + \tfrac{\mathsf{L}_n^t}{2} \|\mathbf{x}_n^{t+1} - \mathbf{x}_n^t\|_2^2. \quad (23)$$

Since $h_n(\mathbf{x}_n; \mu^t)$ is convex, we have:

$$h_n(\mathbf{x}_n^{t+1}; \mu^t) - h_n(\mathbf{x}_n^t; \mu^t)$$
$$\le \quad \langle \mathbf{x}_n^{t+1} - \mathbf{x}_n^{t+1}, \nabla h_n(\mathbf{x}_i^{t+1}; \mu^t) \rangle$$
$$\stackrel{①}{=} \quad \langle \mathbf{x}_n^{t+1} - \mathbf{x}_n^{t+1}, -\nabla_{\mathbf{x}_n} G(\mathbf{x}_{[1,n-1]}^{t+1}, \mathbf{x}_n^t, \mathbf{z}^t; \beta^t) \rangle - \theta_2 \mathsf{L}_n^t(\mathbf{x}_n^{t+1} - \mathbf{x}_n^t) \rangle, \quad (24)$$

where step ① uses the the first-order optimality condition of $\mathbf{x}_n^{t+1}$ that:

$$0 = \nabla h_n(\mathbf{x}_n^{t+1}; \mu^t) + \nabla_{\mathbf{x}_n} G(\mathbf{x}_{[1,n-1]}^{t+1}, \mathbf{x}_n^t, \mathbf{z}^t; \beta^t)) + \theta_2 \mathsf{L}_n^t(\mathbf{x}_n^{t+1} - \mathbf{x}_n^t).$$

Adding Inequalities (23) and (24) together, we have:

$$h_n(\mathbf{x}_n^{t+1}; \mu^t) - h_n(\mathbf{x}_n^t; \mu^t) + G(\mathbf{x}_{[1,n-1]}^{t+1}, \mathbf{x}^{t+1}, \mathbf{z}^t; \beta^t) - G(\mathbf{x}_{[1,n-1]}^{t+1}, \mathbf{x}_n^t, \mathbf{z}^t; \beta^t)$$

$$\leq \quad \frac{\mathsf{L}_n^t}{2} \|\mathbf{x}_n^{t+1} - \mathbf{x}_n^t\|_2^2 - \theta_2 \mathsf{L}_n^t \|\mathbf{x}_n^{t+1} - \mathbf{x}_n^t\|_2^2$$

$$= \quad (\tfrac{1}{2} - \theta_2) \cdot \mathsf{L}_n^t \|\mathbf{x}_n^{t+1} - \mathbf{x}_n^t\|_2^2.$$

This results in the following inequality:

$$\mathcal{L}(\mathbf{x}^{t+1}, \mathbf{z}^t; \beta^t, \mu^t) - \mathcal{L}(\mathbf{x}_{[1,n-1]}^{t+1}, \mathbf{x}_n^t, \mathbf{z}^t; \beta^t, \mu^t) \leq (\tfrac{1}{2} - \theta_2) \cdot \mathsf{L}_n^t \|\mathbf{x}_n^{t+1} - \mathbf{x}_n^t\|_2^2. \tag{25}$$

**Part (c).** We now focus on sufficient decrease for variable $\{\mathbf{z}\}$. We have:

$$\mathcal{L}(\mathbf{x}^{t+1}, \mathbf{z}^{t+1}; \beta^t, \mu^t) - \mathcal{L}(\mathbf{x}^{t+1}, \mathbf{z}^t; \beta^t, \mu^t)$$

$$= \quad \langle \mathbf{A}\mathbf{x}^{t+1} - \mathbf{b}, \mathbf{z}^{t+1} - \mathbf{z}^t \rangle$$

$$\overset{①}{=} \quad \langle \tfrac{1}{\sigma \beta^t}(\mathbf{z}^{t+1} - \mathbf{z}^t), \mathbf{z}^{t+1} - \mathbf{z}^t \rangle$$

$$= \quad \tfrac{1}{\sigma \beta^t} \|\mathbf{z}^{t+1} - \mathbf{z}^t\|_2^2, \tag{26}$$

where step ① uses $\mathbf{z}^{t+1} = \mathbf{z}^t + \sigma \beta^t (\mathbf{A}\mathbf{x}^{t+1} - \mathbf{b})$ with $\mathbf{A}\mathbf{x}^{t+1} \triangleq \sum_{j=1}^n \mathbf{A}_j \mathbf{x}_j^{t+1}$.

**Part (d).** We now focus on sufficient decrease for variable $\{\beta\}$. We have:

$$\mathcal{L}(\mathbf{x}^{t+1}, \mathbf{z}^{t+1}; \beta^{t+1}, \mu^t) - \mathcal{L}(\mathbf{x}^{t+1}, \mathbf{z}^{t+1}; \beta^t, \mu^t)$$

$$= \quad (\tfrac{\beta^{t+1}}{2} - \tfrac{\beta^t}{2}) \|\mathbf{A}\mathbf{x}^{t+1} - \mathbf{b}\|_2^2$$

$$\overset{①}{=} \quad (\tfrac{\beta^{t+1}}{2} - \tfrac{\beta^t}{2}) \|\tfrac{1}{\sigma \beta^t}(\mathbf{z}^{t+1} - \mathbf{z}^t)\|_2^2$$

$$\overset{②}{\leq} \quad (\tfrac{(1+\xi)\beta^t}{2} - \tfrac{\beta^t}{2}) \|\tfrac{1}{\sigma \beta^t}(\mathbf{z}^{t+1} - \mathbf{z}^t)\|_2^2$$

$$= \quad \tfrac{\xi}{2\sigma} \cdot \tfrac{1}{\sigma \beta^t} \|\mathbf{z}^{t+1} - \mathbf{z}^t\|_2^2, \tag{27}$$

where step ① uses $\mathbf{z}^{t+1} = \mathbf{z}^t + \sigma \beta^t (\mathbf{A}\mathbf{x}^{t+1} - \mathbf{b})$; step ② uses Lemma 3.1 that $\beta^{t+1} \leq \beta^t(1 + \xi)$.

**Part (e).** We now focus on sufficient decrease for variable $\{\mu\}$. We have:

$$\mathcal{L}(\mathbf{x}^{t+1}, \mathbf{z}^{t+1}; \beta^{t+1}, \mu^{t+1}) - \mathcal{L}(\mathbf{x}^{t+1}, \mathbf{z}^{t+1}; \beta^{t+1}, \mu^t)$$

$$= \quad h_n(\mathbf{x}_n^{t+1}; \mu^{t+1}) - h_n(\mathbf{x}_n^{t+1}; \mu^t)$$

$$\overset{①}{\leq} \quad \tfrac{1}{2} C_h(\mu^t - \mu^{t+1}), \tag{28}$$

where step ① uses Lemma 3.4.

Combining Inequalities (22), (25), (26), (27), and (28), we have:

$$\mathcal{L}(\mathbf{x}^{t+1}, \mathbf{z}^{t+1}; \beta^{t+1}, \mu^{t+1}) - \mathcal{L}(\mathbf{x}^t, \mathbf{z}^t; \beta^t, \mu^t)$$

$$\leq \quad [\textstyle\sum_{i=1}^{n-1} \{(\tfrac{1}{2} - \tfrac{\theta_1}{2}) \cdot \mathsf{L}_i^t \|\mathbf{x}_i^{t+1} - \mathbf{x}_i^t\|_2^2\}] + (\tfrac{1}{2} - \theta_2) \cdot \mathsf{L}_n^t \|\mathbf{x}_n^{t+1} - \mathbf{x}_n^t\|_2^2$$

$$+ (1 + \tfrac{\xi}{2\sigma}) \cdot \tfrac{1}{\sigma \beta^t} \|\mathbf{z}^{t+1} - \mathbf{z}^t\|_2^2 + \tfrac{1}{2} C_h(\mu^t - \mu^{t+1}). \tag{29}$$

We define $\Theta_L^t \triangleq \mathcal{L}(\mathbf{x}^t, \mathbf{z}^t; \beta^t, \mu^t) + \tfrac{1}{2} C_h \mu^t$, $\varepsilon_3 \triangleq \xi$, $\varepsilon_1 \triangleq \tfrac{1}{2} \theta_1 - \tfrac{1}{2}$, and $\mathcal{E}^{t+1} \triangleq \tfrac{\varepsilon_3}{\beta^t} \|\mathbf{z}^{t+1} - \mathbf{z}^t\|_2^2 + \varepsilon_2 \mathsf{L}_n^t \|\mathbf{x}_n^{t+1} - \mathbf{x}_n^t\|_2^2 + \varepsilon_1 \sum_{i=1}^{n-1} \mathsf{L}_i^t \|\mathbf{x}_i^{t+1} - \mathbf{x}_i^t\|_2^2$. We have:

$$\mathcal{E}^{t+1} + \Theta_L^{t+1} - \Theta_L^t$$

$$\leq \quad (\tfrac{1}{2} - \theta_2 + \varepsilon_2) \cdot \mathsf{L}_n^t \|\mathbf{x}_n^{t+1} - \mathbf{x}_n^t\|_2^2 + (1 + \tfrac{\xi}{2\sigma} + \sigma\xi) \cdot \tfrac{1}{\sigma \beta^t} \|\mathbf{z}^{t+1} - \mathbf{z}^t\|_2^2.$$

$$\square$$

## D.2 PROOF OF LEMMA 4.2

*Proof.* For any $i \in [n]$, we define $\mathbf{u}_i^{t+1} \triangleq \boldsymbol{\theta}_i \mathsf{L}_i^t [\mathbf{x}_i^{t+1} - \mathbf{x}_i^t] - \beta^t \mathbf{A}_i^\mathsf{T} [\sum_{j=i}^n \mathbf{A}_j (\mathbf{x}_j^{t+1} - \mathbf{x}_j^t)]$, and let $\mathbf{w}_i^{t+1} \in \partial h_i(\mathbf{x}_i^{t+1}) + \nabla f_i(\mathbf{x}_i^t)$.

We notice that $\mathbf{x}_i^{t+1}$ is the minimizer of the following problem:

$$\mathbf{x}_i^{t+1} \in \arg\min_{\mathbf{x}_i} \frac{\theta \mathsf{L}_i^t}{2} \|\mathbf{x}_i - \mathbf{x}_i^t\|_2^2 + h_i(\mathbf{x}_i) + \langle \mathbf{x}_i - \mathbf{x}_i^t, \nabla_{\mathbf{x}_i} G(\mathbf{x}_{[1,i-1]}^{t+1}, \mathbf{x}_{[i,n]}^t, \mathbf{z}^t; \beta^t) \rangle.$$

Using the necessary first-order optimality condition of the solution $\mathbf{x}_i^{t+1}$, we have:

$$\nabla_{\mathbf{x}_i} G(\mathbf{x}_{[1,i-1]}^{t+1}, \mathbf{x}_{[i,n]}^t, \mathbf{z}^t; \beta^t) \in -\partial h_i(\mathbf{x}_i^{t+1}) - \theta \mathsf{L}_i^t(\mathbf{x}_i^{t+1} - \mathbf{x}_i^t). \tag{30}$$

Using the definition of the function $G(\mathbf{x}, \mathbf{z}; \beta) \triangleq \langle [\sum_{j=1}^n \mathbf{A}_j \mathbf{x}_j] - \mathbf{b}, \mathbf{z} \rangle + \frac{\beta}{2} \|[\sum_{j=1}^n \mathbf{A}_j \mathbf{x}_j] - \mathbf{b}\|_2^2 + \sum_{j=1}^n f_j(\mathbf{x}_j)$, we have:

$$
\begin{aligned}
& \nabla_{\mathbf{x}_i} G(\mathbf{x}_{[1,i-1]}^{t+1}, \mathbf{x}_{[i,n]}^t, \mathbf{z}^t; \beta^t) \\
=\ & \nabla f_i(\mathbf{x}_i^t) + \mathbf{A}_i^\mathsf{T} \mathbf{z}^t + \beta^t \mathbf{A}_i^\mathsf{T} \{ [\sum_{j=1}^{i-1} \mathbf{A}_j \mathbf{x}_j^{t+1}] + [\sum_{j=i}^n \mathbf{A}_j \mathbf{x}_j^t] - \mathbf{b} \} \\
=\ & \nabla f_i(\mathbf{x}_i^t) + \mathbf{A}_i^\mathsf{T} \mathbf{z}^t + \beta^t \mathbf{A}_i^\mathsf{T} \{ \mathbf{A}\mathbf{x}^{t+1} - \mathbf{b} + [\sum_{j=i}^n \mathbf{A}_j (\mathbf{x}_j^t - \mathbf{x}_j^{t+1})] \} \\
\overset{①}{=}\ & \nabla f_i(\mathbf{x}_i^t) + \mathbf{A}_i^\mathsf{T} \mathbf{z}^t + \frac{1}{\sigma} \mathbf{A}_i^\mathsf{T} (\mathbf{z}^{t+1} - \mathbf{z}^t) + \beta^t \mathbf{A}_i^\mathsf{T} \{ \sum_{j=i}^n \mathbf{A}_j (\mathbf{x}_j^t - \mathbf{x}_j^{t+1}) \},
\end{aligned} \tag{31}
$$

where step ① uses the update rule of $\mathbf{z}^{t+1}$ that $\mathbf{z}^{t+1} - \mathbf{z}^t = \sigma \beta^t (\sum_{i=1}^n \mathbf{A}_i \mathbf{x}_i^{t+1} - \mathbf{b})$. Combining the Equalities (30) and (31), we obtain the following result:

$$
\begin{aligned}
\mathbf{0} \in\ & \partial h_i(\mathbf{x}_i^{t+1}) + \boldsymbol{\theta}_i \mathsf{L}_i^t [\mathbf{x}_i^{t+1} - \mathbf{x}_i^t] + \nabla f_i(\mathbf{x}_i^t) \\
& + \mathbf{A}_i^\mathsf{T} \mathbf{z}^t + \beta^t \mathbf{A}_i^\mathsf{T} [\sum_{j=i}^n \mathbf{A}_j (\mathbf{x}_j^t - \mathbf{x}_j^{t+1})] + \frac{1}{\sigma} \mathbf{A}_i^\mathsf{T} (\mathbf{z}^{t+1} - \mathbf{z}^t).
\end{aligned}
$$

Using the definition of $\mathbf{w}_i^{t+1}$ and $\mathbf{u}_i^{t+1}$ for all $i \in [n]$, we have: $\mathbf{0} = \mathbf{w}_i^{t+1} + \mathbf{u}_i^{t+1} + \mathbf{A}_i^\mathsf{T} \mathbf{z}^t + \frac{1}{\sigma} \mathbf{A}_i^\mathsf{T} (\mathbf{z}^{t+1} - \mathbf{z}^t)$. Multiplying both sides by $\sigma \in (0, 2)$, for all $t \geq 0$, we have:

$$\mathbf{0} = \sigma \mathbf{w}_i^{t+1} + \sigma \mathbf{A}_i^\mathsf{T} \mathbf{z}^t + \mathbf{A}_i^\mathsf{T} (\mathbf{z}^{t+1} - \mathbf{z}^t) + \sigma \mathbf{u}_i^{t+1}. \tag{32}$$

Given that $t$ can take on any integer value, for all $t \geq 1$, we derive:

$$\mathbf{0} = \sigma \mathbf{w}_i^t + \sigma \mathbf{A}_i^\mathsf{T} \mathbf{z}^{t-1} + \mathbf{A}_i^\mathsf{T} (\mathbf{z}^t - \mathbf{z}^{t-1}) + \sigma \mathbf{u}_i^t. \tag{33}$$

Combining Equality (32) and Equality (33), for all $t \geq 1$, we have:

$$\mathbf{A}_i^\mathsf{T} (\mathbf{z}^{t+1} - \mathbf{z}^t) = (1 - \sigma) \mathbf{A}_i^\mathsf{T} (\mathbf{z}^t - \mathbf{z}^{t-1}) - \sigma (\mathbf{w}_i^{t+1} - \mathbf{w}_i^t) - \sigma (\mathbf{u}_i^{t+1} - \mathbf{u}_i^t). \tag{34}$$

In view of (34), we let $i = n$ and arrive at the following two distinct identities:

$$\mathbb{BI} : \underbrace{\mathbf{A}_n^\mathsf{T} (\mathbf{z}^{t+1} - \mathbf{z}^t)}_{\triangleq \mathbf{a}^{t+1}} = (1 - \sigma) \underbrace{(\mathbf{A}_n^\mathsf{T} (\mathbf{z}^t - \mathbf{z}^{t-1}))}_{\triangleq \mathbf{a}^t} + \sigma \underbrace{(\mathbf{u}_n^t - \mathbf{u}_n^{t+1} + \mathbf{w}_n^t - \mathbf{w}_n^{t+1})}_{\mathbb{c}^t}.$$

$$\mathbb{SU} : \underbrace{\mathbf{A}_n^\mathsf{T} (\mathbf{z}^{t+1} - \mathbf{z}^t) + \sigma \mathbf{u}_n^{t+1}}_{\triangleq \mathbf{a}^{t+1}} = (1 - \sigma) \underbrace{(\mathbf{A}_n^\mathsf{T} (\mathbf{z}^t - \mathbf{z}^{t-1}) + \sigma \mathbf{u}_n^t)}_{\triangleq \mathbf{a}^t} + \sigma \underbrace{(\sigma \mathbf{u}_n^t + \mathbf{w}_n^t - \mathbf{w}_n^{t+1})}_{\triangleq \mathbb{c}^t}.$$

$\square$

## D.3 PROOF OF LEMMA 4.3

*Proof.* We denote $\mathbf{Q}^t \triangleq \theta_2 \mathsf{L}_n^t \mathbf{I} - \beta^t \mathbf{A}_n^\mathsf{T} \mathbf{A}_n \in \mathbb{R}^{\mathbf{d}_i \times \mathbf{d}_i}$.

We assume $\mathbf{A}_n^\mathsf{T} \mathbf{A}_n$ has the singular value decomposition $\mathbf{A}_n^\mathsf{T} \mathbf{A}_n = \tilde{\mathbf{U}}^\mathsf{T} \mathrm{diag}(\boldsymbol{\lambda}) \tilde{\mathbf{U}}$, where $\tilde{\mathbf{U}} \in \mathbb{R}^{\mathbf{d}_i \times \mathbf{d}_i}$, $\boldsymbol{\lambda} \in \mathbb{R}^{\mathbf{d}_i \times 1}$, and $\tilde{\mathbf{U}}^\mathsf{T} \tilde{\mathbf{U}} = \tilde{\mathbf{U}} \tilde{\mathbf{U}}^\mathsf{T} = \mathbf{I}_{\mathbf{d}_i}$. Here, $\mathrm{diag}(\boldsymbol{\lambda})$ denotes a diagonal matrix with $\boldsymbol{\lambda}$ as the main diagonal entries.

**Part (a).** We derive:

$$\mathsf{L}_n^t \triangleq L_n + \beta^t \overline{\lambda} \overset{①}{\leq} \beta^t \overline{\lambda} (\delta + 1), \tag{35}$$

where step ① uses Lemma 3.1 that $L_n \leq \delta\beta^t\overline{\lambda}$.

**Part (b).** We have:

$$\|\mathbf{Q}^t\| \overset{①}{=} \|\theta_2\mathsf{L}_n^t - \beta^t\boldsymbol{\lambda}\|_\infty \overset{②}{=} \theta_2\mathsf{L}_n^t - \min(\beta^t\boldsymbol{\lambda}) \overset{③}{\leq} \overline{\lambda}\beta^t \cdot \underbrace{(\theta_2(1+\delta) - \underline{\lambda}'/\overline{\lambda})}_{\triangleq q},$$

where step ① uses $\|\theta_2\mathsf{L}_n^t\mathbf{I} - \beta^t\mathbf{A}_n^\mathsf{T}\mathbf{A}_n\| = \|\tilde{\mathbf{U}}^\mathsf{T}\mathrm{diag}(\theta_2\mathsf{L}_n^t - \beta^t\boldsymbol{\lambda})\tilde{\mathbf{U}}\| = \|\theta_2\mathsf{L}_n^t - \beta^t\boldsymbol{\lambda}\|_\infty$; step ② uses the fact that $\|\rho - \mathbf{x}\|_\infty = \max(\rho - \mathbf{x}) = \rho - \min(\mathbf{x})$ whenever $\rho \geq \max(\mathbf{x})$ for all $\rho$ and $\mathbf{x}$; step ③ uses Inequality (35).

**Part (c).** Given $\mathsf{u}_n^{t+1} \triangleq \mathbf{Q}^t(\mathbf{x}_n^{t+1} - \mathbf{x}_n^t)$ as presented in Lemma 4.2, we have: $\|\mathsf{u}_n^{t+1}\| \leq \|\mathbf{Q}^t\| \cdot \|\mathbf{x}_n^{t+1} - \mathbf{x}_n^t\| \leq q\overline{\lambda}\beta^t\|\mathbf{x}_n^{t+1} - \mathbf{x}_n^t\|$.

$\square$

### D.4 PROOF OF LEMMA 4.4

*Proof.* For any $\sigma \in [1, 2)$, we define $\sigma_1 \triangleq \frac{\sigma}{(1-|1-\sigma|)^2}$, and $\sigma_2 \triangleq \frac{|1-\sigma|}{\sigma(1-|1-\sigma|)}$.

We define $\mathsf{w}_n^{t+1} = \nabla h_n(\mathbf{x}_n^{t+1}; \mu^t) + \nabla f_n(\mathbf{x}_n^t)$.

We define $\mathsf{a}^{t+1} \triangleq \mathbf{A}_n^\mathsf{T}(\mathbf{z}^{t+1} - \mathbf{z}^t)$, and $\mathsf{c}^t \triangleq \mathsf{u}_n^t - \mathsf{u}_n^{t+1} + \mathsf{w}_n^t - \mathsf{w}_n^{t+1}$.

We define $a = \frac{\omega\sigma_2}{\underline{\lambda}}$, and $\mathbb{A}^t \triangleq \frac{1}{\beta^t}\|\mathsf{a}^t\|_2^2$.

We define $b = \frac{3\omega\sigma_1}{\underline{\lambda}}$, and $\mathbb{B}^t \triangleq \frac{1}{\beta^t}(L_n\|\mathbf{x}_n^t - \mathbf{x}_n^{t-1}\| + \|\mathsf{u}_n^t\|)^2$.

We define $\mathbb{U}^t \triangleq \frac{C_h^2 b}{\beta^t} \cdot (\frac{\mu^{t-1}}{\mu^t} - 1)^2$.

First, we bound the term $\|\mathsf{c}^t\|$. For all $t \geq 1$, we have:

$$\|\mathsf{c}^t\| = \|\mathsf{w}_n^t - \mathsf{w}_n^{t+1} + \mathsf{u}_n^t - \mathsf{u}_n^{t+1}\|$$
$$\overset{①}{\leq} \|\nabla h_n(\mathbf{x}_n^{t+1}; \mu^t) - \nabla h_n(\mathbf{x}_n^t; \mu^{t-1})\| + \|\nabla f_n(\mathbf{x}_n^t) - \nabla f_n(\mathbf{x}_n^{t-1})\| + \|\mathsf{u}_n^t - \mathsf{u}_n^{t+1}\|$$
$$\overset{②}{\leq} \|\nabla h_n(\mathbf{x}_n^{t+1}; \mu^t) - \nabla h_n(\mathbf{x}_n^t; \mu^{t-1})\| + L_n\|\mathbf{x}_n^t - \mathbf{x}_n^{t-1}\| + \|\mathsf{u}_n^t - \mathsf{u}_n^{t+1}\|$$
$$= \|\nabla h_n(\mathbf{x}_n^{t+1}; \mu^t) - \nabla h_n(\mathbf{x}_n^t; \mu^t) + \nabla h_n(\mathbf{x}_n^t; \mu^t) - \nabla h_n(\mathbf{x}_n^t; \mu^{t-1})\|$$
$$+ L_n\|\mathbf{x}_n^t - \mathbf{x}_n^{t-1}\| + \|\mathsf{u}_n^t - \mathsf{u}_n^{t+1}\|$$
$$\overset{③}{\leq} \frac{1}{\mu^t}\|\mathbf{x}_n^{t+1} - \mathbf{x}_n^t\| + (\frac{\mu^{t-1}}{\mu^t} - 1)C_h + L_n\|\mathbf{x}_n^t - \mathbf{x}_n^{t-1}\| + \|\mathsf{u}_n^t\| + \|\mathsf{u}_n^{t+1}\|, \tag{36}$$

where step ① uses the triangle inequality; step ② uses the fact that $f_n(\mathbf{x})$ is $L_n$-smooth; step ③ uses Lemma 3.5 and Lemma 3.3.

Second, we bound the term $\frac{\omega\sigma_1}{\underline{\lambda}\beta^t}\|\mathsf{c}^t\|_2^2$. For all $t \geq 1$, we have:

$$\frac{\omega\sigma_1}{\underline{\lambda}\beta^t}\|\mathsf{c}^t\|_2^2$$
$$\overset{①}{\leq} \frac{3\omega\sigma_1}{\underline{\lambda}\beta^t}(\frac{1}{\mu^t}\|\mathbf{x}_n^{t+1} - \mathbf{x}_n^t\| + \|\mathsf{u}_n^{t+1}\|)^2 + \underbrace{\frac{3\omega\sigma_1}{\underline{\lambda}\beta^t}C_h^2(\frac{\mu^{t-1}}{\mu^t} - 1)^2}_{\triangleq \mathbb{U}^t} + \underbrace{\frac{3\omega\sigma_1}{\underline{\lambda}\beta^t}(L_n\|\mathbf{x}_n^t - \mathbf{x}_n^{t-1}\| + \|\mathsf{u}_n^t\|)^2}_{\triangleq b\mathbb{B}^t}$$
$$\overset{②}{=} \frac{3\omega\sigma_1}{\underline{\lambda}\beta^t}\{(\frac{1}{\mu^t}\|\mathbf{x}_n^{t+1} - \mathbf{x}_n^t\| + \|\mathsf{u}_n^{t+1}\|)^2 + (L_n\|\mathbf{x}_n^{t+1} - \mathbf{x}_n^t\| + \|\mathsf{u}_n^{t+1}\|)^2\} + \mathbb{U}^t + b(\mathbb{B}^t - \mathbb{B}^{t+1})$$
$$\overset{③}{\leq} \frac{3\omega\sigma_1}{\underline{\lambda}\beta^t} \cdot 2((\delta + q)\overline{\lambda}\beta^t\|\mathbf{x}_n^{t+1} - \mathbf{x}_n^t\|)^2 + \mathbb{U}^t + b(\mathbb{B}^t - \mathbb{B}^{t+1})$$
$$= \underbrace{6\omega\sigma_1\kappa(\delta + q)^2}_{\triangleq \chi} \cdot \overline{\lambda}\beta^t \cdot \|\mathbf{x}_n^{t+1} - \mathbf{x}_n^t\|_2^2 + \mathbb{U}^t + b(\mathbb{B}^t - \mathbb{B}^{t+1})$$
$$\overset{④}{\leq} \chi\mathsf{L}_n^t\|\mathbf{x}_n^{t+1} - \mathbf{x}_n^t\|_2^2 + \mathbb{U}^t + b(\mathbb{B}^t - \mathbb{B}^{t+1}), \tag{37}$$

where step ① uses Inequality 40 and the fact that $(a+b+c)^2 \leq 3a^2 + 3b^2 + 3c^2$ for all $a, b, c \in \mathbb{R}$; step ② uses the definitions of $\{b, \mathbb{B}^t, \mathbb{U}^t\}$; step ③ uses Lemma 4.3 that: $\frac{1}{\mu^t} \leq \delta \overline{\lambda} \beta^t$, $L_n \leq \delta \overline{\lambda} \beta^t$, and $\|\mathbb{u}_n^{t+1}\| \leq \|\mathbf{Q}^t\| \cdot \|\mathbf{x}_n^{t+1} - \mathbf{x}_n^t\| \leq q \overline{\lambda} \beta^t \|\mathbf{x}_n^{t+1} - \mathbf{x}_n^t\|$; step ④ uses $\beta^t \overline{\lambda} \leq \mathsf{L}_n^t \triangleq \beta^t \overline{\lambda} + L_n$.

Finally, we derive the following inequalities for all $t \geq 1$:

$$
\begin{aligned}
\frac{\omega}{\sigma \beta^t} \|\mathbf{z}^{t+1} - \mathbf{z}^t\|_2^2 \;\overset{①}{\leq}\;& \frac{\omega}{\underline{\lambda} \sigma \beta^t} \|\mathbf{A}_n^{\mathsf{T}}(\mathbf{z}^{t+1} - \mathbf{z}^t)\|_2^2 = \frac{\omega}{\sigma \underline{\lambda} \beta^t} \|\mathbb{a}^t\|_2^2 \\
\overset{②}{\leq}\;& \frac{\sigma_2 \omega}{\underline{\lambda}} \left( \frac{1}{\beta^t} \|\mathbb{a}^t\|_2^2 - \frac{1}{\beta^t} \|\mathbb{a}^{t+1}\|_2^2 \right) + \frac{\omega \sigma_1}{\underline{\lambda} \beta^t} \|\mathbb{c}^t\|_2^2 \\
\overset{③}{\leq}\;& \underbrace{\frac{\sigma_2 \omega}{\underline{\lambda}} \cdot \frac{1}{\beta^t} \|\mathbb{a}^t\|_2^2}_{\triangleq a \mathbb{A}^t} - \frac{\sigma_2 \omega}{\underline{\lambda}} \cdot \frac{1}{\beta^{t+1}} \|\mathbb{a}^{t+1}\|_2^2 + \frac{\omega \sigma_1}{\underline{\lambda}} \cdot \frac{1}{\beta^t} \|\mathbb{c}^t\|_2^2 \\
\overset{④}{\leq}\;& a(\mathbb{A}^t - \mathbb{A}^{t+1}) + \chi \mathsf{L}_n^t \|\mathbf{x}_n^{t+1} - \mathbf{x}_n^t\|_2^2 + \mathbb{U}^t + b(\mathbb{B}^t - \mathbb{B}^{t+1}),
\end{aligned}
$$

where step ① uses $\underline{\lambda} \|\mathbf{z}\|_2^2 \leq \|\mathbf{A}_n^{\mathsf{T}} \mathbf{z}\|_2^2$ for all $\mathbf{z}$; step ② uses Lemma A.2 with $\mathbf{b} = \mathbb{a}^t$, $\mathbf{b}^+ = \mathbb{a}^{t+1}$, and $\mathbf{a} = \mathbb{c}^t$ that:

$$
\frac{1}{\sigma \beta^t} \|\mathbb{a}^{t+1}\|_2^2 \leq \frac{\sigma_2}{\beta^t} (\|\mathbb{a}^t\|_2^2 - \|\mathbb{a}^{t+1}\|_2^2) + \frac{\sigma_1}{\beta^t} \|\mathbb{c}^t\|_2^2;
$$

step ③ uses $-\frac{1}{\beta^t} \leq -\frac{1}{\beta^{t+1}}$; step ④ uses Inequality (37). $\qquad \square$

## D.5 PROOF OF LEMMA 4.5

*Proof.* We define $\varepsilon_1 \triangleq \frac{1}{2} \theta_1 - \frac{1}{2}$, and $\varepsilon_2 \triangleq \theta_2 - \frac{1}{2} - \chi$.

We define $f(\theta_2) \triangleq (\theta_2 - \frac{1}{2}) - \varrho(\delta + \theta_2 + \delta \theta_2 - 1/\kappa)^2$, and $\chi \triangleq \varrho(\delta + \theta_2 + \theta_2 \delta - 1/\kappa)^2$.

We define $\Theta^t \triangleq \Theta_L^t + \Theta_+^t$, where $\Theta_+^t \triangleq a \mathbb{A}^t + b \mathbb{B}^t$.

We define $\mathcal{E}^{t+1} \triangleq [\varepsilon_1 \sum_{i=1}^{n-1} \mathsf{L}_i^t \|\mathbf{x}_i^{t+1} - \mathbf{x}_i^t\|_2^2] + \varepsilon_2 \mathsf{L}_n^t \|\mathbf{x}_n^{t+1} - \mathbf{x}_n^t\|_2^2 + \frac{\varepsilon_3}{\beta^t} \|\mathbf{z}^{t+1} - \mathbf{z}^t\|_2^2$.

**Part (a).** With the choice $\theta_1 = 1.01$, it clearly holds that $\varepsilon_1 \triangleq \frac{1}{2} \theta_1 - \frac{1}{2} > 0$.

Using Lemma A.8 with $\theta = \theta_2$, we have $f(\theta_2) \geq \frac{1}{8\varrho} > 0$, leading to $\varepsilon_2 \triangleq (\theta_2 - \frac{1}{2}) - \chi > 0$.

**Part (b).** Using Lemmas 4.1 and 4.4, we derive the following two respective inequalities:

$$
\mathcal{E}^{t+1} + \Theta_L^{t+1} - \Theta_L^t \;\leq\; (\tfrac{1}{2} - \theta_2 + \varepsilon_2) \cdot \mathsf{L}_n^t \|\mathbf{x}_n^{t+1} - \mathbf{x}_n^t\|_2^2 + \frac{\omega}{\sigma \beta^t} \|\mathbf{z}^{t+1} - \mathbf{z}^t\|_2^2 \tag{38}
$$

$$
\frac{\omega}{\sigma \beta^t} \|\mathbf{z}^{t+1} - \mathbf{z}^t\|_2^2 \;\leq\; \Theta_+^t - \Theta_+^{t+1} + \chi \mathsf{L}_n^t \|\mathbf{x}_n^{t+1} - \mathbf{x}_n^t\|_2^2 + \mathbb{U}^t. \tag{39}
$$

Adding Inequalities (38) and (39) together, we have:

$$
\mathcal{E}^{t+1} + \Theta^{t+1} - \Theta^t - \mathbb{U}^t \leq \mathsf{L}_n^t \|\mathbf{x}_n^{t+1} - \mathbf{x}_n^t\|_2^2 \cdot \{\tfrac{1}{2} - \theta_2 + \varepsilon_2 + \chi\} \overset{①}{=} 0,
$$

where step ① uses the definition of $\varepsilon_2 \triangleq \theta_2 - \frac{1}{2} - \chi$. $\qquad \square$

## D.6 PROOF OF LEMMA 4.6

*Proof.* For any $\sigma \in (0, 1)$, we define $\sigma_1 \triangleq \frac{\sigma}{(1 - |1 - \sigma|)^2}$, and $\sigma_2 \triangleq \frac{|1 - \sigma|}{\sigma(1 - |1 - \sigma|)}$.

We define $\mathbb{w}_n^{t+1} = \nabla h_n(\mathbf{x}_n^{t+1}; \mu^t) + \nabla f_n(\mathbf{x}_n^t)$.

We define $\mathbb{a}^{t+1} \triangleq \mathbf{A}_n^{\mathsf{T}}(\mathbf{z}^{t+1} - \mathbf{z}^t) + \sigma \mathbb{u}_n^t$, and $\mathbb{c}^t \triangleq \sigma \mathbb{u}_n^t + \mathbb{w}_n^t - \mathbb{w}_n^{t+1}$.

We define $a \triangleq \frac{2 \omega \sigma_2}{\underline{\lambda}}$, and $\mathbb{A}^t \triangleq \frac{1}{\beta^t} \|\mathbb{a}^t\|_2^2$.

We define $b \triangleq \frac{6 \omega \sigma_1}{\underline{\lambda}}$, and $\mathbb{B}^t \triangleq \frac{1}{\beta^t} (L_n \|\mathbf{x}_n^t - \mathbf{x}_n^{t-1}\| + \sigma \|\mathbb{u}_n^t\|)^2$.

We define $\mathbb{U}^t \triangleq \frac{C_h^2 b}{\beta^t} \cdot (\frac{\mu^{t-1}}{\mu^t} - 1)^2$.

First, we bound the term $\|\mathbb{c}^t\|$. For all $t \geq 1$, we have:

$$\|\mathbb{c}^t\| = \|\mathbb{w}_n^t - \mathbb{w}_n^{t+1} + \sigma \mathbb{u}_n^t\|$$

$$\overset{①}{\leq} \|\nabla h_n(\mathbf{x}_n^{t+1}; \mu^t) - \nabla h_n(\mathbf{x}_n^t; \mu^{t-1})\| + \|\nabla f_n(\mathbf{x}_n^t) - \nabla f_n(\mathbf{x}_n^{t-1})\| + \sigma \|\mathbb{u}_n^t\|$$

$$\overset{②}{\leq} \|\nabla h_n(\mathbf{x}_n^{t+1}; \mu^t) - \nabla h_n(\mathbf{x}_n^t; \mu^{t-1})\| + L_n \|\mathbf{x}_n^t - \mathbf{x}_n^{t-1}\| + \sigma \|\mathbb{u}_n^t\|$$

$$= \|\nabla h_n(\mathbf{x}_n^{t+1}; \mu^t) - \nabla h_n(\mathbf{x}_n^t; \mu^t) + \nabla h_n(\mathbf{x}_n^t; \mu^t) - \nabla h_n(\mathbf{x}_n^t; \mu^{t-1})\| + L_n \|\mathbf{x}_n^t - \mathbf{x}_n^{t-1}\| + \sigma \|\mathbb{u}_n^t\|$$

$$\overset{③}{\leq} \frac{1}{\mu^t} \|\mathbf{x}_n^{t+1} - \mathbf{x}_n^t\| + (\frac{\mu^{t-1}}{\mu^t} - 1)C_h + L_n \|\mathbf{x}_n^t - \mathbf{x}_n^{t-1}\| + \sigma \|\mathbb{u}_n^t\|, \tag{40}$$

where step ① uses the triangle inequality; step ② uses the fact that $f_n(\mathbf{x})$ is $L_n$-smooth; step ③ uses Lemma 3.3 and Lemma 3.5.

Second, we bound the term $\frac{2\omega\sigma}{\underline{\lambda}\beta^t}\|\mathbb{u}_n^t\|_2^2 + \frac{2\omega}{\sigma\underline{\lambda}\beta^t}\|\mathbb{c}^t\|_2^2$. For all $t \geq 1$, we have:

$$\frac{2\omega\sigma}{\underline{\lambda}\beta^t}\|\mathbb{u}_n^{t+1}\|_2^2 + \frac{2\omega\sigma_1}{\underline{\lambda}\beta^t}\|\mathbb{c}^t\|_2^2$$

$$\overset{①}{\leq} \frac{2\omega\sigma}{\underline{\lambda}\beta^t}\|\mathbb{u}_n^{t+1}\|_2^2 + \frac{6\omega\sigma_1}{\underline{\lambda}\beta^t}(\frac{1}{\mu^t}\|\mathbf{x}_n^{t+1} - \mathbf{x}_n^t\|)^2 + \underbrace{\frac{6\omega\sigma_1}{\underline{\lambda}\beta^t}(\frac{\mu^{t-1}}{\mu^t} - 1)^2 C_h^2}_{\mathbb{U}^t} + \underbrace{\frac{6\omega\sigma_1}{\underline{\lambda}\beta^t}(L_n\|\mathbf{x}_n^t - \mathbf{x}_n^{t-1}\| + \sigma\|\mathbb{u}_n^t\|)^2}_{\triangleq b\mathbb{B}^t}$$

$$\overset{②}{=} \frac{2\omega\sigma_1}{\beta^t\underline{\lambda}} \cdot \{\frac{\sigma}{\sigma_1}\|\mathbb{u}_n^{t+1}\|_2^2 + 3(\frac{1}{\mu^t}\|\mathbf{x}_n^{t+1} - \mathbf{x}_n^t\|)^2 + 3(L_n\|\mathbf{x}_n^{t+1} - \mathbf{x}_n^t\| + \sigma\|\mathbb{u}_n^{t+1}\|)^2\} + \mathbb{U}^t + b(\mathbb{B}^t - \mathbb{B}^{t+1})$$

$$\overset{③}{\leq} \frac{2\omega\sigma_1}{\beta^t\underline{\lambda}} \cdot \overline{\lambda}^2(\beta^t)^2 \cdot \{\frac{\sigma}{\sigma_1}q^2 + 3\delta^2 + 3(\delta + \sigma q)^2\}\|\mathbf{x}_n^{t+1} - \mathbf{x}_n^t\|_2^2 + \mathbb{U}^t + b(\mathbb{B}^t - \mathbb{B}^{t+1})$$

$$\overset{④}{\leq} \underbrace{\frac{2\omega\kappa}{\sigma} \cdot \{\sigma^2 q^2 + 3\delta^2 + 3(\delta + \sigma q)^2\}}_{\triangleq \chi} \cdot \overline{\lambda}\beta^t\|\mathbf{x}_n^{t+1} - \mathbf{x}_n^t\|_2^2 + \mathbb{U}^t + b(\mathbb{B}^t - \mathbb{B}^{t+1})$$

$$\overset{⑤}{\leq} \chi \cdot \mathsf{L}_n^t \|\mathbf{x}_n^{t+1} - \mathbf{x}_n^t\|_2^2 + \mathbb{U}^t + b(\mathbb{B}^t - \mathbb{B}^{t+1}), \tag{41}$$

where step ① uses Inequality (40) and the fact that $(a + b + c)^2 \leq 3a^2 + 3b^2 + 3c^2$ for all $a, b, c \in \mathbb{R}$; step ② uses the definitions of $\{b, \mathbb{B}^t, \mathbb{U}^t\}$; step ③ uses $\|\mathbb{u}_n^{t+1}\| \leq \|\mathbf{Q}^t\|\|\mathbf{x}_n^{t+1} - \mathbf{x}_n^t\| \leq \beta^t\overline{\lambda}q\|\mathbf{x}_n^{t+1} - \mathbf{x}_n^t\|$ and $L_n \leq \overline{\lambda}\beta^t\delta$, as has been shown respectively in Lemma 4.3 and Lemma 3.1, as well as the fact that $\frac{1}{\mu^t} = \beta^t\overline{\lambda}\delta$; step ④ uses $\kappa = \overline{\lambda}/\underline{\lambda}$, and the fact that $\sigma_1 = \frac{1}{\sigma}$ when $\sigma \in (0, 1)$; step ⑤ uses $\beta^t\overline{\lambda} \leq \mathsf{L}_n^t \triangleq \beta^t\overline{\lambda} + L_n$.

Finally, for all $t \geq 1$, we derive:

$$\frac{\omega}{\sigma\beta^t}\|\mathbf{z}^{t+1} - \mathbf{z}^t\|_2^2$$

$$\overset{①}{\leq} \frac{\omega}{\sigma\beta^t \cdot \underline{\lambda}}\|\mathbf{A}_n^\mathsf{T}(\mathbf{z}^{t+1} - \mathbf{z}^t)\|_2^2$$

$$\overset{②}{=} \frac{\omega}{\underline{\lambda}} \cdot \frac{1}{\sigma\beta^t}\|\mathbb{a}^{t+1} - \sigma\mathbb{u}_n^{t+1}\|_2^2$$

$$\overset{③}{\leq} \frac{2\omega}{\underline{\lambda}} \cdot \{\frac{1}{\sigma\beta^t}\|\mathbb{a}^{t+1}\|_2^2 + \frac{\sigma}{\beta^t}\|\mathbb{u}_n^{t+1}\|_2^2\}$$

$$\overset{④}{\leq} \frac{2\omega}{\underline{\lambda}} \cdot \{\frac{\sigma_2}{\beta^t}\|\mathbb{a}^t\|_2^2 - \frac{\sigma_2}{\beta^t}\|\mathbb{a}^{t+1}\|_2^2 + \frac{\sigma_1}{\beta^t}\|\mathbb{c}^t\|_2^2\} + \frac{2\omega\sigma}{\beta^t\underline{\lambda}}\|\mathbb{u}_n^{t+1}\|_2^2$$

$$\overset{⑤}{\leq} \underbrace{\frac{2\omega}{\underline{\lambda}}\frac{\sigma_2}{\beta^t}\|\mathbb{a}^t\|_2^2}_{\triangleq a\mathbb{A}^t} - \underbrace{\frac{2\omega}{\underline{\lambda}}\frac{\sigma_2}{\beta^{t+1}}\|\mathbb{a}^{t+1}\|_2^2}_{\triangleq a\mathbb{A}^{t+1}} + \frac{2\omega}{\underline{\lambda}}\frac{\sigma_1}{\beta^t}\|\mathbb{c}^t\|_2^2 + \frac{2\omega\sigma}{\beta^t\underline{\lambda}}\|\mathbb{u}_n^{t+1}\|_2^2$$

$$\overset{⑥}{\leq} a\mathbb{A}^t - a\mathbb{A}^{t+1} + \chi\mathsf{L}_n^t\|\mathbf{x}_n^{t+1} - \mathbf{x}_n^t\|_2^2 + \mathbb{U}^t + b\mathbb{B}^t - b\mathbb{B}^{t+1},$$

where step ① uses the fact that $\underline{\lambda}\|\mathbf{x}\|_2^2 \leq \|\mathbf{A}_n^\mathsf{T}\mathbf{x}\|_2^2$ for all $\mathbf{x}$; step ② uses the definition of $\mathbb{a}^{t+1}$; step ③ uses the inequality $\|\mathbf{a} + \mathbf{b}\|_2^2 \leq 2\|\mathbf{a}\|_2^2 + 2\|\mathbf{b}\|_2^2$ for all $\mathbf{a}$ and $\mathbf{b}$; step ④ uses Lemma A.2 with $\mathbf{b} = \mathbb{a}^t$, $\mathbf{b}^+ = \mathbb{a}^{t+1}$, and $\mathbf{a} = \mathbb{c}^t$ that

$$\frac{1}{\sigma\beta^t}\|\mathbb{a}^{t+1}\|_2^2 \leq \frac{\sigma_1}{\beta^t}\|\mathbb{c}^t\|_2^2 + \frac{\sigma_2}{\beta^t}(\|\mathbb{a}^t\|_2^2 - \|\mathbb{a}^{t+1}\|_2^2);$$

step ⑤ uses $-\frac{1}{\beta^t} \leq -\frac{1}{\beta^{t+1}}$ and $\sigma_1 = \frac{1}{\sigma}$ when $\sigma \in (0,1)$; step ⑥ uses Inequality (41).

$\square$

## D.7 Proof of Lemma 4.7

*Proof.* We define $\mathcal{E}^{t+1} \triangleq [\varepsilon_1 \sum_{i=1}^{n-1} \mathsf{L}_i^t \|\mathbf{x}_i^{t+1} - \mathbf{x}_i^t\|_2^2] + \varepsilon_2 \mathsf{L}_n^t \|\mathbf{x}_n^{t+1} - \mathbf{x}_n^t\|_2^2 + \frac{\varepsilon_3}{\beta^t} \|\mathbf{z}^{t+1} - \mathbf{z}^t\|_2^2$.

We define $\Theta^t \triangleq \Theta_L^t + \Theta_+^t$, where $\Theta_+^t \triangleq a\mathbb{A}^t + b\mathbb{B}^t$.

**Part (a).** We assume $\xi = \delta = \sigma = \frac{c}{\kappa}$, where $c \in (0, 1)$. We have:

$$\omega \triangleq 1 + \frac{\xi}{\sigma} = 2 \qquad (42)$$

$$q \triangleq \theta_2 + \theta_2 \delta \overset{①}{\leq} \theta_2 + \theta_2 c. \qquad (43)$$

where step ① uses $\delta = c/\kappa \leq c$ since $\kappa \geq 1$. We further obtain:

$$\begin{aligned}
\varepsilon_2 &\triangleq \theta_2 - \tfrac{1}{2} - \tfrac{6\omega\kappa}{\sigma}\{\tfrac{1}{3}\sigma^2 q^2 + (\delta + \sigma q)^2 + \delta^2\} \\
&\overset{①}{\geq} \theta_2 - \tfrac{1}{2} - \tfrac{12}{c}\{\tfrac{1}{3}c^2 q^2 + (c + cq)^2 + c^2\} \\
&= \theta_2 - \tfrac{1}{2} - 12c\{\tfrac{1}{3}q^2 + (1 + q)^2 + 1\} \\
&\overset{②}{\geq} \theta_2 - \tfrac{1}{2} - 12c\{\tfrac{(\theta_2 + \theta_2 c)^2}{3} + (1 + \theta_2 + \theta_2 c)^2 + 1\} \\
&\overset{③}{>} 0.02,
\end{aligned}$$

where step ① uses (42), $\sigma \leq c$, $\delta \leq c$; step ② uses (43); step ③ uses the choice $c = 0.01$ and $\theta_2 = 1.5$.

**Part (b).** Using Lemmas 4.1 and 4.6, we derive the following two respective inequalities:

$$\mathcal{E}^{t+1} + \Theta_L^{t+1} - \Theta_L^t \leq (\tfrac{1}{2} - \theta_2 + \varepsilon_2) \cdot \mathsf{L}_n^t \|\mathbf{x}_n^{t+1} - \mathbf{x}_n^t\|_2^2 + \tfrac{\omega}{\sigma\beta^t} \|\mathbf{z}^{t+1} - \mathbf{z}^t\|_2^2,$$

$$\tfrac{\omega}{\sigma\beta^t} \|\mathbf{z}^{t+1} - \mathbf{z}^t\|_2^2 + \Theta_+^{t+1} - \Theta_+^t \leq \chi \mathsf{L}_n^t \|\mathbf{x}_n^{t+1} - \mathbf{x}_n^t\|_2^2 + \mathbb{U}^t.$$

Adding the two inequalities above together leads to:

$$\mathcal{E}^{t+1} + \Theta^{t+1} - \Theta^t - \mathbb{U}^t \leq \mathsf{L}_n^t \|\mathbf{x}_n^{t+1} - \mathbf{x}_n^t\|_2^2 \cdot \{\tfrac{1}{2} - \theta_2 + \varepsilon_2 + \chi\} \overset{①}{=} 0,$$

where step ① uses the definition of $\varepsilon_2 \triangleq \theta_2 - \tfrac{1}{2} - \chi$.

$\square$

## D.8 Proof of Lemma 4.8

*Proof.* The proof of this lemma closely resembles that of Theorem 6 in (Boţ et al., 2019).

We denote $\underline{\Theta} \triangleq \underline{\Theta}' - \mu^0 C_h^2$, where $\underline{\Theta}'$ is defined in Assumption 1.4

Initially, for all $t \geq 1$, we have:

$$\begin{aligned}
\Theta^t &\overset{①}{\triangleq} \mathcal{L}(\mathbf{x}^t, \mathbf{z}^t; \beta^t, \mu^t) + \tfrac{1}{2}C_h\mu^t + a\mathbb{A}^t + b\mathbb{B}^t \\
&\overset{②}{\geq} \mathcal{L}(\mathbf{x}^t, \mathbf{z}^t; \beta^t, \mu^t) \\
&\overset{③}{=} h_n(\mathbf{x}_n^t; \mu^t) + \{\sum_{i=1}^{n-1} h_i(\mathbf{x}_i^t)\} + \sum_{i=1}^n f_i(\mathbf{x}_i^t) + \langle \mathbf{A}\mathbf{x}^t - \mathbf{b}, \mathbf{z}\rangle + \tfrac{\beta^t}{2}\|\mathbf{A}\mathbf{x}^t - \mathbf{b}\|_2^2 \\
&\overset{④}{\geq} -\mu^0 C_h^2 + \{\sum_{i=1}^n h_i(\mathbf{x}_i^t)\} + \{\sum_{i=1}^n f_i(\mathbf{x}_i^t)\} + \langle \mathbf{A}\mathbf{x}^t - \mathbf{b}, \mathbf{z}\rangle + \tfrac{\beta^t}{2}\|\mathbf{A}\mathbf{x}^t - \mathbf{b}\|_2^2 \\
&\overset{⑤}{\geq} \langle \mathbf{A}\mathbf{x}^t - \mathbf{b}, \mathbf{z}^t\rangle + \underbrace{\underline{\Theta}' - \mu^0 C_h^2}_{\triangleq \underline{\Theta}},
\end{aligned} \qquad (44)$$

where step ① uses the definition of $\Theta^t$; step ② uses $\frac{1}{2}C_h\mu^t + a\mathbb{A}^t + b\mathbb{B}^t \geq 0$; step ③ uses the definition of $\mathcal{L}(\mathbf{x}^t, \mathbf{z}^t; \beta^t, \mu^t)$ in Equation (4); step ④ uses $0 \leq h_n(\mathbf{u}) - h_n(\mathbf{u}; \mu) \leq \mu C_h^2$ as shown in Lemma 3.3, and the fact that $\mu^t \leq \mu^0$; step ⑤ uses Assumption 1.4.

We now conclude the proof of this lemma through contradiction. Suppose that there exists $t_0 \geq 1$ such that $\Theta^{t_0} < \underline{\Theta}$. We derive the following inequalities:

$$
\begin{aligned}
\sum_{t=1}^T (\Theta^t - \underline{\Theta}) &= [\sum_{t=1}^{t_0-1}(\Theta^t - \underline{\Theta})] + [\sum_{t=t_0}^T(\Theta^t - \underline{\Theta})] \\
&\leq [\sum_{t=1}^{t_0-1}(\Theta^t - \underline{\Theta})] + (T+1-t_0)\cdot\max_{t=t_0}^T(\Theta^t - \underline{\Theta}) \\
&\overset{①}{\leq} [\sum_{t=1}^{t_0-1}(\Theta^t - \underline{\Theta})] + (T+1-t_0)\cdot(\Theta^{t_0} - \underline{\Theta}),
\end{aligned}
\tag{45}
$$

where step ① uses $\Theta^t \leq \Theta^{t_0}$ for all $t \geq t_0$. We closely examine Inequality (45). As $t_0$ is finite, the sum $\sum_{t=1}^{t_0-1}(\Theta^t - \underline{\Theta})$ is upper bounded. Considering the negativity of the term $(\Theta^{t_0} - \underline{\Theta})$, we deduce from Inequality (45):

$$
\lim_{T\to\infty} \sum_{t=1}^T(\Theta^t - \underline{\Theta}) = -\infty.
\tag{46}
$$

Meanwhile, for all $t \geq 1$, the following inequalities hold:

$$
\begin{aligned}
\Theta^t - \underline{\Theta} &\overset{①}{\geq} \frac{1}{\sigma\beta^{t-1}}\langle \mathbf{z}^t - \mathbf{z}^{t-1}, \mathbf{z}^t\rangle \\
&\overset{②}{=} \frac{1}{2\sigma}\{\frac{1}{\beta^{t-1}}\|\mathbf{z}^t\|_2^2 - \frac{1}{\beta^{t-1}}\|\mathbf{z}^{t-1}\|_2^2 + \frac{1}{\beta^{t-1}}\|\mathbf{z}^t - \mathbf{z}^{t-1}\|_2^2\} \\
&\overset{③}{\geq} \frac{1}{2\sigma}\{\frac{1}{\beta^t}\|\mathbf{z}^t\|_2^2 - \frac{1}{\beta^{t-1}}\|\mathbf{z}^{t-1}\|_2^2 + 0\},
\end{aligned}
\tag{47}
$$

where step ① uses Inequality (44) and $\mathbf{z}^{t+1} = \mathbf{z}^t + \sigma\beta^t(\mathbf{A}\mathbf{x}^{t+1} - \mathbf{b})$; step ② uses the Pythagoras relation in Lemma A.1; step ③ uses $\frac{1}{\beta^{t-1}} \geq \frac{1}{\beta^t}$.

Telescoping Inequality (47) over $t$ from $1$ to $T$, we have:

$$
\sum_{t=1}^T(\Theta^t - \underline{\Theta}) \geq \frac{1}{2\sigma}\cdot\{\frac{1}{\beta^T}\|\mathbf{z}^T\|_2^2 - \frac{1}{\beta^0}\|\mathbf{z}^0\|_2^2\} \geq -\frac{1}{2\sigma\beta^0}\|\mathbf{z}^0\|_2^2.
\tag{48}
$$

The finiteness of the right-hand-side in (48) contradicts with (46).

Therefore, we conclude that $\Theta^t \geq \underline{\Theta}$ for all $t \geq 1$.

$\square$

### D.9 PROOF OF LEMMA 4.9

*Proof.* We define $\overline{\mathbb{U}} \triangleq 3C_h^2\frac{b}{\beta^0}$.

We define $\mathbb{U}^t \triangleq C_h^2\frac{b}{\beta^t}\cdot(\frac{\mu^{t-1}}{\mu^t} - 1)^2$, where $\beta^t = \beta^0(1 + \xi t^p)$, $\mu^t \propto \frac{1}{\beta^t}$.

**Part (a).** Letting $T \in [1, \infty)$, we obtain:

$$
\begin{aligned}
\sum_{t=1}^T(\frac{\mu^{t-1}}{\mu^t} - 1)^2 &\overset{①}{=} \sum_{t=1}^T(\frac{\beta^t}{\beta^{t-1}} - 1)^2 = (\frac{\beta^1}{\beta^0} - 1)^2 + \sum_{t=2}^T(\frac{\beta^t}{\beta^{t-1}} - 1)^2 \\
&\overset{②}{=} (1 + \xi 1^p - 1)^2 + \sum_{t=1}^{T-1}(\frac{\beta^{t+1}}{\beta^t} - 1)^2 \\
&\overset{③}{\leq} 1 + \sum_{t=1}^{\infty}\frac{(\xi(t+1)^p - \xi t^p)^2}{(1 + \xi t^p)^2} \\
&\overset{④}{\leq} 1 + \sum_{t=1}^{\infty}(\frac{(t+1)^p - t^p}{t^p})^2 \\
&\overset{⑤}{\leq} 1 + 2,
\end{aligned}
\tag{49}
$$

where step ① uses $\mu^t \propto \frac{1}{\beta^t}$; step ② uses $\beta^1 = \beta^0(1 + \xi 1^p)$; step ③ uses the definition of $\beta^t = \beta^0 + \beta^0\xi t^p$; step ④ uses $\frac{1}{(1+\xi t^p)^2} \leq \frac{1}{(\xi t^p)^2}$; step ⑤ uses Lemma A.5.

We further obtain:

$$
\sum_{t=1}^{\infty}\mathbb{U}^t \overset{①}{\leq} C_h^2\frac{b}{\beta^0}\cdot\{\sum_{t=1}^{\infty}(\frac{\mu^{t-1}}{\mu^t} - 1)^2\} \overset{②}{\leq} 3C_h^2\frac{b}{\beta^0} \triangleq \overline{\mathbb{U}},
$$

where step ① uses $\beta^t \geq \beta^0$; step ② uses Inequality (49).

**Part (b).** For both conditions $\mathbb{BI}$ and $\mathbb{SU}$, we have from Lemmas (4.5) and (4.7):

$$\mathcal{E}^{t+1} \leq \Theta^t - \Theta^{t+1} + \mathbb{U}^t.$$

Telescoping this inequality over $t$ from 1 to $T$, we have:

$$\sum_{t=1}^T \mathcal{E}^{t+1} \leq \Theta^1 - \Theta^{T+1} + \sum_{t=1}^T \mathbb{U}^t \overset{①}{\leq} \Theta^1 - \underline{\Theta} + \overline{\mathbb{U}} \triangleq \overline{\mathcal{E}}, \tag{50}$$

where step ① uses Lemma 4.8 that $\Theta^t \geq \underline{\Theta}$ for all $t$, and Lemma 4.9.

$\square$

### D.10 PROOF OF LEMMA 4.10

*Proof.* Given $\sigma \in (0, 2)$, we define $\sigma_3 \triangleq \frac{\sigma}{1-|1-\sigma|} \in [1, \infty)$.

We define $\mathbb{w}_n^{t+1} \triangleq \nabla h_n(\mathbf{x}_n^{t+1}, \mu^t) + \nabla f_n(\mathbf{x}_n^t)$.

We define $\mathbb{u}_n^{t+1} \triangleq \mathbf{Q}^t(\mathbf{x}_n^{t+1} - \mathbf{x}_n^t)$, where $\mathbf{Q}^t \triangleq \theta_2 \mathsf{L}_n^t \mathbf{I} - \beta^t \mathbf{A}_n^\mathsf{T} \mathbf{A}_n$.

We define $Z \triangleq \frac{3}{\underline{\lambda}} \cdot (\frac{1}{\beta^0} \overline{\lambda} \|\mathbf{z}^1\|_2^2 + 2\sigma_3 C_h^2 + 2\sigma_3 C_f^2 + \sigma_3 q^2 \overline{\lambda} \frac{\overline{\mathcal{E}}}{\varepsilon_2})$. We define $\ddot{Z} \triangleq \overline{\mathcal{E}}/\varepsilon_3$.

First, we have:

$$\begin{aligned}
\max_{i=1}^\infty \{\|\mathbb{w}_n^{i+1}\|_2^2\} &= \max_{i=1}^\infty \{\|\nabla h_n(\mathbf{x}_n^{i+1}, \mu^i) + \nabla f_n(\mathbf{x}_n^i)\|_2^2\} \\
&\overset{①}{\leq} 2C_h^2 + 2C_f^2
\end{aligned} \tag{51}$$

where step ① uses $\|\mathbf{a} + \mathbf{b}\|_2^2 \leq 2\|\mathbf{a}\|_2^2 + 2\|\mathbf{b}\|_2^2$, and Assumption 1.2.

Second, we have:

$$\begin{aligned}
\max_{i=1}^\infty \{\frac{1}{\beta^i} \|\mathbb{u}_n^{i+1}\|_2^2\} &= \max_{i=1}^\infty \{\frac{1}{\beta^i} \|\mathbf{Q}^i(\mathbf{x}_n^{i+1} - \mathbf{x}_n^i)\|_2^2\} \\
&\overset{①}{\leq} \max_{i=1}^\infty \{\frac{1}{\beta^i} (q\overline{\lambda}\beta^i)^2 \|\mathbf{x}_n^{i+1} - \mathbf{x}_n^i\|_2^2\} \\
&\overset{②}{\leq} q^2\overline{\lambda} \cdot \sum_{i=1}^\infty \{\mathsf{L}_n^i \|\mathbf{x}_n^{i+1} - \mathbf{x}_n^i\|_2^2\} \\
&\overset{③}{\leq} q^2\overline{\lambda} \cdot \frac{\overline{\mathcal{E}}}{\varepsilon_2},
\end{aligned} \tag{52}$$

where step ① uses $\|\mathbf{Q}^t\| \leq \beta^t \overline{\lambda} q$ for all $t \geq 0$, as shown in Lemma 4.3(b); step ② uses $\beta^i \overline{\lambda} \leq \mathsf{L}_n^i \triangleq \beta^i \overline{\lambda} + L_n$; step ③ uses $\sum_{t=1}^\infty \varepsilon_2 \mathsf{L}_n^t \|\mathbf{x}_n^{t+1} - \mathbf{x}_n^t\|_2^2 \leq \sum_{t=1}^\infty \mathcal{E}^{t+1} \leq \overline{\mathcal{E}}$.

**Part (a).** Using Lemma 4.2(b), we have:

$$\mathbf{A}_n^\mathsf{T} \mathbf{z}^{t+1} = (1-\sigma) \cdot \mathbf{A}_n^\mathsf{T} \mathbf{z}^t - \sigma\{\mathbb{w}_n^{t+1} + \mathbb{u}_n^{t+1}\}.$$

For all $t \geq 1$, we have:

$$\|\mathbf{A}_n^\mathsf{T} \mathbf{z}^{t+1}\| \leq |1-\sigma| \cdot \|\mathbf{A}_n^\mathsf{T} \mathbf{z}^t\| + \sigma\{\|\mathbb{w}_n^{t+1}\| + \|\mathbb{u}_n^{t+1}\|\}.$$

Applying Lemma A.7 with $e^t \triangleq \|\mathbf{A}_n^\mathsf{T} \mathbf{z}^t\|$ and $p^t \triangleq \|\mathbb{w}_n^{t+1}\| + \|\mathbb{u}_n^{t+1}\|$, for all $t \geq 1$, we obtain:

$$\begin{aligned}
\|\mathbf{A}_n^\mathsf{T} \mathbf{z}^t\|_2^2 &\leq (\|\mathbf{A}_n^\mathsf{T} \mathbf{z}^1\| + \sigma_3 \max_{i=1}^{t-1}\{\|\mathbb{w}_n^{i+1}\| + \|\mathbb{u}_n^{i+1}\|\})^2 \\
&\overset{①}{\leq} 3 \cdot \{\overline{\lambda}\|\mathbf{z}^1\|_2^2 + \sigma_3 \max_{i=1}^{t-1} \|\mathbb{w}_n^{i+1}\|_2^2 + \sigma_3 \max_{i=1}^{t-1} \|\mathbb{u}_n^{i+1}\|_2^2\} \\
&= \beta^t \cdot 3 \cdot \{\overline{\lambda}\frac{1}{\beta^t}\|\mathbf{z}^1\|_2^2 + \sigma_3 \max_{i=1}^{t-1} \frac{1}{\beta^t} \|\mathbb{w}_n^{i+1}\|_2^2 + \sigma_3 \max_{i=1}^{t-1} \frac{1}{\beta^t} \|\mathbb{u}_n^{i+1}\|_2^2\} \\
&\overset{②}{\leq} \beta^t \cdot 3 \cdot \{\frac{1}{\beta^0}\overline{\lambda}\|\mathbf{z}^1\|_2^2 + \sigma_3 \max_{i=1}^\infty \frac{1}{\beta^i} \|\mathbb{w}_n^{i+1}\|_2^2 + \sigma_3 \max_{i=1}^\infty \frac{1}{\beta^i} \|\mathbb{u}_n^{i+1}\|_2^2\} \\
&\overset{③}{\leq} \beta^t \cdot \underbrace{3 \cdot \{\frac{1}{\beta^0}\overline{\lambda}\|\mathbf{z}^1\|_2^2 + 2\sigma_3 C_h^2 + 2\sigma_3 C_f^2 + \sigma_3 q^2 \overline{\lambda}\frac{\overline{\mathcal{E}}}{\varepsilon_2}\}}_{\triangleq Z\underline{\lambda}},
\end{aligned}$$

where step ① use $(a + b + c)^2 \leq 3(a^2 + b^2 + c^2)$, Assumption 1.3 that $\|\mathbf{A}_n\|_2^2 \leq \overline{\lambda}$; step ② uses $\beta^i \leq \beta^t$ for all $i \leq t$; ③ uses Inequalities (51) and (52). This further leads to

$$\|\mathbf{z}^t\|_2^2 \leq \tfrac{1}{\underline{\lambda}}\|\mathbf{A}_n^\mathsf{T}\mathbf{z}^t\|_2^2 = \tfrac{1}{\underline{\lambda}} \cdot \underline{\lambda} Z\beta^t.$$

**Part (b).** We have:

$$\tfrac{1}{\varepsilon_3}\sum_{t=1}^\infty \tfrac{\varepsilon_3}{\beta^t}\|\mathbf{z}^{t+1} - \mathbf{z}^t\|_2^2 \overset{②}{\leq} \tfrac{1}{\varepsilon_3}\sum_{t=1}^\infty \mathcal{E}^{t+1} \overset{①}{\leq} \overline{\mathcal{E}}/\varepsilon_3 \triangleq \ddot{Z},$$

where step ① uses the definition of $\mathcal{E}^{t+1} \triangleq [\varepsilon_1 \sum_{i=1}^{n-1} \mathsf{L}_i^t\|\mathbf{x}_i^{t+1} - \mathbf{x}_i^t\|_2^2] + \varepsilon_2 \mathsf{L}_n^t\|\mathbf{x}_n^{t+1} - \mathbf{x}_n^t\|_2^2 + \tfrac{\varepsilon_3}{\beta^t}\|\mathbf{z}^{t+1} - \mathbf{z}^t\|_2^2$ in Lemma 4.1; step ② uses Lemma 4.9(b).

$\square$

### D.11 PROOF OF LEMMA 4.11

*Proof.* We let $\sigma \in (0, 2)$.

First, we derive the following inequalities:

$$
\begin{aligned}
\langle \mathbf{A}\mathbf{x}^{t+1} - \mathbf{b}, \mathbf{z}^{t+1} \rangle &= \tfrac{1}{\sigma\beta^t}\langle \mathbf{z}^{t+1} - \mathbf{z}^t, \mathbf{z}^{t+1} \rangle \\
&\overset{①}{=} \tfrac{1}{2\sigma}\{\tfrac{1}{\beta^t}\|\mathbf{z}^{t+1}\|_2^2 - \tfrac{1}{\beta^t}\|\mathbf{z}^t\|_2^2 + \tfrac{1}{\beta^t}\|\mathbf{z}^{t+1} - \mathbf{z}^t\|_2^2\} \\
&\geq -\tfrac{1}{2\sigma\beta^t}\|\mathbf{z}^t\|_2^2,
\end{aligned}
\tag{53}
$$

where step ① uses the Pythagoras relation in Fact A.1.

In view of Lemmas 4.5 and 4.7, given $\mathcal{E}^{i+1} \geq 0$ for all $i \geq 1$, we have:

$$0 \leq \Theta^i - \Theta^{i+1} + \mathbb{U}^i.$$

Telescoping this inequality over $i$ from 1 to $t$, we have:

$$0 \leq \Theta^1 - \Theta^{t+1} + \sum_{i=1}^t \mathbb{U}^i \overset{①}{\leq} \Theta^1 - \Theta^{t+1} + \overline{\mathbb{U}},$$

where step ① uses Lemma 4.9(b). For all $t \geq 1$, we derive the following results:

$$
\begin{aligned}
\Theta^1 + \overline{\mathbb{U}} &\geq \Theta^{t+1} \\
&\overset{①}{=} \Theta_L^{t+1} + a\mathbb{A}^{t+1} + b\mathbb{B}^{t+1} \\
&\overset{②}{=} \mathcal{L}(\mathbf{x}^{t+1}, \mathbf{z}^{t+1}; \beta^{t+1}, \mu^{t+1}) + \tfrac{1}{2}C_h\mu^{t+1} + a\mathbb{A}^{t+1} + b\mathbb{B}^{t+1} \\
&\overset{③}{=} \sum_{i=1}^n f_i(\mathbf{x}_i^{t+1}) + \langle \mathbf{A}\mathbf{x}^{t+1} - \mathbf{b}, \mathbf{z}^{t+1} \rangle + \tfrac{\beta^{t+1}}{2}\|\mathbf{A}\mathbf{x}^{t+1} - \mathbf{b}\|_2^2 \\
&\quad + \{\sum_{i=1}^{n-1} h_i(\mathbf{x}_i^{t+1})\} + h_n(\mathbf{x}_n^{t+1}; \mu^{t+1}) + \tfrac{1}{2}C_h\mu^{t+1} + a\mathbb{A}^{t+1} + b\mathbb{B}^{t+1} \\
&\overset{④}{\geq} \sum_{i=1}^n [f_i(\mathbf{x}_i^{t+1}) + h_i(\mathbf{x}_i^{t+1})] + \langle \mathbf{A}\mathbf{x}^{t+1} - \mathbf{b}, \mathbf{z}^{t+1} \rangle - \tfrac{1}{2}\mu^{t+1}C_h^2 \\
&\overset{⑤}{\geq} \sum_{i=1}^n [f_i(\mathbf{x}_i^{t+1}) + h_i(\mathbf{x}_i^{t+1})] - \tfrac{1}{2\sigma\beta^t}\|\mathbf{z}^t\|_2^2 - \tfrac{1}{2}\mu^{t+1}C_h^2 \\
&\overset{⑥}{\geq} \sum_{i=1}^n [f_i(\mathbf{x}_i^{t+1}) + h_i(\mathbf{x}_i^{t+1})] - \tfrac{1}{2\sigma\beta^t}\|\mathbf{z}^t\|_2^2 - \tfrac{1}{2}\mu^0 C_h^2,
\end{aligned}
$$

where step ① uses the definition of $\Theta^{t+1}$; step ② uses uses the definition of $\Theta_L^{t+1}$ in Lemma 4.1; step ③ uses the definition of $\mathcal{L}(\mathbf{x}^{t+1}, \mathbf{z}^{t+1}; \beta^{t+1}, \mu^{t+1})$ in (4); step ④ uses $\tfrac{\beta^{t+1}}{2}\|\mathbf{A}\mathbf{x}^{t+1} - \mathbf{b}\|_2^2 \geq 0$, $\tfrac{1}{2}C_h\mu^{t+1} + a\mathbb{A}^{t+1} + b\mathbb{B}^{t+1} \geq 0$, and the fact that $h_n(\mathbf{x}_n^{t+1}; \mu^{t+1}) \geq h_n(\mathbf{x}_n^{t+1}) - \tfrac{1}{2}\mu^{t+1}C_h^2$; step ⑤ uses Inequality (53); step ⑥ uses $\mu^t \leq \mu^0$ for all $t$.

We further obtain:

$$
\begin{aligned}
\sum_{i=1}^n [f_i(\mathbf{x}_i^{t+1}) + h_i(\mathbf{x}_i^{t+1})] &\leq \Theta^1 + \overline{\mathbb{U}} + \tfrac{1}{2\sigma\beta^t}\|\mathbf{z}^t\|_2^2 + \tfrac{1}{2}\mu^0 C_h^2 \\
&\overset{①}{<} +\infty,
\end{aligned}
$$

where step ① uses the boundedness of $\tfrac{1}{\beta^t}\|\mathbf{z}^t\|_2^2$ for all $t \geq 0$, as shown in Lemma 4.10. According to Assumption 1.4, we have $\|\mathbf{x}_i^{t+1}\| < +\infty$ for all $i \in [n]$.

$\square$

## D.12 Proof of Theorem 4.12

*Proof.* We define $K \triangleq \frac{\overline{\mathcal{E}}}{K'}$, where $K' \triangleq \min\{\min(\varepsilon_1, \varepsilon_2)\underline{A}^2, \varepsilon_3\}$, and $\underline{A} \triangleq \min_{i=1}^n \|\mathbf{A}_i\|$.

We define $\mathcal{E}^{t+1} \triangleq [\varepsilon_1 \sum_{i=1}^{n-1} \mathsf{L}_i^t \|\mathbf{x}_i^{t+1} - \mathbf{x}_i^t\|_2^2] + \varepsilon_2 \mathsf{L}_n^t \|\mathbf{x}_n^{t+1} - \mathbf{x}_n^t\|_2^2 + \frac{\varepsilon_3}{\beta^t} \|\mathbf{z}^{t+1} - \mathbf{z}^t\|_2^2$.

**Part (a).** We have:

$$
\begin{aligned}
\overline{\mathcal{E}} \;&\overset{①}{\geq}\; \sum_{t=1}^T \mathcal{E}^{t+1} \\
&\overset{②}{=}\; \sum_{t=1}^T \{\varepsilon_1 \sum_{i=1}^{n-1} \mathsf{L}_i^t \|\mathbf{x}_i^{t+1} - \mathbf{x}_i^t\|_2^2] + \varepsilon_2 \mathsf{L}_n^t \|\mathbf{x}_n^{t+1} - \mathbf{x}_n^t\|_2^2 + \frac{\varepsilon_3}{\beta^t} \|\mathbf{z}^{t+1} - \mathbf{z}^t\|_2^2\} \\
&\overset{③}{\geq}\; \frac{1}{\beta^T} \sum_{t=1}^T \{[\varepsilon_1 \sum_{i=1}^{n-1} \frac{\mathsf{L}_i^t}{\beta^t} \|\beta^t(\mathbf{x}_i^{t+1} - \mathbf{x}_i^t)\|_2^2] + \varepsilon_2 \frac{\mathsf{L}_n^t}{\beta^t} \|\beta^t(\mathbf{x}_n^{t+1} - \mathbf{x}_n^t)\|_2^2 + \varepsilon_3 \|\mathbf{z}^{t+1} - \mathbf{z}^t\|_2^2\} \\
&\overset{④}{\geq}\; \frac{1}{\beta^T} \sum_{t=1}^T \{[\varepsilon_1 \sum_{i=1}^{n-1} \underline{A}^2 \|\beta^t(\mathbf{x}_i^{t+1} - \mathbf{x}_i^t)\|_2^2] + \varepsilon_2 \underline{A}^2 \|\beta^t(\mathbf{x}_n^{t+1} - \mathbf{x}_n^t)\|_2^2 + \varepsilon_3 \|\mathbf{z}^{t+1} - \mathbf{z}^t\|_2^2\} \\
&\overset{⑤}{\geq}\; \frac{1}{\beta^T} \cdot K' \cdot \sum_{t=1}^T \{\sum_{i=1}^n \|\beta^t(\mathbf{x}_i^{t+1} - \mathbf{x}_i^t)\|_2^2 + \|\mathbf{z}^{t+1} - \mathbf{z}^t\|_2^2\} \\
&\overset{⑥}{=}\; \frac{1}{\beta^T} \cdot K' \cdot \sum_{t=1}^T \{\|\beta^t(\mathbf{x}^{t+1} - \mathbf{x}^t)\|_2^2 + \|\mathbf{z}^{t+1} - \mathbf{z}^t\|_2^2\},
\end{aligned}
$$

where step ① uses Lemma (4.9)(b); step ② uses the definition of $\mathcal{E}^{t+1}$; step ③ uses $\beta^T \geq \beta^t$ for all $t \leq T$; step ④ uses $\frac{\mathsf{L}_i^t}{\beta^t} = \frac{L_i + \beta^t \|\mathbf{A}_i\|_2^2}{\beta^t} \geq \|\mathbf{A}_i\|_2^2 \geq \underline{A}^2$; step ⑤ uses the definition of $K' \triangleq \min\{\min(\varepsilon_1, \varepsilon_2)\underline{A}^2, \varepsilon_3\}$; step ⑥ uses $\sum_{i=1}^n \|\mathbf{x}_i^{t+1} - \mathbf{x}_i^t\|_2^2 = \|\mathbf{x}^{t+1} - \mathbf{x}^t\|_2^2$. Therefore, we obtain:

$$
\sum_{t=1}^T \{\|\beta^t(\mathbf{x}^{t+1} - \mathbf{x}^t)\|_2^2 + \|\mathbf{z}^{t+1} - \mathbf{z}^t\|_2^2\} \leq \frac{\overline{\mathcal{E}}}{K'} \beta^T = K\beta^T.
$$

**Part (b).** By dividing both sides of the above inequality by $T$, we obtain:

$$
\begin{aligned}
\frac{K\beta^T}{T} \;&\geq\; \frac{1}{T} \sum_{t=1}^T \{\|\beta^t(\mathbf{x}^{t+1} - \mathbf{x}^t)\|_2^2 + \|\mathbf{z}^{t+1} - \mathbf{z}^t\|_2^2\} \\
&\geq\; \min_{t=1}^T \{\|\beta^t(\mathbf{x}^{t+1} - \mathbf{x}^t)\|_2^2 + \|\mathbf{z}^{t+1} - \mathbf{z}^t\|_2^2\}.
\end{aligned}
$$

We conclude that there exists an index $\bar{t}$ with $\bar{t} \leq T$ such that $\|\mathbf{z}^{\bar{t}+1} - \mathbf{z}^{\bar{t}}\|_2^2 + \|\beta^{\bar{t}}(\mathbf{x}^{\bar{t}+1} - \mathbf{x}^{\bar{t}})\|_2^2 \leq \frac{K\beta^T}{T}$. $\qquad\square$

## D.13 Proof of Theorem 4.15

To prove this theorem, we first provide the following lemma.

**Lemma D.1.** *We define* $\mathbf{q}^t \triangleq \{\mathbf{x}_1^t, \mathbf{x}_2^t, \ldots, \mathbf{x}_{n-1}^t, \breve{\mathbf{x}}_n^t\}$. *We have:*

*(a)* $\|\mathbf{A}\mathbf{q}^{t+1} - \mathbf{b}\| \leq c_1 \|\mathbf{z}^{t+1} - \mathbf{z}^t\| + c_2 (\beta^t)^{-1}$.
*(b)* $\mathrm{dist}(\mathbf{0}, \partial h_n(\breve{\mathbf{x}}_n^{t+1}) + \nabla_{\mathbf{x}_n} f_n(\breve{\mathbf{x}}_n^{t+1}) + \mathbf{A}_n^\mathsf{T} \mathbf{z}^{t+1}) \leq c_3 \|\mathbf{z}^{t+1} - \mathbf{z}^t\| + c_4 \|\beta^t(\mathbf{x}^{t+1} - \mathbf{x}^t)\| + c_5 (\beta^t)^{-1}$.
*(c)* $\sum_{i=1}^{n-1} \mathrm{dist}(\mathbf{0}, \partial h_i(\mathbf{x}_i^{t+1}) + \nabla_{\mathbf{x}_i} f_i(\mathbf{x}_i^{t+1}) + \mathbf{A}_i^\mathsf{T} \mathbf{z}^{t+1}) \leq c_6 \|\mathbf{z}^{t+1} - \mathbf{z}^t\| + c_7 \|\beta^t(\mathbf{x}^{t+1} - \mathbf{x}^t)\|$.

*Here,* $c_1 = \frac{1}{\sigma \beta^0}$, $c_2 = \overline{A} \frac{C_h}{\delta\overline{\lambda}}$, $c_3 = (1 - \frac{1}{\sigma})\overline{A}$, $c_4 = q\overline{\lambda} + \frac{L_n}{\beta^0}$, $c_5 = \frac{L_n C_h}{\delta\overline{\lambda}}$, $c_6 = (1 - \frac{1}{\sigma})\overline{A}(n-1)$, *and* $c_7 = \frac{\overline{L}\sqrt{n-1}}{\beta^0} + \theta_1(\frac{\overline{L}}{\beta^0} + \overline{A}^2) + \overline{A}^2(n-1)$. *Furthermore,* $\overline{A} \triangleq \max_{i=1}^n \|\mathbf{A}_i\|$, *and* $\overline{L} \triangleq \max_{i=1}^n L_i$.

*Proof.* We define $\overline{A} \triangleq \max_{i=1}^n \|\mathbf{A}_i\|$, and $\overline{L} \triangleq \max_{i=1}^n L_i$.

We define $\mathsf{u}_i^{t+1} = \theta_1 \mathsf{L}_i^t (\mathbf{x}_i^{t+1} - \mathbf{x}_i^t) - \beta^t \mathbf{A}_i^\mathsf{T} [\sum_{j=i}^n \mathbf{A}_j(\mathbf{x}_j^{t+1} - \mathbf{x}_j^t)]$ with $i \in [n-1]$.

We define $\mathsf{u}_n^{t+1} \triangleq \mathbf{Q}^t(\mathbf{x}_n^{t+1} - \mathbf{x}_n^t)$ with $\mathbf{Q}^t \triangleq \theta_2 \mathsf{L}_n^t \mathbf{I} - \beta^t \mathbf{A}_n^\mathsf{T} \mathbf{A}_n$.

**Part (a).** We have:

$$
\begin{aligned}
\|\mathbf{A}\mathbf{q}^{t+1} - \mathbf{b}\| &= \|[\textstyle\sum_{i=1}^{n}\mathbf{A}_i\mathbf{x}_i^{t+1}] - \mathbf{A}_n\mathbf{x}_n^{t+1} + \mathbf{A}_n\breve{\mathbf{x}}_n^{t+1} - \mathbf{b}\| \\
&\leq \|\textstyle\sum_{i=1}^{n}\mathbf{A}_i\mathbf{x}_i^{t+1} - \mathbf{b}\| + \|\mathbf{A}_n(\mathbf{x}_n^{t+1} - \breve{\mathbf{x}}_n^{t+1})\| \\
&\overset{①}{\leq} \|\mathbf{A}\mathbf{x}^{t+1} - \mathbf{b}\| + \overline{\mathrm{A}}\mu^t C_h \\
&\overset{②}{=} \|\tfrac{1}{\sigma\beta^t}(\mathbf{z}^{t+1} - \mathbf{z}^t)\| + \overline{\mathrm{A}}\tfrac{C_h}{\delta\overline{\lambda}\beta^t}, \\
&\overset{③}{\leq} \underbrace{\tfrac{1}{\sigma\beta^0}}_{\triangleq c_1}\|\mathbf{z}^{t+1} - \mathbf{z}^t\| + \underbrace{\overline{\mathrm{A}}(\tfrac{C_h}{\delta\overline{\lambda}})}_{\triangleq c_2}\cdot(\beta^t)^{-1},
\end{aligned}
$$

where step ① uses $\|\mathbf{A}_n\| \leq \overline{\mathrm{A}}$ and Lemma 3.6(c); step ② uses $\mathbf{z}^{t+1} = \mathbf{z}^t + \beta^t\sigma(\mathbf{A}\mathbf{x}^{t+1} - \mathbf{b})$, and $\mu^t = \frac{1}{\delta\overline{\lambda}\beta^t}$; step ③ uses $\beta^0 \leq \beta^t$.

**Part (b).** We first have the following inequalities:

$$
\begin{aligned}
\|\nabla f_n(\mathbf{x}_n^t) - \nabla f_n(\breve{\mathbf{x}}_n^{t+1})\| &= \|\nabla f_n(\mathbf{x}_n^t) - \nabla f_n(\mathbf{x}_n^{t+1}) + \nabla f_n(\mathbf{x}_n^{t+1}) - \nabla f_n(\breve{\mathbf{x}}_n^{t+1})\| \\
&\leq \|\nabla f_n(\mathbf{x}_n^t) - \nabla f_n(\mathbf{x}_n^{t+1})\| + \|\nabla f_n(\mathbf{x}_n^{t+1}) - \nabla f_n(\breve{\mathbf{x}}_n^{t+1})\| \\
&\overset{①}{\leq} L_n\|\mathbf{x}_n^{t+1} - \mathbf{x}_n^t\| + L_n\|\breve{\mathbf{x}}_n^{t+1} - \mathbf{x}_n^{t+1}\| \\
&\overset{②}{\leq} L_n\|\mathbf{x}_n^{t+1} - \mathbf{x}_n^t\| + L_n\mu^t C_h \\
&\overset{③}{\leq} L_n\|\mathbf{x}_n^{t+1} - \mathbf{x}_n^t\| + \underbrace{L_n\tfrac{1}{(\delta\overline{\lambda})}C_h}_{\triangleq c_5}\cdot\tfrac{1}{\beta^t},
\end{aligned}\tag{54}
$$

where step ① uses the fact that $f_n(\mathbf{x}_n)$ is $L_n$-smooth; step ② uses Lemma 3.6(c) that: $\|\breve{\mathbf{x}}_n^{t+1} - \mathbf{x}_n^{t+1}\| \leq \mu^t C_h$; step ③ uses $\mu^t = \frac{1}{\delta\overline{\lambda}\beta^t}$.

We further obtain:

$$
\begin{aligned}
&\mathrm{dist}(\mathbf{0}, \partial h_n(\breve{\mathbf{x}}_n^{t+1}) + \nabla f_i(\breve{\mathbf{x}}_i^{t+1}) + \mathbf{A}_i^\mathsf{T}\mathbf{z}^{t+1}) \\
\overset{①}{=}& \|\theta_2\mathsf{L}_n^t(\mathbf{c}^t - \mathbf{x}_n^{t+1}) + \nabla f_i(\breve{\mathbf{x}}_i^{t+1}) + \mathbf{A}_i^\mathsf{T}\mathbf{z}^{t+1}\| \\
\overset{②}{=}& \|\theta_2\mathsf{L}_n^t(\mathbf{x}_n^t - \tfrac{1}{\theta_2\mathsf{L}_n^t}\ddot{\mathbf{g}}_n^t - \mathbf{x}_n^{t+1}) + \nabla f_i(\breve{\mathbf{x}}_i^{t+1}) + \mathbf{A}_i^\mathsf{T}\mathbf{z}^{t+1}\| \\
\overset{③}{=}& \|(\theta_2\mathsf{L}_n^t - \beta^t\mathbf{A}_n^\mathsf{T}\mathbf{A}_n)(\mathbf{x}_n^t - \mathbf{x}_n^{t+1}) + \nabla f_n(\breve{\mathbf{x}}_n^{t+1}) - \nabla f_n(\mathbf{x}_n^t) + (1-\tfrac{1}{\sigma})\mathbf{A}_n^\mathsf{T}(\mathbf{z}^{t+1} - \mathbf{z}^t)\| \\
\overset{④}{\leq}& \|\mathbf{Q}(\mathbf{x}_n^t - \mathbf{x}_n^{t+1}) + (1-\tfrac{1}{\sigma})\mathbf{A}_n^\mathsf{T}(\mathbf{z}^{t+1} - \mathbf{z}^t)\| + \|\nabla f_n(\breve{\mathbf{x}}_n^{t+1}) - \nabla f_n(\mathbf{x}_n^t)\| \\
\overset{⑤}{\leq}& (1-\tfrac{1}{\sigma})\overline{\mathrm{A}}\|\mathbf{z}^{t+1} - \mathbf{z}^t\| + \|\mathbf{Q}(\mathbf{x}_n^t - \mathbf{x}_n^{t+1})\| + \|\nabla f_n(\breve{\mathbf{x}}_n^{t+1}) - \nabla f_n(\mathbf{x}_n^t)\| \\
\overset{⑥}{\leq}& \underbrace{(1-\tfrac{1}{\sigma})\overline{\mathrm{A}}}_{\triangleq c_3}\|\mathbf{z}^{t+1} - \mathbf{z}^t\| + \underbrace{\{q\overline{\lambda} + L_n\tfrac{1}{\beta^0}\}}_{\triangleq c_4}\cdot\|\beta^t(\mathbf{x}_n^t - \mathbf{x}_n^{t+1})\| + \tfrac{c_5}{\beta^t},
\end{aligned}\tag{55}
$$

where step ① uses the optimality condition as shown in Lemma 3.6(b) that:

$$\rho(\mathbf{c}^t - \mathbf{x}_n^{t+1}) \in \partial h_n(\breve{\mathbf{x}}_n^{t+1}), \text{ with } \rho = \theta_2\mathsf{L}_n^t;$$

step ② uses $\mathbf{c}^t = \mathbf{x}_n^t - \ddot{\mathbf{g}}_n^t/\rho$ as shown in Algorithm 1; step ③ uses the fact that:

$$\ddot{\mathbf{g}}_n^t = \nabla f_n(\mathbf{x}_n^t) + \mathbf{A}_n^\mathsf{T}\mathbf{z}^t + \tfrac{1}{\sigma}\mathbf{A}_n^\mathsf{T}(\mathbf{z}^{t+1} - \mathbf{z}^t) + \beta^t\mathbf{A}_n^\mathsf{T}\mathbf{A}_n(\mathbf{x}_n^t - \mathbf{x}_n^{t+1}),$$

step ④ uses the definition of $\mathbf{Q} \triangleq \theta_2\mathsf{L}_n^t\mathbf{I} - \beta^t\mathbf{A}_n^\mathsf{T}\mathbf{A}_n$, as in Lemma 4.2; step ⑤ uses $\|\mathbf{A}_n\| \leq \overline{\mathrm{A}}$; step ⑥ uses $\|\mathbf{Q}^t\| \leq \beta^t\overline{\lambda}q$ as shown in Lemma 4.3, Inequality (54), and the fact that $\beta^0 \leq \beta^t$.

**Part (c).** We first have the following inequalities:

$$
\begin{aligned}
\sum_{i=1}^{n-1} \|\mathbf{u}_i^{t+1}\| \quad &\overset{①}{=}\quad \| \sum_{i=1}^{n-1}\{\theta_1 \mathsf{L}_i^t(\mathbf{x}_i^{t+1}-\mathbf{x}_i^t) - \beta^t \mathbf{A}_i^\mathsf{T}[\sum_{j=i}^n \mathbf{A}_j(\mathbf{x}_j^{t+1}-\mathbf{x}_j^t)]\}\| \\
&\leq\quad \theta_1\|\sum_{i=1}^{n-1}\mathsf{L}_i^t(\mathbf{x}_i^{t+1}-\mathbf{x}_i^t)\| + \beta^t\|\sum_{i=1}^{n-1}\mathbf{A}_i^\mathsf{T}[\sum_{j=i}^n \mathbf{A}_j(\mathbf{x}_j^{t+1}-\mathbf{x}_j^t)]\| \\
&\overset{②}{\leq}\quad \theta_1(\max_{i=1}^n \mathsf{L}_i^t)\sum_{i=1}^{n-1}\|\mathbf{x}_i^{t+1}-\mathbf{x}_i^t\| + \beta^t\overline{\mathbf{A}}^2(n-1)\|\mathbf{x}^{t+1}-\mathbf{x}^t\| \\
&\overset{③}{\leq}\quad \theta_1(\max_{i=1}^{n-1}\mathsf{L}_i^t)\sqrt{n-1}\|\mathbf{x}^{t+1}-\mathbf{x}^t\| + \overline{\mathbf{A}}^2(n-1)\|\beta^t(\mathbf{x}^{t+1}-\mathbf{x}^t)\| \\
&\overset{④}{\leq}\quad \theta_1(\tfrac{\overline{L}}{\beta^0}+\overline{\mathbf{A}}^2)\cdot\|\beta^t(\mathbf{x}^{t+1}-\mathbf{x}^t)\| + \overline{\mathbf{A}}^2(n-1)\|\beta^t(\mathbf{x}^{t+1}-\mathbf{x}^t)\| \\
&=\quad \underbrace{\{\theta_1(\tfrac{\overline{L}}{\beta^0}+\overline{\mathbf{A}}^2)+\overline{\mathbf{A}}^2(n-1)\}}_{\triangleq c_7'}\cdot\|\beta^t(\mathbf{x}^{t+1}-\mathbf{x}^t)\|, \quad\quad (56)
\end{aligned}
$$

where step ① uses the definition of $\mathbf{u}_i^{t+1}$ for all $i\in[n-1]$; step ② uses $\overline{\mathbf{A}}\triangleq\max_{i=1}^n\|\mathbf{A}_i\|$; step ③ uses $\sum_{j=1}^{n-1}\|\mathbf{x}_j^{t+1}-\mathbf{x}_j^t\|\leq\sqrt{n-1}\|\mathbf{x}^{t+1}-\mathbf{x}^t\|$; step ④ uses $\mathsf{L}_i^t = L_i+\beta^t\|\mathbf{A}_i\|_2^2 \leq \frac{\beta^t L_i}{\beta^0}+\beta^t\overline{\mathbf{A}}^2 \leq \frac{\beta^t\overline{L}}{\beta^0}+\beta^t\overline{\mathbf{A}}^2$ for all $i$.

We have the following results:

$$
\begin{aligned}
&\quad \sum_{i=1}^{n-1}\mathrm{dist}(\partial h_i(\mathbf{x}_i^{t+1})+\nabla f_i(\mathbf{x}_i^{t+1})+\mathbf{A}_i^\mathsf{T}\mathbf{z}^{t+1}) \\
&\overset{①}{=}\quad \sum_{i=1}^{n-1}\|(1-\tfrac{1}{\sigma})\mathbf{A}_i^\mathsf{T}(\mathbf{z}^{t+1}-\mathbf{z}^t)-\nabla f_i(\mathbf{x}_i^t)-\mathbf{u}_i^{t+1}+\nabla f_i(\mathbf{x}_i^{t+1})\| \\
&\leq\quad \sum_{i=1}^{n-1}\|(1-\tfrac{1}{\sigma})\mathbf{A}_i^\mathsf{T}(\mathbf{z}^{t+1}-\mathbf{z}^t)\| + \sum_{i=1}^{n-1}\|\nabla f_i(\mathbf{x}_i^t)-\nabla f_i(\mathbf{x}_i^{t+1})\| + \sum_{i=1}^{n-1}\|\mathbf{u}_i^{t+1}\| \\
&\overset{②}{\leq}\quad \underbrace{(1-\tfrac{1}{\sigma})\overline{\mathbf{A}}(n-1)}_{\triangleq c_6}\|\mathbf{z}^{t+1}-\mathbf{z}^t\|_2^2 + \tfrac{\overline{L}}{\beta^0}\sqrt{n-1}\|\beta^t(\mathbf{x}^t-\mathbf{x}^{t+1})\| + \sum_{i=1}^{n-1}\|\mathbf{u}_i^{t+1}\| \\
&\overset{③}{=}\quad c_6\|\mathbf{z}^{t+1}-\mathbf{z}^t\|_2^2 + \underbrace{(\tfrac{\overline{L}\sqrt{n-1}}{\beta^0}+c_7')}_{\triangleq c_7}\cdot\|\beta^t(\mathbf{x}^t-\mathbf{x}^{t+1})\|_2^2,
\end{aligned}
$$

where step ① uses Part (**a**) in Lemma 4.2 that:

$$
i\in[n-1],\ \partial h_i(\mathbf{x}_i^{t+1})\ni -\mathbf{u}_i^{t+1}-\mathbf{A}_i^\mathsf{T}\mathbf{z}^t-\tfrac{1}{\sigma}\mathbf{A}_i^\mathsf{T}(\mathbf{z}^{t+1}-\mathbf{z}^t)-\nabla f_i(\mathbf{x}_i^t);
$$

step ② uses $\|\mathbf{A}_i\|\leq\overline{\mathbf{A}}$, $f_i(\mathbf{x}_i)$ is $L_i$-smooth, $L_i\leq\overline{L}$, and $\beta^0\leq\beta^t$; step ③ uses Inequality (56).

$\square$

Now, we proceed to prove the theorem.

*Proof.* We define $\mathrm{Crit}(\mathbf{x},\mathbf{z})\triangleq\|\mathbf{A}\mathbf{x}-\mathbf{b}\|+\sum_{i=1}^n\mathrm{dist}(\mathbf{0},\nabla f_i(\mathbf{x}_i)+\partial h_i(\mathbf{x}_i)+\mathbf{A}_i^\mathsf{T}\mathbf{z})$.

We define $\mathbf{q}^t\triangleq\{\mathbf{x}_1^t,\mathbf{x}_2^t,\dots,\mathbf{x}_{n-1}^t,\breve{\mathbf{x}}_n^t\}$.

First, we deduce from Theorem 4.12(a) that

$$
\begin{aligned}
K\beta^T \quad&\geq\quad \sum_{t=1}^T\|\mathbf{z}^{t+1}-\mathbf{z}^t\|_2^2 + \|\beta^t(\mathbf{x}^{t+1}-\mathbf{x}^t)\|_2^2 \\
&\overset{①}{\geq}\quad \tfrac{1}{2T}(\sum_{t=1}^T\|\mathbf{z}^{t+1}-\mathbf{z}^t\|+\|\beta^t(\mathbf{x}^{t+1}-\mathbf{x}^t)\|)^2
\end{aligned}
$$

where step ① uses the inequality $\|\mathbf{a}\|_2^2\geq\tfrac{1}{2T}(\|\mathbf{a}\|_1)^2$ for all $\mathbf{a}\in\mathbb{R}^{2T}$. This leads to:

$$
\sum_{t=1}^T\{\|\mathbf{z}^{t+1}-\mathbf{z}^t\|+\|\beta^t(\mathbf{x}^{t+1}-\mathbf{x}^t)\|\}\leq\sqrt{K\beta^T\cdot 2T}=\mathcal{O}(T^{(p+1)/2}). \quad\quad (57)
$$

Second, using lemma D.1, for all $t\geq 0$, we have:

$$
\begin{aligned}
&\quad \mathrm{Crit}(\mathbf{q}^{t+1},\mathbf{z}^{t+1}) \\
&\leq\quad \underbrace{(c_1+c_3+c_6)}_{\triangleq d_1}\|\mathbf{z}^{t+1}-\mathbf{z}^t\| + \underbrace{(c_4+c_7)}_{\triangleq d_2}\|\beta^t(\mathbf{x}^{t+1}-\mathbf{x}^t)\| + \underbrace{(c_2+c_5)}_{\triangleq d_3}(\beta^t)^{-1}. \quad (58)
\end{aligned}
$$

We further derive:

$$\frac{1}{T}\sum_{t=1}^{T}\mathrm{Crit}(\mathbf{q}^{t+1},\mathbf{z}^{t+1})$$

$$\overset{①}{\leq}\quad \frac{1}{T}\max(d_1,d_2)\sum_{t=0}^{T}\{\|\mathbf{z}^{t+1}-\mathbf{z}^t\|+\|\beta^t(\mathbf{x}^{t+1}-\mathbf{x}^t)\|\}+\frac{d_3}{T}\sum_{t=0}^{T}(\beta^t)^{-1}$$

$$\overset{②}{\leq}\quad \mathcal{O}(T^{(p-1)/2})+\frac{d_3}{T}\sum_{t=0}^{T}(\beta^t)^{-1}$$

$$\overset{③}{\leq}\quad \mathcal{O}(T^{(p-1)/2})+\mathcal{O}(T^{1-p-1}),$$

Here, step ① uses Inequality 58; step ② uses Inequality (57); step ③ uses $\beta^t=\mathcal{O}(t^p)$, and the fact that $\sum_{t=1}^{T}t^{-p}\leq\frac{T^{(1-p)}}{1-p}$ if $p\in(0,1)$ (refer to Lemma A.6).

In particular, with the choice $p=1/3$, we have: $\frac{1}{T}\sum_{t=1}^{T}\mathrm{Crit}(\mathbf{q}^{t+1},\mathbf{z}^{t+1})\leq\mathcal{O}(T^{-1/3})$.

$\square$

### D.14 PROOF OF LEMMA 4.17

*Proof.* We let $\frac{\mathbf{z}^t}{\sqrt{\beta^t}}\triangleq\hat{\mathbf{z}}^t$ for all $t$.

Initially, we derive:

$$\sum_{t=1}^{\infty}(1-\sqrt{\tfrac{\beta^t}{\beta^{t+1}}})^2\quad\overset{①}{=}\quad\sum_{t=1}^{\infty}(1-\sqrt{\tfrac{1+\xi t^p}{1+\xi(t+1)^p}})^2$$

$$\overset{②}{\leq}\quad\sum_{t=1}^{\infty}(1-\sqrt{\tfrac{t^p}{(t+1)^p}})^2$$

$$=\quad\sum_{t=1}^{\infty}\frac{\{(t+1)^{p/2}-t^{p/2}\}^2}{(t+1)^p}$$

$$\overset{③}{\leq}\quad\sum_{t=1}^{\infty}\frac{\{\frac{p}{2}\cdot t^{(p/2-1)}\}^2}{t^p}$$

$$\overset{④}{\leq}\quad\frac{1}{4}\sum_{t=1}^{\infty}\frac{t^{(p-2)}}{t^p}$$

$$\overset{⑤}{\leq}\quad 1/2, \tag{59}$$

where step ① uses $\beta^t=\beta^0(1+\xi t^p)$ for all $t\geq0$; step ② uses $\frac{1+\xi t^p}{1+\xi(t+1)^p}\leq\frac{\xi t^p}{\xi(t+1)^p}$; step ③ uses Lemma A.4 that $(t+1)^{p/2}-t^{p/2}\leq\frac{p}{2}t^{(p/2-1)}$ for all $t\geq1$ and $\frac{p}{2}\in(0,1)$; step ④ uses $p\leq1$ and $\frac{1}{t+1}\leq\frac{1}{t}$; step ⑤ uses $\sum_{t=1}^{\infty}\frac{1}{t^2}=\frac{\pi^2}{6}<2$.

**Part (a).** We have: $\|\hat{\mathbf{z}}^t\|_2^2=\|\frac{\mathbf{z}^t}{\sqrt{\beta^t}}\|_2^2=\frac{1}{\beta^t}\|\mathbf{z}^t\|_2^2\overset{①}{\leq}Z<+\infty$, where step ① uses Lemma 4.10.

**Part (b).** We derive the following inequalities:

$$\sum_{t=1}^{\infty}\|\hat{\mathbf{z}}^{t+1}-\hat{\mathbf{z}}^t\|_2^2\quad\overset{①}{=}\quad\sum_{t=1}^{\infty}\|\frac{\mathbf{z}^{t+1}}{\sqrt{\beta^{t+1}}}-\frac{\mathbf{z}^t}{\sqrt{\beta^t}}\|_2^2$$

$$=\quad\sum_{t=1}^{\infty}\|\frac{\mathbf{z}^{t+1}-\mathbf{z}^t}{\sqrt{\beta^{t+1}}}-\mathbf{z}^t(\frac{1}{\sqrt{\beta^t}}-\frac{1}{\sqrt{\beta^{t+1}}})\|_2^2$$

$$\overset{②}{\leq}\quad 2\sum_{t=1}^{\infty}\|\frac{\mathbf{z}^{t+1}-\mathbf{z}^t}{\sqrt{\beta^{t+1}}}\|_2^2+2\sum_{t=1}^{\infty}\|\mathbf{z}^t(\frac{1}{\sqrt{\beta^t}}-\frac{1}{\sqrt{\beta^{t+1}}})\|_2^2$$

$$\overset{③}{\leq}\quad 2\sum_{t=1}^{\infty}\frac{1}{\beta^t}\|\mathbf{z}^{t+1}-\mathbf{z}^t\|_2^2+2\sum_{t=1}^{\infty}\frac{1}{\beta^t}\|(1-\sqrt{\tfrac{\beta^t}{\beta^{t+1}}})\cdot\mathbf{z}^t\|_2^2$$

$$\overset{④}{\leq}\quad 2\ddot{Z}+2Z\cdot\sum_{t=1}^{\infty}(1-\sqrt{\tfrac{\beta^t}{\beta^{t+1}}})^2$$

$$\overset{⑤}{\leq}\quad 2\ddot{Z}+2Z\cdot\frac{1}{2}$$

$$<\quad 2(\ddot{Z}+Z),$$

where step ① uses the definition $\frac{\mathbf{z}^t}{\sqrt{\beta^t}} \triangleq \hat{\mathbf{z}}^t$ for all $t$; step ② uses $\|\mathbf{a} - \mathbf{b}\|_2^2 \leq 2\|\mathbf{a}\|_2^2 + 2\|\mathbf{b}\|_2^2$; step ③ uses $\frac{1}{\beta^{t+1}} \leq \frac{1}{\beta^t}$; step ④ uses $\sum_{t=1}^{\infty} \frac{1}{\beta^t} \|\mathbf{z}^{t+1} - \mathbf{z}^t\|_2^2 \leq \ddot{Z}$ and $\frac{1}{\beta^t}\|\mathbf{z}^t\|_2^2 \leq Z$, as shown in Lemma 4.10; step ⑤ uses Inequality (59).

$\square$

# E   ADDITIONAL EXPERIMENT DETAILS AND RESULTS

We offer further experimental details in Sections E.1 and E.2, and include additional results in Section E.3.

## E.1   DATASETS

We incorporate four datasets in our experiments, including both randomly generated data and publicly available real-world data. These datasets serve as our data matrices $\mathbf{D} \in \mathbb{R}^{\dot{m} \times \dot{d}}$. The dataset names are as follows: 'TDT2-$\dot{m}$-$\dot{d}$', 'sector-$\dot{m}$-$\dot{d}$', 'mnist-$\dot{m}$-$\dot{d}$', and 'randn-$\dot{m}$-$\dot{d}$'. Here, randn($m, n$) refers to a function that generates a standard Gaussian random matrix with dimensions $m \times n$. The matrix $\mathbf{D} \in \mathbb{R}^{\dot{m} \times \dot{d}}$ is constructed by randomly selecting $\dot{m}$ examples and $\dot{d}$ dimensions from the original real-world dataset (http://www.cad.zju.edu.cn/home/dengcai/Data/TextData.html,https://www.csie.ntu.edu.tw/~cjlin/libsvm/). We normalize each column of $\mathbf{D}$ to have a unit norm and center the data by subtracting the mean.

## E.2   PROJECTION ON ORTHOGONALITY CONSTRAINTS

When $h(\mathbf{x}) = \iota_{\mathcal{M}}(\mathrm{mat}(\mathbf{x}))$ with $\Omega \triangleq \{\mathbf{V} \mid \mathbf{V}^{\mathsf{T}}\mathbf{V} = \mathbf{I}\}$, computing the proximal operator reduces to the following optimization problem:

$$\bar{\mathbf{x}} \in \arg\min_{\mathbf{x}} \frac{\mu}{2}\|\mathbf{x} - \mathbf{x}'\|_2^2, \, s.t. \, \mathrm{mat}(\mathbf{x}) \in \mathcal{M} \triangleq \{\mathbf{V} \mid \mathbf{V}^{\mathsf{T}}\mathbf{V} = \mathbf{I}\}.$$

This is the nearest orthogonality matrix problem, and the optimal solution can be computed as $\bar{\mathbf{x}} = \mathrm{vec}(\hat{\mathbf{U}}\hat{\mathbf{V}}^{\mathsf{T}})$, where $\mathrm{mat}(\mathbf{x}') = \hat{\mathbf{U}}\mathrm{Diag}(\mathbf{s})\hat{\mathbf{U}}^{\mathsf{T}}$ is the singular value decomposition of the matrix $\mathrm{mat}(\mathbf{x}')$. Please refer to (Lai & Osher, 2014).

## E.3   ADDITIONAL EXPERIMENT RESULTS

We present the convergence curves of the compared methods for solving sparse PCA with varying $\dot{\rho} = \{1, 10, 100, 1000\}$ and $\beta^0 = \{10\dot{\rho}, 50\dot{\rho}, 100\dot{\rho}, 500\dot{\rho}\}$, as shown in Figures 2 to 17. Please refer to Table 2 for the mapping between $(\dot{\rho}, \beta^0)$ and the corresponding convergence curves. The results demonstrate that the proposed IPDS-ADMM consistently outperforms the other methods in terms of speed for solving the sparse PCA problem, particularly for the ranges $\dot{\rho} = \{1, 10, 100, 1000\}$ and $\beta^0 = \{50\dot{\rho}, 100\dot{\rho}\}$.

Table 2: The mapping between $(\dot{\rho}, \beta^0)$ and the corresponding convergence curves for sparse PCA.

|  | $\beta^0 = 10\dot{\rho}$ | $\beta^0 = 50\dot{\rho}$ | $\beta^0 = 100\dot{\rho}$ | $\beta^0 = 500\dot{\rho}$ |
|---|---|---|---|---|
| $\dot{\rho} = 1$ | Figure 2 | Figure 6 | Figure 10 | Figure 14 |
| $\dot{\rho} = 10$ | Figure 3 | Figure 7 | Figure 11 | Figure 15 |
| $\dot{\rho} = 100$ | Figure 4 | Figure 8 | Figure 12 | Figure 16 |
| $\dot{\rho} = 1000$ | Figure 5 | Figure 9 | Figure 13 | Figure 17 |

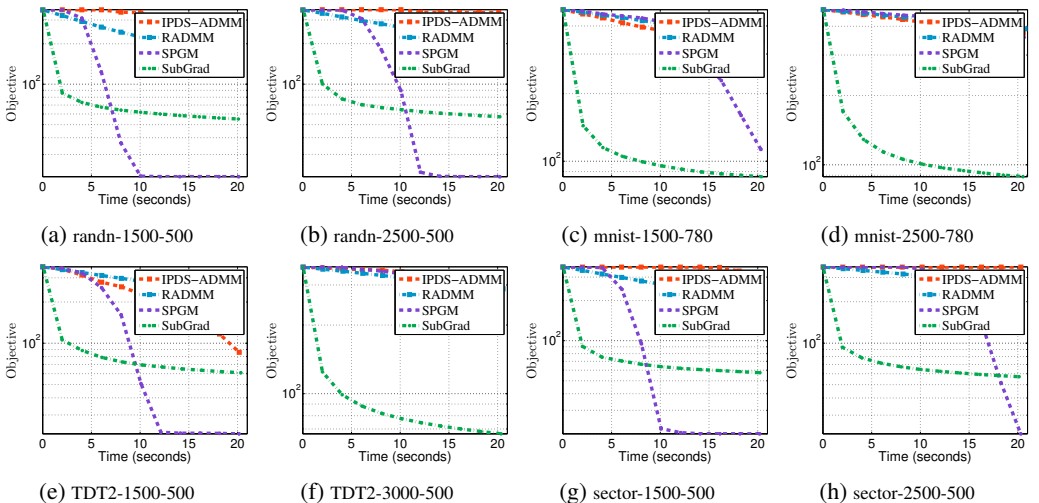

Figure 2: Convergence curves of methods for sparse PCA with $\dot{\rho} = 1$ and $\beta^0 = 10\dot{\rho}$.

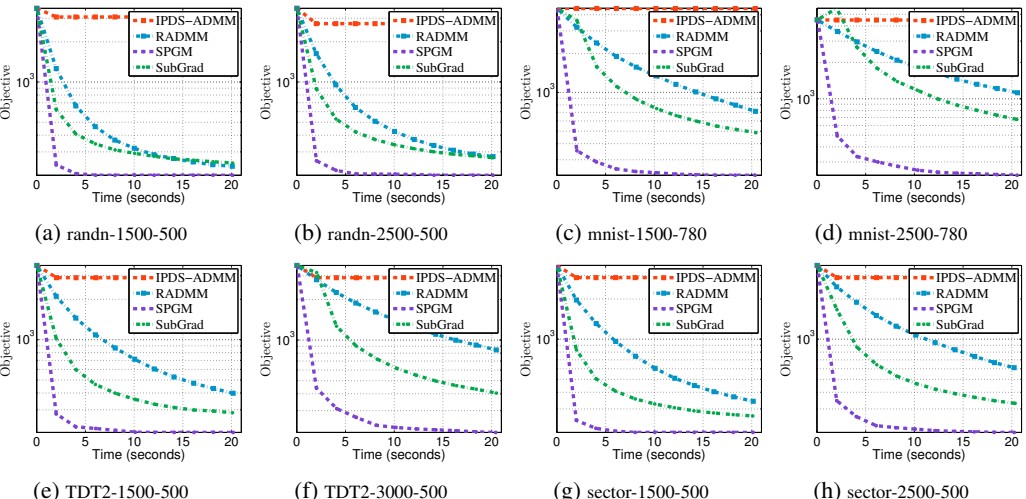

Figure 3: Convergence curves of methods for sparse PCA with $\dot{\rho} = 10$ and $\beta^0 = 10\dot{\rho}$.

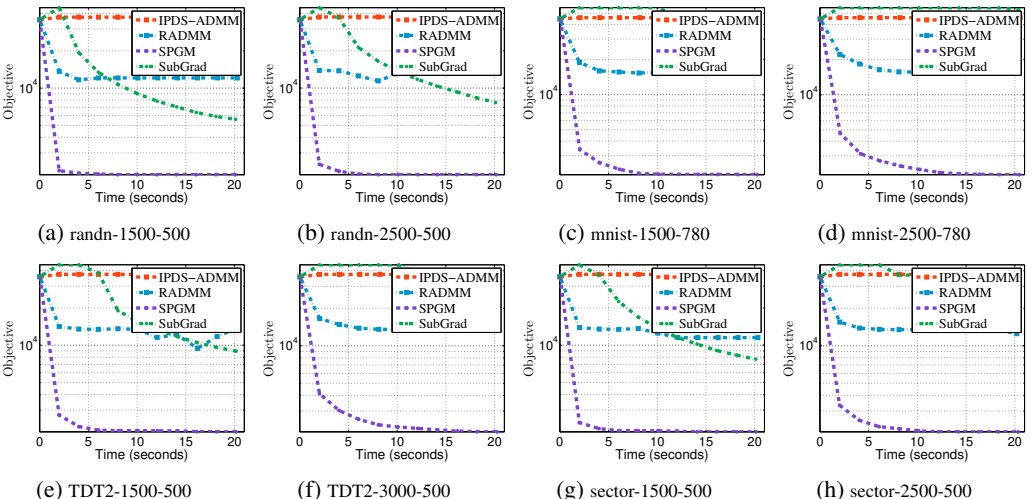

Figure 4: Convergence curves of methods for sparse PCA with $\dot{\rho} = 100$ and $\beta^0 = 10\dot{\rho}$.

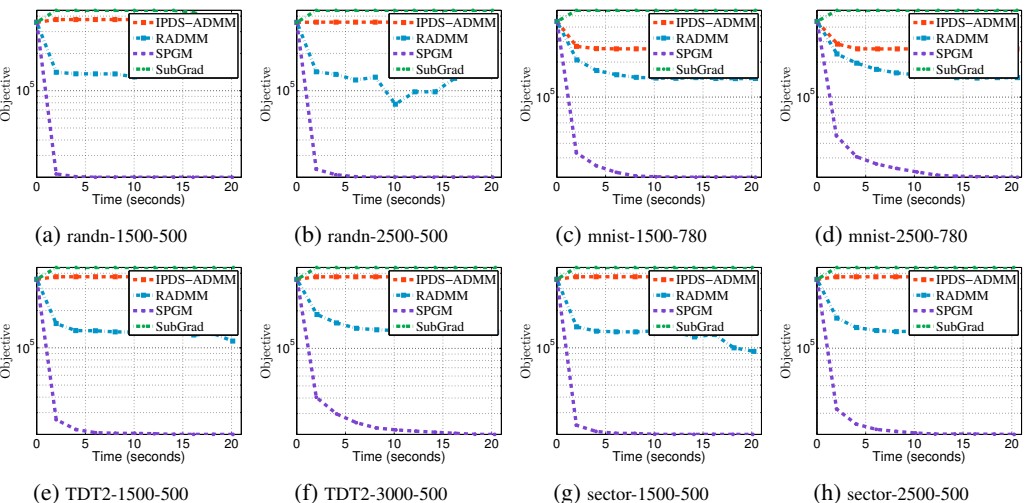

Figure 5: Convergence curves of methods for sparse PCA with $\dot{\rho} = 1000$ and $\beta^0 = 10\dot{\rho}$.

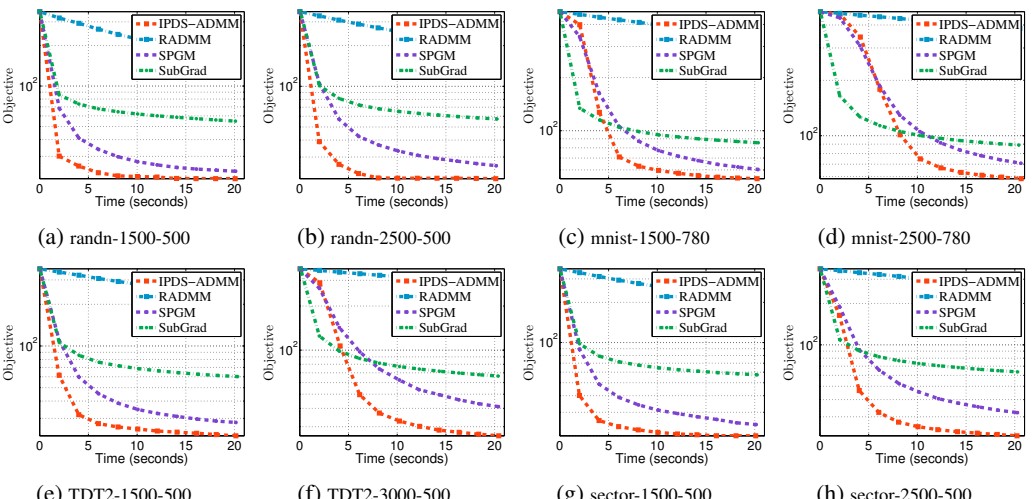

Figure 6: Convergence curves of methods for sparse PCA with $\dot{\rho} = 1$ and $\beta^0 = 50\dot{\rho}$.

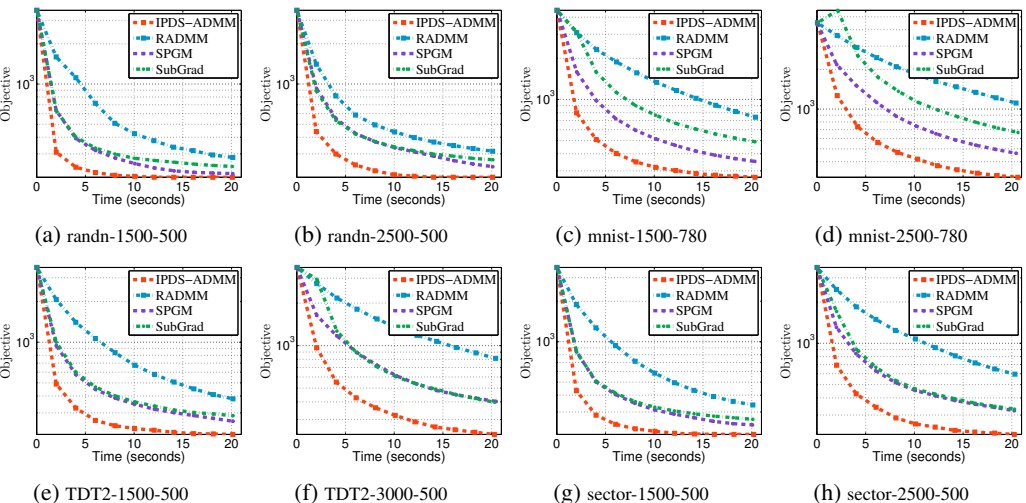

Figure 7: Convergence curves of methods for sparse PCA with $\dot{\rho} = 10$ and $\beta^0 = 50\dot{\rho}$.

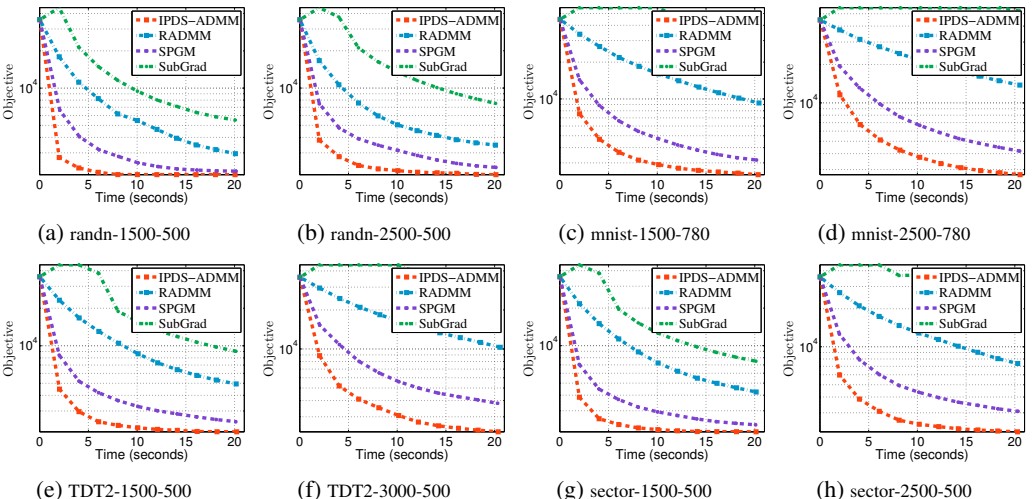

Figure 8: Convergence curves of methods for sparse PCA with $\dot{\rho} = 100$ and $\beta^0 = 50\dot{\rho}$.

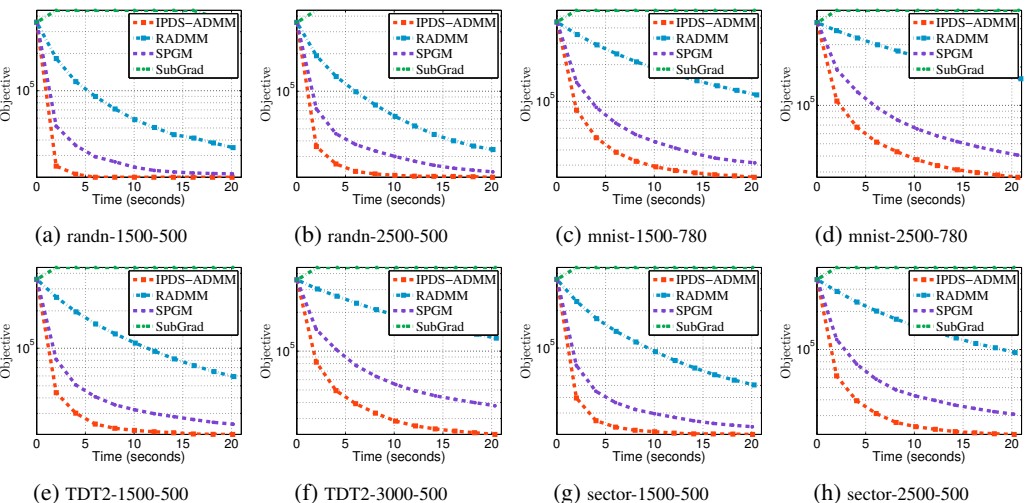

Figure 9: Convergence curves of methods for sparse PCA with $\dot{\rho} = 1000$ and $\beta^0 = 50\dot{\rho}$.

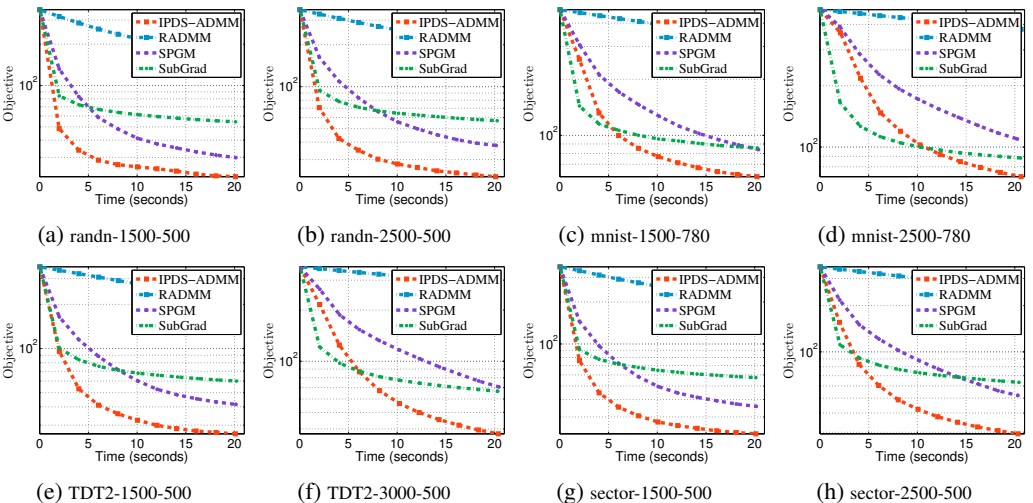

Figure 10: Convergence curves of methods for sparse PCA with $\dot{\rho} = 1$ and $\beta^0 = 100\dot{\rho}$.

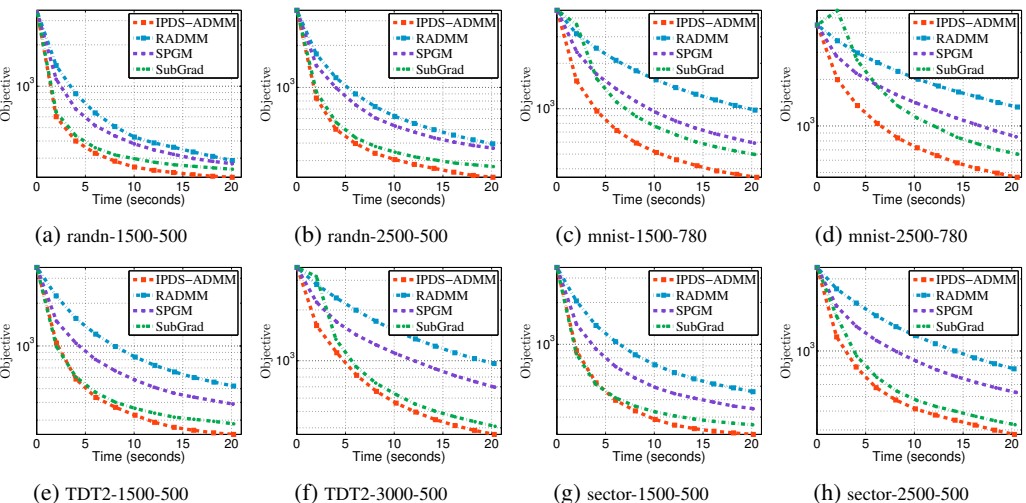

Figure 11: Convergence curves of methods for sparse PCA with $\dot{\rho} = 10$ and $\beta^0 = 100\dot{\rho}$.

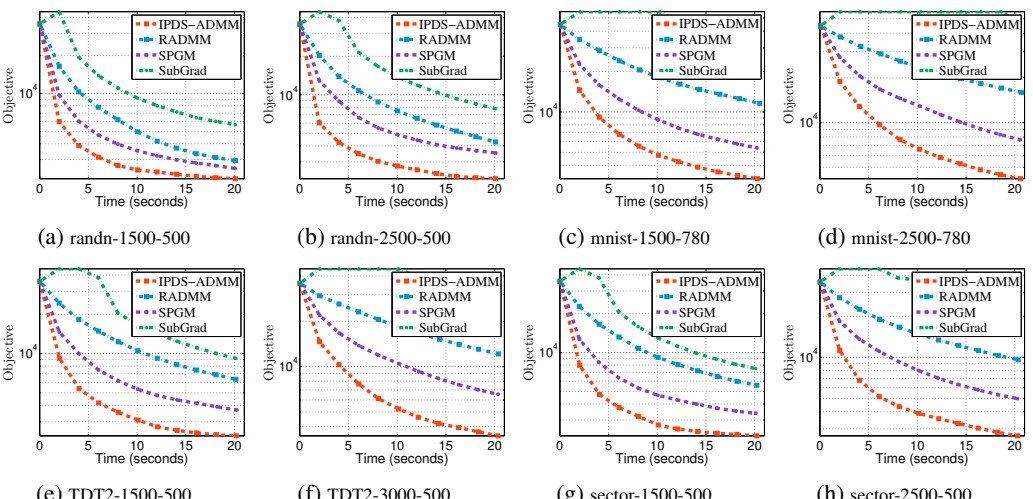

Figure 12: Convergence curves of methods for sparse PCA with $\dot{\rho} = 100$ and $\beta^0 = 100\dot{\rho}$.

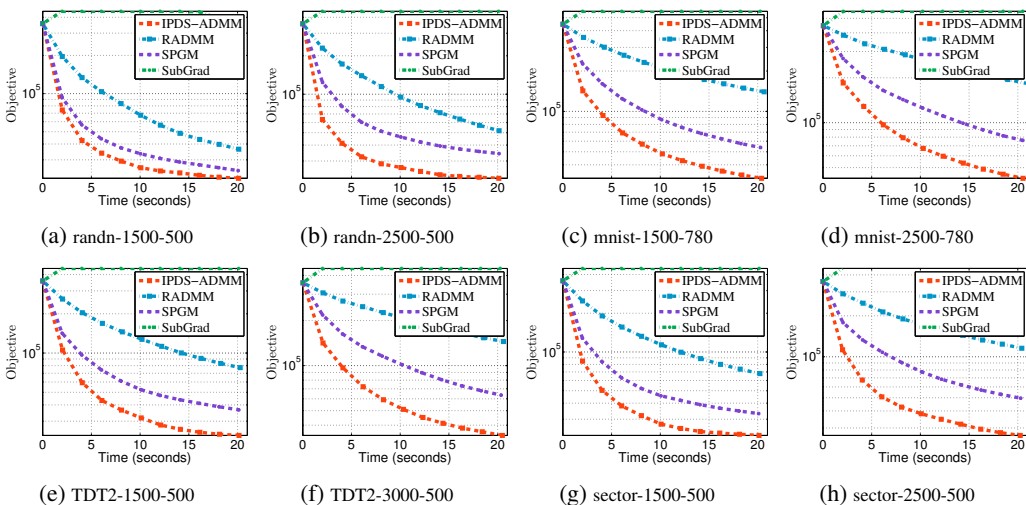

Figure 13: Convergence curves of methods for sparse PCA with $\dot{\rho} = 1000$ and $\beta^0 = 100\dot{\rho}$.

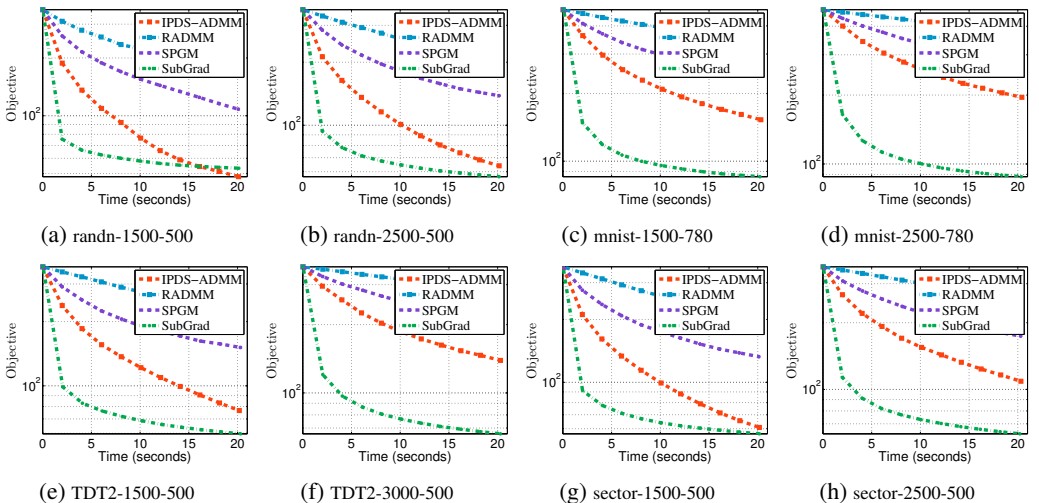

Figure 14: Convergence curves of methods for sparse PCA with $\dot{\rho} = 1$ and $\beta^0 = 500\dot{\rho}$.

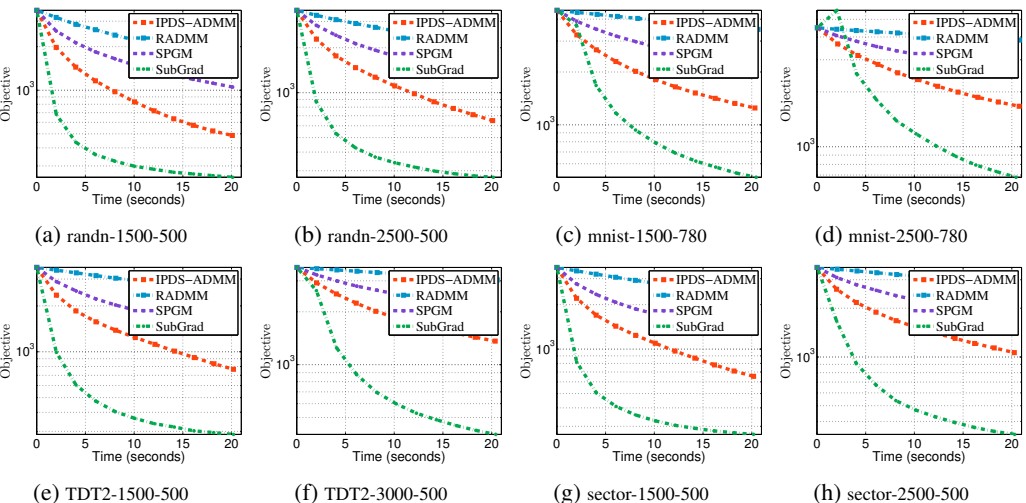

Figure 15: Convergence curves of methods for sparse PCA with $\dot{\rho} = 10$ and $\beta^0 = 500\dot{\rho}$.

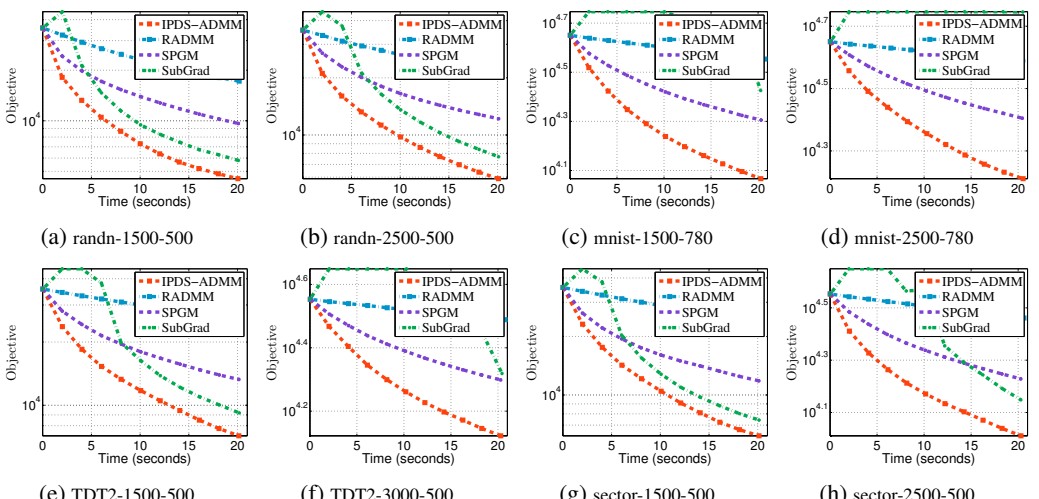

Figure 16: Convergence curves of methods for sparse PCA with $\dot{\rho} = 100$ and $\beta^0 = 500\dot{\rho}$.

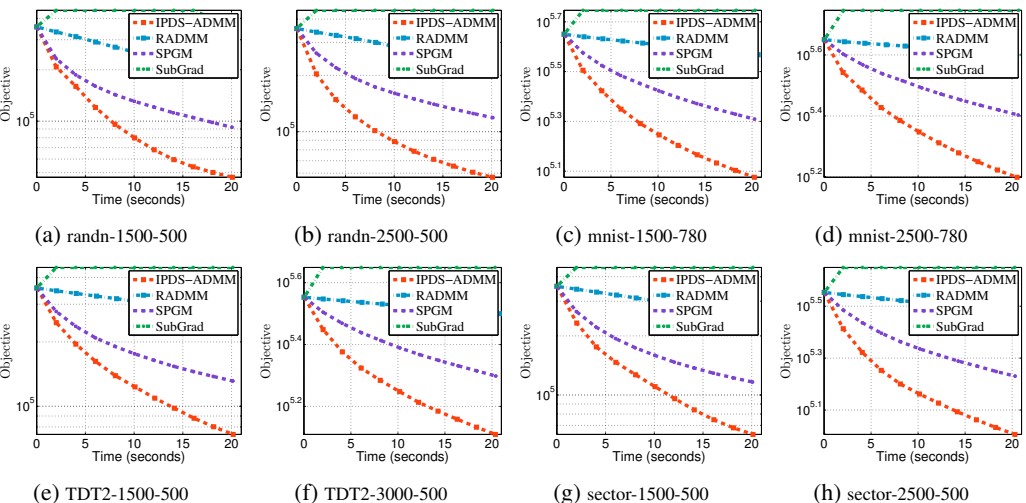

Figure 17: Convergence curves of methods for sparse PCA with $\dot{\rho} = 1000$ and $\beta^0 = 500\dot{\rho}$.

