# OpenReview forum: "ADMM for Nonconvex Optimization under Minimal Continuity Assumption"
_ICLR.cc/2025/Conference — ICLR 2025 Poster_

### Official Review · Reviewer_ARrq · 2024-10-30

**Soundness:** 3
**Presentation:** 3
**Contribution:** 2
**Rating:** 6
**Confidence:** 2

**Summary:**

This paper proposes a variety ADMM algorithm called IPDS-ADMM for multi-block nonconvex composite optimization problems, introducing an Increasing Penalization and Decreasing Smoothing strategy using Moreau envelope technique to further decrease the smoothness conditions on the objective function of previous works. The authors also conducted some experiments on sparse PCA problem to show the effectiveness of the proposed method.

**Strengths:**

1.	The proposed IPDS-ADMM weaken the smoothness assumption for nonconvex multi-block composite optimization that requiring continuity in only the last block of the objective function.

2.	The convergence analysis of the proposed algorithm is given for the case that the associated matrix is either bijective or surjective.

**Weaknesses:**

1.  The iteration complexity of the proposed algorithm seems to be not outstanding compared to previous nonconvex ADMM approaches mentioned in Table 1. Also, in section 3, the authors claim that IPDS-ADMM improve the complexity from O(ϵ^(-4)) to O(ϵ^(-3)) compared with RADMM, but it is unclear whether the comparison is fair due to RADMM focus on the manifold optimization problem and have the different assumptions with the proposed algorithm.

2. Regarding the experiments, how did the author choose the hyper-parameters of the other compared algorithms and are those hyper-parameters optimized for each algorithm? Further details are needed to ensure the fairness of the comparison. Also, the authors conduct the experiment on some small-scale datasets. Is the proposed algorithm efficient in such tasks with large-scale datasets?

3. A typo. ‘Mereau Envelope’ in Line 221 should be Moreau Envelope.

**Questions:**

1.  The iteration complexity of the proposed algorithm seems to be not outstanding compared to previous nonconvex ADMM approaches mentioned in Table 1. Also, in section 3, the authors claim that IPDS-ADMM improve the complexity from O(ϵ^(-4)) to O(ϵ^(-3)) compared with RADMM, but it is unclear whether the comparison is fair due to RADMM focus on the manifold optimization problem and have the different assumptions with the proposed algorithm.

2. Regarding the experiments, how did the author choose the hyper-parameters of the other compared algorithms and are those hyper-parameters optimized for each algorithm? Further details are needed to ensure the fairness of the comparison. Also, the authors conduct the experiment on some small-scale datasets. Is the proposed algorithm efficient in such tasks with large-scale datasets?

3. A typo. ‘Mereau Envelope’ in Line 221 should be Moreau Envelope.

---

> ### Author Response · Authors · 2024-11-18
>
> **Weakness 1-a. The iteration complexity of the proposed algorithm seems to be not outstanding compared to previous nonconvex ADMM approaches mentioned in Table 1.**
>
> **Response.**
>
> 1. The complexity comparison with all existing methods, except for RADMM, is not comparable, as these methods assume $h_n(x_n) = 0$. Please refer to the third column of Table 1.
>
> 2. RADMM is the only comparable method, as it permits  $h_n(x_n)$ to be the indicator function of the orthogonality constraint. However, it cannot handle linearly constrained problems, especially when $A_n$ is surjective or bijective. **Importantly**, RADMM results in suboptimal complexity.
>
> **Weakness 1-b. Also, in section 3, the authors claim that IPDS-ADMM improve the complexity from O(ϵ^(-4)) to O(ϵ^(-3)) compared with RADMM, but it is unclear whether the comparison is fair due to RADMM focus on the manifold optimization problem and have the different assumptions with the proposed algorithm.**
>
> **Response.** The comparison is fair for the following reasons:
>
> 1. Our assumptions are less restrictive than those used in RADMM. Specifically, Assumptions 1.1–1.7 are satisfied by the sparse PCA problem, and we also allow $A_n$ to be surjective or bijective. For details, please refer to the "Existing Challenges" section in Lines 113–120.
>
> 2. Both methods use the same definition of an $\epsilon$-approximate critical point.
>
>
> **Weakness 2-a. Regarding the experiments, how did the author choose the hyper-parameters of the other compared algorithms and are those hyper-parameters optimized for each algorithm? Further details are needed to ensure the fairness of the comparison.**
>
> **Response.** **Question 3:  Given that the algorithm involves several parameters, is it straightforward to determine their values in practice? It would be helpful if additional experiments were conducted to demonstrate the robustness of the algorithm with respect to parameter selection and to provide guidance on how to choose these parameters effectively.**
>
>  **Response.** The algorithm generally involves six parameters: $(\beta^0, \theta_1, \theta_2, \sigma, \delta, \xi )$. However, we argue that these parameters are primarily selected to ensure the theoretical convergence of the algorithm, and the algorithm's performance is not highly sensitive to their specific values. We consider the sparse PCA problem.
>
> 1. The proximal parameter $\theta_1$ is typically set to a constant slightly greater than $1$, such as 1.01.
>
> 2. The proximal parameter $\theta_2$ is typically set to a constant slightly greater than $1/2+\chi_1$, such as $1/2+\chi_1+10^{-12}$.
>
> 3. The relaxation parameter $\sigma$ is suggested to be around the golden ratio $1.618$, as suggested by Min Li, Defeng Sun, and Kim-Chuan Toh in *"A Majorized ADMM with Indefinite Proximal Terms for Linearly Constrained Convex Composite Optimization" (SIOPT, 2016)*.
>
> 4. The parameter $\delta$ only effects $\mu^t$ through $\mu^t = 1/ (\overline{\lambda} \delta \beta^t)$, where $\delta$ is a sufficiently small constant, and $\beta^t$ can be relatively large. Thus, the choice of $\delta$ is not expected to significantly affect the overall performance of the proposed method.
>
> 5. The parameter $\xi$ is typically set to a constant $0.5$.
>
> 6. In our revised paper, we present the convergence curves of the compared methods with varying $\dot{\rho} \in (1, 10, 100, 1000)$ and $\beta^0 = (10\dot{\rho}, 50 \dot{\rho}, 100 \dot{\rho}, 500 \dot{\rho})$, as shown in Figures 2 to 17. Please refer to Page 36-Page 43 of the manuscript.
>
> **Weakness 2-b. Also, the authors conduct the experiment on some small-scale datasets. Is the proposed algorithm efficient in such tasks with large-scale datasets?**
>
> **Response.** This paper primarily focuses on improving the complexity of current algorithms. We believe that, with appropriately chosen parameters, our algorithm continues to perform well, even on large-scale datasets.
>
> **Weakness 3.A typo. ‘Mereau Envelope’ in Line 221 should be Moreau Envelope.**
>
> **Response.** Thank you for your careful reading. We will correct it in the updated paper.

---

> > ### Comment · Reviewer_ARrq · 2024-11-25
> >
> > Thank you  for the reply. After reading the rebuttal, I think my concerns are addressed and I have raised my rating to 6.

---

### Official Review · Reviewer_GkTh · 2024-11-03

**Soundness:** 3
**Presentation:** 2
**Contribution:** 2
**Rating:** 6
**Confidence:** 3

**Summary:**

This paper introduces a novel proximal linearized ADMM algorithm with an Increasing Penalization and Decreasing Smoothing (IPDS) strategy, termed IPDS-ADMM. The proposed method tackles multi-block nonconvex composite optimization problems under a less stringent assumptions, requiring continuity in only one block of the objective function. Additionally, this work provides the first complexity result for nonconvex, nonsmooth minimax problems.

**Strengths:**

* The algorithm guarantees convergence for ADMM under less stringent conditions.

* It establishes the first convergence results for solving nonconvex, nonsmooth minimax problems.

* The IPDS strategy ensures convergence for matrices $A$ that are either bijective or surjective.

**Weaknesses:**

* It remains unclear whether the IPDS strategy can be extended to handle inequality or more general constraints.

* In Section 4, given the compact results and dense notation, it would be helpful to emphasize the role of the IPDS strategy in the analysis, with additional explanatory comments to enhance clarity.

* A suggestion is to use alternative notation in Lemma 4.8 for $\epsilon_1,\epsilon_2,\epsilon_3$, as these are typically reserved for sufficiently small constants.

* While the sparse PCA experiments provide valuable insights, demonstrating the method’s effectiveness across a broader range of applications could further support its effictiveness in diverse problem settings.

**Questions:**

See weaknesses.

---

> ### Author Response · Authors · 2024-11-18
>
> **Weakness 1. It remains unclear whether the IPDS strategy can be extended to handle inequality or more general constraints.**
>
> **Response.** Our algorithm can be extended to handle inequality, as we permit blocks of function $h_i(x_i)$ for $i\in[n-1]$ to be nonsmooth and non-Lipschitz, such as indicator functions of constraint sets. Refer to L152-154, and the Structured Sparse Phase Retrieval application in L184-191.
>
> Our algorithm can also be extended to handle general constraints, provided the Jacobian mapping is Lipschitz continuous and smooth. Nevertheless, the optimization framework in (1) is already flexible enough to address various applications, including Sparse PCA, Structured SPR, Robust Sparse Regression, Dual PCP, and Robust LRA (see L176-191 and L934-L956). For simplicity, we focus only on linear constraints in this work, leaving the extension to general constraints for future research.
>
> **Weakness 2. In Section 4, given the compact results and dense notation, it would be helpful to emphasize the role of the IPDS strategy in the analysis, with additional explanatory comments to enhance clarity.**
>
> **Response.** We emphasize that the IPDS strategy plays a crucial role in improving the iteration complexity of RADMM (Li et al., 2022), reducing it from $O(\epsilon^{-4})$ to $O(\epsilon^{-3})$. We have highlighted this important point in Lines 303–305. Thank you for the suggestion.
>
> **Weakness 3. A suggestion is to use alternative notation in Lemma 4.8 for $\epsilon_1,\epsilon_2,\epsilon_3$, as these are typically reserved for sufficiently small constants.**
>
> **Response.** We do use $\epsilon_1=\tfrac{1}{2}-\tfrac{\theta_1}{2},\epsilon_2= \theta_2 -1/2 -\chi_1$ to represent sufficiently small constants. In practice, the proximal parameter $\theta_1$ is typically set to a value slightly greater than $1$, such as $1.01$, while $\theta_2$ is suggested to be slightly greater than $1/2+\chi_1$, for example, $1/2+\chi_1+10^{-12}$.
>
> **Weakness 4.While the sparse PCA experiments provide valuable insights, demonstrating the method’s effectiveness across a broader range of applications could further support its effictiveness in diverse problem settings.**
>
> **Response.** We have demonstrated the applicability of the proposed method to Structured SPR (Lines 184–191) and other applications such as Robust Sparse Regression, Dual PCP, and Robust LRA (Lines 934–956). To facilitate a comparison with existing state-of-the-art techniques, we briefly discuss the special case of the sparse PCA problem and compare our algorithm with the RADMM algorithm.

---

> > ### Comment · Reviewer_GkTh · 2024-11-24
> >
> > Thank you for addressing my questions. I will raise the score accordingly.

---

### Official Review · Reviewer_KMzj · 2024-11-04

**Soundness:** 3
**Presentation:** 2
**Contribution:** 2
**Rating:** 6
**Confidence:** 3

**Summary:**

This paper investigates the ADMM for multi-block composite optimization problems. An adaptive penalization technique is employed to enhance convergence. The global convergence result of the proposed proximal linearized ADMM is derived under mild smoothness conditions.

**Strengths:**

This theoretical paper provides complete theoretical results to prove the proposed algorithms achieve competitive complexity under mild smoothness conditions. The analysis covers a broad range of problems, including those with linear constraints and problems that involve explicit proximal operators (e.g., manifold optimization). This approach encompasses a wide variety of problem categories.

**Weaknesses:**

The paper is written in a way that compiles all the technical material with complex notations, making it difficult for readers to follow. It would be beneficial if the author could summarize the results more effectively and provide insights and discussions.

In the numerical section on sparse PCA, it is unclear why the parameters $\dot{\rho} = 10$ and $\beta^0 = 50\dot{\rho}$ are used consistently. These values appear to be dependent on the Lipschitz constant as suggested in the theoretical section. Given that the problem scales differ in the experiments, it would be more consistent with the theoretical results to adjust the parameters accordingly.

Additionally, there are notational errors and typos that should be corrected. For example, in line 1565, the RHS should include coefficient 2. I have noticed several similar issues throughout the paper that should be rechecked.

**Questions:**

1. Could you provide more insight into the increasing $\beta$ update rule in (2)? I notice in line 1809 that there is a trade-off between two terms to derive the best complexity results. However, can it be shown that such type of $\beta$ update rule is optimal? A discussion or analysis regarding the optimality of this approach would be valuable.

2. In the numerical experiments section, the function used in the sparse PCA example does not appear to satisfy the smoothness assumption. It would be beneficial to conduct further analysis that incorporates the properties of the orthogonal constraints, leading to an equivalent formulation that fully aligns with the assumptions and supports the theoretical analysis.

3. Given that the algorithm involves several parameters, is it straightforward to determine their values in practice? It would be helpful if additional experiments were conducted to demonstrate the robustness of the algorithm with respect to parameter selection and to provide guidance on how to choose these parameters effectively.

---

> ### Author Response · Authors · 2024-11-18
>
> **Weakness 1. Complex notations and difficult to follow the results**
>
> **Response.** As our focus is on multi-block nonsmooth composite problems, these notations are necessary. We have provided a high-level overview of the proof strategy in L342-346.
>
> **Weakness 2. why the parameters $\dot{\rho}=10$  and $\beta^0=50 \dot{\rho}$ are used consistently. Given that the problem scales differ in the experiments, it would be more consistent with the theoretical results to adjust the parameters accordingly.**
>
> **Response.**
>
> 1. In the first draft of this paper, we have experimented with different values of  $\dot{\rho}$, specifically $\dot{\rho}\in(1,10,100)$, corresponding to Figure 2, 1, and 3, respectively.
>
> 2. The condition $\beta^t \geq L_n/ (\delta \overline{\lambda})$ from the second inequality in (2) can always be met when $t$ is sufficiently large. Therefore, $\beta^0$ can be set to any positive constant.
>
> 3.  In our revision, we present the convergence curves of the compared methods with varying $\dot{\rho} \in (1, 10, 100, 1000)$ and $\beta^0 = (10\dot{\rho}, 50 \dot{\rho}, 100 \dot{\rho}, 500 \dot{\rho})$, as shown in Figures 2 to 17. Please refer to Table 2 in L1892-1899 for the mapping between ($\dot{\rho}, \beta^0)$ and the corresponding convergence curves. These results demonstrate that the proposed IPDS-ADMM consistently outperforms other methods in terms of speed for solving the sparse PCA problem, particularly for the ranges $\dot{\rho} \in (1, 10, 100, 1000)$  and $\beta^0 = (50 \dot{\rho}, 100 \dot{\rho})$. We have updated the supplementary material to ensure reproducibility.
>
> **Weakness 3. Additionally, there are notational errors and typos that should be corrected. For example, in line 1565, the RHS should include coefficient 2. I have noticed several similar issues throughout the paper that should be rechecked.**
>
>  **Response.** Thank you for bringing the typo in Line 1565 to our attention. We have carefully reviewed this issue and confirm that, while the typo is present, it does not affect the correctness of the lemma.
>
> We appreciate your thorough review and kindly request that you explicitly point out any additional issues you have identified in the paper to ensure we address them comprehensively.
>
> **Question 1: Could you provide more insight into the increasing $\beta$ update rule in (2)? I notice in line 1809 that there is a trade-off between two terms to derive the best complexity results. However, can it be shown that such type of $\beta$ update rule is optimal? A discussion or analysis regarding the optimality of this approach would be valuable.**
>
>  **Response.** Thank you for your suggestions. We will include the following brief discussion:
>
> We observe that $$\tfrac{1}{T}\sum_{t=0}^T Crit(q^{t+1},z^{t+1}) \leq O(T^{p-1}) + O(T^{-1}) + O(T^{-2p})$$ with $p\in(0,1/2)$. Minimizing the worse-case complexity of the right-hand side of this inequality with respect to $p$ yields: $$\arg \min_{p \in (0,1/2)}~\max(p-1,-2p) =1/3.$$ Thus, choosing $p=1/3$ achieves the optimal trade-off between the two terms, resulting in the best complexity bounds.
>
> **Question 2: In the numerical experiments section, the function used in the sparse PCA example does not appear to satisfy the smoothness assumption. It would be beneficial to conduct further analysis that incorporates the properties of the orthogonal constraints, leading to an equivalent formulation that fully aligns with the assumptions and supports the theoretical analysis.**
>
>  **Response.**
>
> 1. The sparse PCA problem, $\min_{X} f(X) + \dot{\rho} ||X||_1 + \iota(X)$, satisfies the continuity assumption in our analysis, with $h_2(x_2) = \dot{\rho} ||X||_1$. Please refer to L180–183 in the paper.
>
> 2. Assumptions 1.1–1.7 are satisfied by the sparse PCA problem. Consequently, the complexity bound for achieving an $\epsilon$-approximate critical point remains valid and is comparable to that of RADMM.
>
> 1. This paper provides a general optimization framework for solving linear-constrained problems, including sparse PCA as a special case. We focus on sparse PCA to facilitate comparisons with existing state-of-the-art techniques. Further exploration of the properties of orthogonal constraints to analyze the convergence behavior of ADMM is an interesting topic, which we leave for future work.

---

> ### Author Response · Authors · 2024-11-22
>
> **Question 3:  Given that the algorithm involves several parameters, is it straightforward to determine their values in practice? It would be helpful if additional experiments were conducted to demonstrate the robustness of the algorithm with respect to parameter selection and to provide guidance on how to choose these parameters effectively.**
>
>  **Response.** The algorithm generally involves six parameters: $(\beta^0, \theta_1, \theta_2, \sigma, \delta, \xi )$. However, we argue that these parameters are primarily selected to ensure the theoretical convergence of the algorithm, and the algorithm's performance is not highly sensitive to their specific values. We consider the sparse PCA problem.
>
> 1. The proximal parameter $\theta_1$ is typically set to a constant slightly greater than $1$, such as 1.01.
>
> 2. The proximal parameter $\theta_2$ is typically set to a constant slightly greater than $1/2+\chi_1$, such as $1/2+\chi_1+10^{-12}$.
>
> 3. The relaxation parameter $\sigma$ is suggested to be around the golden ratio $1.618$, as suggested by Min Li, Defeng Sun, and Kim-Chuan Toh in *"A Majorized ADMM with Indefinite Proximal Terms for Linearly Constrained Convex Composite Optimization" (SIOPT, 2016)*.
>
> 4. The parameter $\delta$ only effects $\mu^t$ through $\mu^t = 1/ (\overline{\lambda} \delta \beta^t)$, where $\delta$ is a sufficiently small constant, and $\beta^t$ can be relatively large. Thus, the choice of $\delta$ is not expected to significantly affect the overall performance of the proposed method.
>
> 5. The parameter $\xi$ is typically set to a constant $0.5$.
>
> 6. In our revised paper, we present the convergence curves of the compared methods with varying $\dot{\rho} \in (1, 10, 100, 1000)$ and $\beta^0 = (10\dot{\rho}, 50 \dot{\rho}, 100 \dot{\rho}, 500 \dot{\rho})$, as shown in Figures 2 to 17. Please refer to Page 36-Page 43 of the manuscript.

---

> > ### Comment · Reviewer_KMzj · 2024-11-25
> >
> > Thanks the authors for the detailed response. The additional numerical results exploring different parameter choices enhance the alignment with the theoretical findings and demonstrate the robustness of the proposed algorithm. Based on this, I would like to increase my score.

---

### Meta-Review · Area_Chair_v3T4 · 2024-12-14

**Metareview:**

This paper designs ADMM for solving multi-block nonconvex composite optimization problems. The proposed approach requires less stringent conditions comparing with existing work. An improved oracle complexity is also established for the proposed method. Numerical results demonstrate the effectiveness of the proposed algorithm. While the ideas are interesting and the results are new, the authors are advised to improve the readability of the paper, as pointed out by the reviewers.

**Additional Comments On Reviewer Discussion:**

Additional numerical results were added. Further clarification on parameter selection.

---

### Decision · Program_Chairs · 2025-01-22

Accept (Poster)